**Tropical Pacific Climate Variability under Solar Geoengineering: Impacts on ENSO**
**Extremes**

**Abdul Malik[1,2,3], Peer J. Nowack[1,4,5,6], Joanna D. Haigh[1,4], Long Cao[7], Luqman Atique[7],**
**Yves Plancherel[1]**

[1]Grantham Institute – Climate Change and the Environment, Imperial College London,
London, United Kingdom
[2]Oeschger Centre for Climate Change Research, and Institute of Geography, University of
Bern, Bern, Switzerland
[3]4700 King Abdullah University of Science and Technology, Thuwal 23955-6900, Kingdom
of Saudi Arabia
[4]Department of Physics, Blackett Laboratory, Imperial College London, United Kingdom
[5]Data Science Institute, Imperial College London, United Kingdom
[6]School of Environmental Sciences, University of East Anglia, Norwich, United Kingdom
[7]School of Earth Sciences, Zhejiang University, Hangzhou, China
*Correspondence to:* Abdul Malik (abdul.malik@kaust.edu.sa)

**Abstract**

Many modelling studies suggest that the El Niño Southern Oscillation (ENSO), in interaction
with the tropical Pacific background climate, will change with rising atmospheric greenhouse
gas concentrations. Solar geoengineering (reducing the solar flux from outer space) has been
proposed as a means to counteract anthropogenic climate change. However, the effectiveness
of solar geoengineering concerning a variety of aspects of Earth's climate is uncertain. Robust
results are particularly challenging to obtain for ENSO because existing geoengineering
simulations are too short (typically ~50-year) to detect statistically significant changes in the
highly variable tropical Pacific background climate. We here present results from a 1000-year
long solar geoengineering simulation, G1, carried out with the coupled atmosphere-ocean
general circulation model HadCM3L. In agreement with previous studies, reducing the solar
irradiance (4%) to offset global mean surface warming in the model more than compensates
the warming in the tropical Pacific that develops in the $4\times CO_2$ scenario. We see an
overcooling of $0.3^{\circ}$C and a 0.23-mm day$^{-1}$ (5%) reduction in mean rainfall over tropical
Pacific relative to preindustrial conditions in the G1 simulation, owing to the different
latitudinal distributions of the shortwave (solar) and longwave ($CO_2$) forcings. The location
of the Intertropical Convergence Zone (ITCZ) in the tropical Pacific, which moved $7.5^{\circ}$
southwards under $4\times CO_2$, is restored to its preindustrial position. However, other aspects of
the tropical Pacific mean climate are not reset as effectively. Relative to preindustrial
conditions, in G1 the time-averaged zonal wind stress, zonal sea surface temperature (SST)
gradient, and meridional SST gradient are each statistically significantly reduced by around
10%, and the Pacific Walker Circulation (PWC) is consistently weakened resulting in
conditions conducive to increased frequency of El Niño events. The overall amplitude of
ENSO strengthens by 9-10% in G1, but there is a 65 % reduction in the asymmetry between
cold and warm events: cold events intensify more than warm events. Notably, the frequency
of extreme El Niño and La Niña events increases by ca. 60% and 30%, respectively, while
the total number of El Niño events increases by around 10%. All of these changes are

statistically significant either at 95 or 99% confidence level. Somewhat paradoxically, while the number of total and extreme events increases, the extreme El Niño events become weaker relative to the preindustrial state while the extreme La Niña events become even stronger. That is, such extreme El Niño events in G1 become less intense than under preindustrial conditions, but also more frequent. In contrast, extreme La Niña events become stronger in G1, which is in agreement with the general overcooling of the tropical Pacific in G1 relative to preindustrial conditions.

## 1 Introduction and Background

Since the industrial revolution, anthropogenic emissions of Greenhouse Gases (GHGs) have led to globally increasing surface temperatures (Stocker, 2013). Higher temperatures, in turn, and more generally a rapidly changing climate, can have adverse effects on humans, plants, and animals through changes in various ecosystems, rising sea levels, melting glaciers, and could significantly impact the frequency and intensity of extreme weather events (Moore et al., 2015). Various strategies, principally a reduction of GHG emissions and enhancements of carbon dioxide sinks (Pachauri et al., 2014), have been proposed to mitigate anthropogenic climate change. Another group of strategies involves the intentional modification of Earth's radiation balance on a global scale, known as solar geoengineering (Crutzen, 2006; Wigley, 2006; Curry et al., 2014). For any serious consideration of such geoengineering strategies, it is essential to understand their potential perils as well as benefits. One route to study the potential impacts of geoengineering on various components of Earth's climate system (e.g., atmosphere, ocean, cryosphere, etc.) is through employing state-of-the-art coupled atmosphere-ocean general circulation models (AOGCMs).

In this context, Kravitz et al. (2011) proposed the Geoengineering Model Intercomparison Project (GeoMIP), which initially consisted of a set of four experiments (viz. G1, G2, G3, and G4). These experiments are designed to investigate the effects of geoengineering on the regional and global climate when it is implemented to offset the annual mean global radiative forcing at the top of the Earth's atmosphere introduced by GHGs. These experiments are collectively called Solar Radiation Management (SRM) or solar geoengineering (Kravitz et al., 2013a). In the G1 experiment, atmospheric $CO_2$ is instantaneously quadrupled, but the global GHG-induced longwave radiative effects are offset by a simultaneous reduction in the shortwave Total Solar Irradiance, TSI, (Kravitz et al., 2011). In terms of radiative forcing, the quadrupling of $CO_2$ is similar to the year 2100 in the RCP8.5 emission scenario (Representative Concentration Pathway with a radiative forcing of 8.5 W m$^{-2}$ by the year 2100; Schmidt et al., 2012). In this paper, we focus on the G1 experiment to investigate how effectively solar geoengineering could mitigate the effects of substantial changes in atmospheric $CO_2$ on the tropical Pacific climate.

The El Niño Southern Oscillation (ENSO) is an important coupled ocean-atmosphere mode of interannual variability in the tropical Pacific (Park et al., 2009; Vecchi and Wittenberg 2010), which affects both regional and global climate (see Ropelewski and Halpert, 1987; Bove et al., 1998; Malik et al., 2017). ENSO oscillates between a warm, El Niño, and a cold, La Niña, phase every 2-7-year (Santoso et al., 2017). Based on Empirical Orthogonal

Function Analysis (EOF) of Sea Surface Temperature (SST) in the tropical Pacific (see Takahashi et al., 2011), ENSO can be contrasted into two distinct modes of variability, i.e. eastern and central Pacific ENSO modes (Kao and Yu, 2009; Yu and Kim, 2010; Xie and Jin, 2018). The eastern Pacific ENSO mode (EOF1) shows maximum SST anomaly in the eastern equatorial Pacific (Niño3 region: $5^o$N-$5^o$S; $150^o$W-$90^o$W) whereas the central Pacific ENSO mode (EOF2) indicates maximum SST anomaly in the central Pacific (Niño4 region: $5^o$N-$5^o$S; $160^o$E-$150^o$W) (Kao and Yu, 2009; Cai et al., 2018).

As diagnosed from SST indices in state-of-the-art AOGCMs, there was no intermodel consensus about change in frequency of ENSO events and amplitude in a warming climate (Vega-Westhoff and Sriver, 2017; Yang et al., 2018). However recently, Cai et al. (2018), using SST indices based on Principal Component Analysis (PCA), showed an enhanced frequency of extreme El Niño events and strengthening of ENSO amplitude under increased GHG forcing. However, before that, Cai et al. (2014 and 2015b) also showed evidence of a doubling of El Niño and La Niña events in the Coupled Model Intercomparison Project (CMIP) phases 3 (A2 scenario) and 5 (RCP8.5) by investigating a performance-based subset of models using rainfall-based ENSO indices instead of SST-based indices. Similarly, Wang et al. (2017) also reported a doubling of extreme El Niño events, relative to the preindustrial level, in the RCP2.6 transient scenario a century after stabilization of global mean temperature. Chen et al. (2017), analysing 20 CMIP5 models (RCP8.5), found both strengthening (in 6 models) and weakening (in 8 models) of ENSO amplitude. However, Cai et al. (2018) later found robust evidence of a consistent increase in El Niño amplitude in the subset of CMIP5 climate models, which were capable of simulating both eastern and central Pacific ENSO modes. In summary, changes in ENSO characteristics such as amplitude and ENSO extremes are projected in a warming climate (e.g., Cai et al., 2014, 2015b, 2018; Kim et al., 2014; Wang et al., 2018).

Increasing GHGs have distinct effects on the tropical Pacific mean climate. In CMIP3 and CMIP5 simulations, the equatorial tropical Pacific consistently shows a significant mean state warming response to increased GHG forcing (van Oldenborgh et al., 2005; Collins et al., 2010; Vecchi and Wittenberg 2010; Huang and Ying 2015; Luo et al., 2015). CMIP3 and CMIP5 models generally show more warming on than off-equatorial tropical Pacific (Liu et al., 2005; Collins et al., 2010; Cai et al., 2015a). Consistent with these warming patterns, studies typically found a weakening of zonal SST gradient (ZSSTG), Pacific Walker Circulation (PWC), zonal wind stress, and a shoaling of the equatorial tropical Pacific thermocline (see van Oldenborgh et al., 2005; Latif et al., 2009; Park et al., 2009; Yeh et al., 2009; Collins et al., 2010; Kim et al., 2014; Cai et al., 2015a; Zhou et al., 2015; Coats and Karnauskas 2017; Vega-Westhoff and Sriver 2017). Changes in the mean state of the tropical Pacific can bring about variations in ENSO properties such as amplitude, frequency, and spatial pattern (Collins et al., 2010; Vecchi and Wittenberg, 2010; Cai et al., 2015a).

We note that a previous study by Guo et al. (2018) found no statistically significant change in the intensity of Walker Circulation in GeoMIP models when comparing preindustrial simulations to the G1 experiment. Similarly, Gabriel and Robock (2015) found no statistically significant change in frequency and amplitude of ENSO events under both global

warming and geoengineering scenarios in 6 GeoMIP models that captured ENSO variability best. However, these authors themselves highlighted the length of their simulations (~50 years) as a key constraint for their studies. They suggested that long term simulations (>50 years) would be required to detect possible ENSO changes. Guo et al. (2018) concluded that 60 or more years of model simulations are required to detect changes in the PWC, while Vecchi et al. (2006) and Vecchi and Soden (2007) argued that 130-yrs are necessary to identify any robust change in the PWC (Gabriel and Robock, 2015). Similarly, Stevenson et al. (2010) estimated that 250 years are needed to detect changes in ENSO variability with a statistical significance of 90%. Here we aim to address this gap in the literature and establish a baseline for future studies through the analysis of long-term (1000 year) simulations of a single climate model.

Here, we employ three 1000-year long climate model simulations (preindustrial forcing, abrupt-4xCO$_2$ forcing, and G1) to estimate the efficacy of solar geoengineering in resetting the tropical Pacific circulation. Specifically, we investigate: (1) if solar geoengineering can mitigate the changes in mean tropical Pacific climate found in previous GHG warming studies, and even bring it back to the preindustrial conditions; (2) if ENSO frequency and amplitude are different under G1 conditions than under preindustrial simulations; and (3) if the G1 experiment reduces the increase in the frequency of extreme ENSO events, as shown by Cai et al. (2014, 2015b and 2018), under increased GHG forcing, relative to the preindustrial state. For this purpose, we are primarily interested in the more subtle differences in climate between G1 and preindustrial conditions, but also consider the profound changes under 4xCO$_2$ where, by design, the global mean surface temperature is much higher, and thus many other climate aspects vastly differ from the other two scenarios.

Section 2 describes the climate model HadCM3L, the data and the statistical methods used to detect changes in tropical Pacific and ENSO variability. The same section also evaluates the capability of HadCM3L to model ENSO. Section 3 evaluates the response of a list of metrics used to understand how the mean state and ENSO variability are affected in different experiments (preindustrial, 4xCO$_2$, G1). Section 4 elaborates on the mechanism of ENSO variability under GHG forcing and solar geoengineering for the given model system. Finally, Section 5 presents the discussion and conclusions.

## 2 Data and methods

### 2.1 Climate model

HadCM3L (Cox et al., 2000) has a horizontal resolution of 2.5$^o$ latitude × 3.75$^o$ longitude (~T42) with 19 (L19) atmospheric and 20 (L20) ocean levels. HadCM3L stems from the family of HadCM3 climate models; the only difference is lower ocean resolution (HadCM3: 1.25$^o$ × 1.25$^o$; Valdes et al., 2017). In HadCM3L, land surface processes are simulated by the MOSES-2 module (Essery and Clark, 2003; Cao et al., 2016). HadCM3L does not include an interactive atmospheric chemistry scheme and thus does not consider effects of ozone changes on ENSO amplitude and surface warming under 4xCO$_2$ (e.g., Nowack et al., 2015; 2017, 2018) or G1 (e.g., Nowack et al., 2016). Instead, we use preindustrial background

ozone climatology, prescribed on pressure levels. In section 2.4, we evaluate the ability of HadCM3L to model ENSO. We acknowledge that some of our results will necessarily be model-dependent, and underline the need for similar studies with other climate models. Still, by using much longer simulations than used previously, our results provide statistical robustness for the given model system.

## 2.2 Simulations and observational data

Here, we use HadCM3L simulations carried out by Cao et al. (2016). To achieve a quasi-equilibrium preindustrial climate state, the model was spun up for 3000 years with constant $CO_2$ concentrations (280 ppmv; parts per million by volume) and TSI (1365 W m$^{-2}$). Then, three 1000-year long experiments were carried out, starting from this preindustrial climate state. These experiments are: (1) the preindustrial control (piControl) experiment with constant values of $CO_2$ (280 ppmv) and TSI (1365 W m$^{-2}$); (2) a quadrupled $CO_2$ (4×$CO_2$) experiment in which $CO_2$ is suddenly increased to 1120 ppmv; and (3) sunshade geoengineering (G1) experiment where the radiative effects of the instantaneously quadrupled $CO_2$ are offset by simultaneously reducing TSI (by 4%). All experiments follow the GeoMIP protocol (see Kravitz et al., 2011); the only difference being that simulations were run for 1000 years (see Cao et al., 2016) instead of 50 years as in GeoMIP.

The monthly SST dataset from HadISST (1$^o$ latitude × 1$^o$ longitude; Rayner et al., 2003) and the rainfall data from the Global Precipitation Climatology Project (GPCP; Adler et al., 2003) version 2.3 (2.5$^o$ latitude × 2.5$^o$ longitude) over the period 1979-2017 are used to provide observational constraints and to identify the rainfall threshold to be used for defining extreme El Niño events. Further, we use ERA5 reanalysis data (Copernicus Climate Change Service (C3S), 2017) covering years 1979-2019 to evaluate the capability of HadCM3L to simulate ENSO variability. ERA5 has a horizontal resolution of 0.25$^o$ latitude × 0.25$^o$ longitude. Specifically, we use monthly mean surface latent heat flux (lh), sensible heat flux (sh), net shortwave radiation flux (sw), net longwave radiation flux (lw), ocean temperature, and zonal and meridional components of wind stress.

## 2.3 Definitions and statistical tests

We analyse changes in the tropical Pacific (25$^o$ N-25$^o$ S; 90$^o$ E-60$^o$ W) mean climate. We present climatologies for SSTs, rainfall, Intertropical Convergence Zone (ITCZ), vertical velocity averaged between 500 and 100 hPa (Omega500-100), PWC, zonal wind stress, zonal and meridional SST gradients (ZSSTG and MSSTG, respectively), and thermocline depth. We calculate mean climatological differences for all these variables simulated under 4×$CO_2$ and G1 relative to piControl and assess their statistical significance using non-parametric Wilcoxon signed-rank and Wilcoxon rank-sum tests (Hollander and Wolfe, 1999; Gibbons and Chakraborti, 2011). All analyses are performed on re-gridded (2$^o$ longitude × 2.5$^o$ latitude) HadCM3L output for model years 11 to 1000 unless otherwise stated. The first ten years are skipped to remove the initially significant atmospheric transient effects stemming from instantaneously increasing $CO_2$ (see Kravitz et al., 2013b; Hong et al., 2017). Since ENSO events peak in boreal winter (December-January-February; DJF; Cai et al., 2014;

Gabriel and Robock 2015; Santoso et al., 2017), the entire analysis is performed for DJF, unless otherwise stated. Accordingly, we also analyse mean state changes in the tropical Pacific during boreal winter.

Both rainfall and SST-based ENSO indices are used in the present study. Niño3 ($5^o$ N-$5^o$ S; $150^o$ W-$90^o$ W) and Niño4 ($5^o$ N-$5^o$ S; $160^o$ E-$150^o$ W) indices are defined by averaging SST over corresponding ENSO regions. Normalised ENSO anomalies (i.e., the ENSO indices) are calculated relative to piControl mean and standard deviation (s.d.) and are quadratically detrended before analysis. The Niño3 index is chosen for studying the characteristics of extreme El Niño events since during an extreme El Niño event, following the highest SSTs, convective activity moves towards the eastern Pacific, and the ITCZ moves over the Niño3 region resulting in rainfall higher than 5 mm day$^{-1}$ (Cai et al., 2014). Similar to Cai et al. (2014, 2017), events with Niño3 rainfall greater than 5 mm day$^{-1}$ are considered extreme El Niño events, whereas events with Niño3 SST index greater than 0.5 s.d. and Niño3 rainfall less than 5 mm day$^{-1}$ are defined as moderate events unless otherwise stated. The Niño4 index is chosen for studying the characteristics of extreme La Niña events since maximum cold temperatures occur in this region (Cai et al., 2015a, 2015b). La Niña extreme (Niño4 < -1.75 s.d.), moderate ( -1 > Niño4 > -1.75), and weak (-0.5 > Niño4 > -1) events are defined following Cai et al. (2015b). These definitions classify the 1988 and 1998 La Niñas in observations as extreme events (see Cai et al., 2015b), and HadCM3L can capture such extreme anomalies (see Sect. 3.2), which allows us to study changes in their number and magnitude.

To understand the mechanisms responsible for changes in ENSO variability, we have calculated ENSO feedbacks (e.g., Bjerkness (BJ) and heat flux (hf) feedbacks) and ocean stratification. BJ feedback is a dynamical response of equatorial zonal wind stress to equatorial SST anomalies. It is positive feedback that maintains the ZSSTG (Lloyd et al., 2011). Here, we calculate the BJ feedback by point-wise linear regression (Bellenger et al., 2014) of the zonal wind stress anomalies over the entire equatorial Pacific ($5^o$ N-$5^o$ S; $120^o$ E-$80^o$ W; Kim and Jin 2011a; Ferret and Collins 2019) onto the eastern equatorial Pacific ($5^o$ N-$5^o$ S; $180^o$ W-$80^o$ W; Kim and Jin 2011a; Ferret and Collins 2019) SST anomalies. We then define the BJ feedback as the mean regression coefficient (Bellenger et al., 2014) over the eastern equatorial Pacific region. The hf feedback is a regression coefficient calculated by point-wise linearly regressing the net surface heat flux (sum of sw, lw, lh, and sh) anomalies into the ocean onto the SST anomalies over the eastern equatorial Pacific ($5^o$ N-$5^o$ S; $180^o$ W-$80^o$ W; Kim and Jin 2011a). This regression coefficient is also termed as a thermal damping coefficient (Kim and Jin, 2011a). It is negative feedback in which an initial positive SST anomaly causes a reduced surface net heat flux into the ocean, thus lessening the initial SST anomaly (Lloyd et al., 2011). Ocean stratification is defined as the difference in the volumetric average of ocean temperatures over the upper 67 m, and the temperature of a single ocean layer at 95 m, both spatially averaged over the region, $5^o$ N-$5^o$ S; $150^o$ E-$140^o$ W, where strong zonal wind stress anomalies also occur (see Fig. 4a and Fig. S1; Cai et al., 2018).

Following Cai et al. (2014), the statistical significance of the change in the frequency of ENSO events is tested using a bootstrap method with 10,000 realisations for the piControl data. We then find the s.d. of events over these 10,000 realisations. If the difference of events of piControl with $4\times CO_2$ and G1 is larger than 2 s.d., the change in frequency is considered statistically significant. The same method is used for testing the statistical significance of a change in ENSO amplitude, ZSSTG, MSSTG, ENSO amplitude asymmetry, ENSO feedbacks, and ocean stratification. All changes in $4\times CO_2$ and G1 are described relative to piControl.

## 2.4 ENSO representation in HadCM3L

Before employing HadCM3L for studying ENSO variability under $4\times CO_2$, and G1, we evaluate its piControl simulation against present-day observational data. There is a non-linear relationship between tropical Pacific SST and rainfall (Ham, 2017), which can be diagnosed by Niño3 region rainfall skewness (Cai et al., 2014). Skewness is a measure of asymmetry around the mean of the distribution (see eq. S1). Positive skewness means that in given data distribution, the tail of the distribution is spread out towards high positive values, and vice versa (Ghandi et al., 2016). The skewness criterion is used to exclude climate models simulating overly wet or dry conditions over the Niño3 region (Cai et al., 2017). During extreme El Niño events, the ITCZ moves equatorward, causing significant increases in rainfall (> 5 mm day$^{-1}$) over the eastern equatorial Pacific that skews the statistical distribution of rainfall in the Niño3 region. Thus, for studying extreme ENSO events, the model should be capable of simulating Niño3 rainfall above 5 mm day$^{-1}$ and Niño3 rainfall skewness of greater than 1 over the entire simulated period (see our Sect. 3.2.2, and Cai et al., 2014 and 2015b). With a Niño3 rainfall skewness of 2.06 for piControl, HadCM3L fulfils this criterion.

In addition, we evaluate the ENSO modelled by HadCM3L following a principal component (PC) approach suggested by Cai et al. (2018). Considering distinct eastern and central Pacific ENSO regimes based on EOF analysis, they found that climate models capable of simulating present-day ENSO diversity show a robust increase in eastern Pacific ENSO amplitude in a greenhouse warming scenario. Specifically, the approach assumes that any ENSO event can be represented by performing EOF analysis on monthly SST anomalies and combining the first two principal patterns (Cai et al., 2018). The first two PCs time series, PC1 and PC2, show a non-linear relationship in observational datasets (Fig. S1m). Climate models that do not show such a non-linear relationship cannot satisfactorily simulate ENSO diversity, and hence are not sufficiently skilful for studying ENSO properties (Cai et al., 2018). Here, we perform EOF analysis on quadratically detrended monthly SST and wind stress anomalies of ERA5 and piControl over a consistent period of 41-year. We evaluate HadCM3L's ability to simulate two distinct ENSO regimes and the non-linear relationship between the first two PCs, i.e., $PC2(t) = \alpha[PC1(t)]^2 + \beta[PC1(t)]^2 + \gamma$ (Fig. S1). From ERA5, $\alpha = -0.36$ (statistically significant at 99% confidence level, hereafter "cl") whereas in piControl $\alpha = -0.31$ (99% cl), which is same as the mean $\alpha = -0.31$ value calculated by Cai et al. (2018) averaged over five reanalysis datasets. The 1$^{st}$ and 2$^{nd}$ EOF patterns of monthly SST and wind stress anomalies of piControl (Fig. S1 b, e) are comparable with that of ERA5 (Fig. S1 a, d). EOF1 of

piControl shows slightly stronger warm anomalies in the eastern equatorial Pacific, whereas negative anomalies over the western Pacific are slightly weaker compared to ERA5. In EOF1, the stronger wind stress anomalies occur to the west of the Niño3 region, which is a characteristic feature during the eastern Pacific El Niño events (see Kim and Jin, 2011a). Compared to ERA5, the spatial pattern of warm eastern Pacific anomalies is slightly stretched westwards, and wind stress anomalies are relatively stronger over the equator and South Pacific Convergence Zone (SPCZ). The $2^{nd}$ EOF, in both ERA5 and piControl, shows warm SST anomalies over the equatorial central Pacific Niño4 region. The variance distributions for ERA5 and HadCM3L match well for EOF1 (ERA5: 82%, piContol: 90%) whereas a large difference exist for EOF2 (ERA5: 18%, piControl: 10%).

The PCA is also useful for evaluating how well HadCM3L represents certain types of ENSO events. Eastern and central Pacific ENSO events can be described by an E-Index (PC1-PC2)/$\sqrt{2}$; Takahashi et al., 2011), which emphasises maximum warm anomalies in the eastern Pacific region (Cai et al., 2018), and a C-Index (PC1+PC2)/$\sqrt{2}$; Takahashi et al., 2011) respectively, which focuses on maximum warm anomalies in the central Pacific (Cai et al., 2018). Here, we show the eastern Pacific (EP) Pattern (Fig. S1 g, h) and central Pacific (CP) pattern (Fig. S1 j, k) by linear regression of mean DJF E- and C-Index, respectively, onto mean DJF SST and wind stress anomalies. We find that model's EP and CP patterns agree reasonably well with that of ERA5. HadCM3L underestimates the E-index skewness (1.16) whereas overestimates the C-Index skewness (-0.89) compared to ERA5 (2.08 and -0.58, respectively) averaged over DJF. HadCM3L's performance averaged over the entire simulated period of piControl is also consistent with ERA5 (Fig. S1; α: -0.32, EOF1: 64%, EOF2, 8%, E-index skewness: 1.30, C-index skewness: -0.42). In general, in HadCM3L, the contrast between the E- and C-index skewness over the entire simulated period is sufficient enough to differentiate relatively strong warm (cold) events in the eastern (central) equatorial Pacific compared to the central (eastern) equatorial Pacific. Finally, we also evaluated the hf and BJ feedbacks which, for piControl, are very similar to those of ERA5 (Table S5-6).

We conclude that HadCM3L has a reasonable skill for studying long-term ENSO variability and its response to solar geoengineering. However, we also highlight the need for and hope to motivate future modelling studies that will help identify model dependencies in the ENSO response.

## 3 Results

### 3.1 Changes in the tropical Pacific mean state

In this section, we analyse several significant changes in the tropical Pacific mean state under $4xCO_2$ and G1. In particular, we look into meridional and zonal SST changes, corresponding surface wind responses, and coupled variations in the thermocline depth. Our analysis reveals that this leads to significant changes in the precipitation climatology among the simulations. Finally, we find consistent effects on the PWC. All these results are important not just as general climatic features but also because they are mechanistically linked to changes in ENSO extremes discussed in detail in Sect. 3.2.

### 3.1.1 Sea surface temperature

Tropical Pacific SSTs are spatially asymmetric along the equator. The western equatorial Pacific (warm pool) is warmer on average than the eastern equatorial Pacific (cold tongue) (Vecchi and Wittenberg, 2010). The piControl simulation (Fig. 1a) reasonably simulates the SST asymmetry between the western and eastern equatorial Pacific well (cf. Fig 1a in Vecchi and Wittenberg, 2010). Under $4 \times CO_2$, the SST zonal asymmetry is significantly reduced (Fig. 1b), and the entire equatorial tropical Pacific shows a warming state (e.g., Meehl and Washington, 1996; Boer et al., 2004). The solar dimming in G1 largely offsets the warming seen under $4 \times CO_2$ and brings the tropical Pacific mean SSTs close to the preindustrial state (Fig. 1c). The SPCZ, where the highest SSTs of the warm pool occur (Cai et al., 2015a; blue line in Fig. 1a), moves towards the equator under $4xCO_2$ (blue line, Fig. 1b), but returns to approximately its preindustrial position in G1 (Fig. 1c).

The tropical Pacific is 3.90$^{o}$C warmer in $4 \times CO_2$ but 0.30 $^{o}$C colder in G1, with both differences being significant at the 99% cl (see Fig. 1d-e, Table S1). The Pacific cold tongue warms more rapidly than the Pacific Warm Pool under $4 \times CO_2$. In contrast, in G1, a stronger cooling occurs in the Pacific Warm Pool and the SPCZ than in the cold tongue region. The Pacific Warm Pool is ~0.4-0.6 $^{o}$C colder in G1, whereas the east Pacific cools less (~-0.2 $^{o}$C in the Niño3 region), indicating a change in SST asymmetry under G1.

Our SST results under $4xCO_2$ qualitatively agree with previous studies (Liu et al., 2005; van Oldenborgh et al., 2005; Collins et al., 2010; Vecchi and Wittenberg, 2010; Cai et al., 2015a; Huang and Ying, 2015; Luo et al., 2015; Kohyama et al., 2017; Nowack et al., 2017). Overcooling of the tropics (and as such, the tropical Pacific) is a robust signal in G1 simulations, even short ones, simply due to the different meridional distribution of shortwave and longwave forcing (Govindasamy and Caldeira, 2000; Lunt et al., 2008; Kravitz et al., 2013b; Curry et al., 2014; Nowack et al., 2016). The results presented here based on a long simulation not only corroborate previously published findings but also statistically demonstrate that under G1, the Warm Pool and SPCZ cool faster than the cold tongue.

### 3.1.2 Precipitation

In the tropical Pacific, there are three dominant bands of rainfall activity: one in the western Pacific Warm Pool, one in the SPCZ, and the last one along the ITCZ situated at around 8$^{o}$ N and 150$^{o}$ W-90$^{o}$ W. Further, the eastern equatorial Pacific is relatively dry compared with these three rainy bands (cf. Fig. 2a Sun et al. 2020). Under piControl, HadCM3L simulates these spatial rainfall patterns well, with maxima of ~6-8, ~12-14, and ~8-10 mm day$^{-1}$ over the Pacific Warm Pool, the SPCZ, and the ITCZ, respectively (Fig. 2a). Under $4 \times CO_2$, the spatial rainfall pattern changes significantly. The ITCZ moves equatorward, and the SPCZ becomes zonally oriented (blue line, Fig. 2b). The rainfall asymmetry between the western and eastern equatorial Pacific decreases under $4 \times CO_2$. Precipitation migrates from the west Pacific to the Niño3 region, with maximum rainfall at ~145$^{o}$ W. The reduced zonal asymmetry in the rainfall between western and eastern Pacific is effectively restored to the preindustrial state in G1 (Fig. 2c).

A statistically significant (99% cl) overall precipitation increase of 0.21 mm day$^{-1}$ (+5%) is seen over the tropical Pacific under 4×$CO_2$ (Fig. 2d). In contrast, the mean rainfall in G1 decreases by 0.23 mm day$^{-1}$ (-5%; Fig. 2e), consistent with the simulated reduction in temperature (-0.30 $^o$C) over the tropical Pacific. However, there is a strong regional structure: under 4×$CO_2$, rainfall decreases to a maximum of ~3 mm day$^{-1}$ over parts of the Pacific Warm Pool and off-equatorial regions, whereas a significant increase of ~15-18 mm day$^{-1}$ develops over the Niño3 region. An overall increase in mean rainfall under the GHG warming scenario has also been reported in many previous studies (e.g., Watanabe et al., 2012; Power et al., 2013; Chung et al., 2014; Nowack et al., 2016). Under G1, rainfall decreases over the Pacific Warm Pool, SPCZ, and ITCZ regions. In contrast, rainfall increases significantly over most parts of central and eastern equatorial Pacific, with a maximum (~ 1.5-2 mm day$^{-1}$) centred at ~150$^o$ W (Fig. 2e). Kravitz et al. (2013b) reported a decrease of 0.2 mm day$^{-1}$ over the tropical regions. Under G1, the magnitude of the lapse rate decreases, resulting in increased atmospheric stability and hence suppressed convection, which leads to an overall reduction of rainfall over the tropics (Bala et al., 2008; Kravitz et al., 2013b).

The position of the ITCZ over the tropical Pacific (25$^o$ N-25$^o$ S; 90$^o$ E-60$^o$ W) is calculated by finding the latitude of maximum rainfall (blue lines, Fig. 2a-e). The median position of this maximum ITCZ (from 154$^o$ W-82$^o$ W) is 7.5$^o$ N, 0$^o$, and 7.5$^o$ N under piControl, 4×$CO_2$, and G1, respectively. Thus, under 4×$CO_2$, the ITCZ mean position shifts over the equator and is positioned within the Niño3 region. G1 restores the ITCZ and SPCZ to their preindustrial orientations. Still, differences in the magnitude of rainfall persist over these regions, as well as over the Pacific Warm Pool (Fig. 2a, c, e). That is, while the relative additional rainfall asymmetry between the western and eastern Pacific in 4×$CO_2$ is mostly resolved in G1, the tropical Pacific is overall wetter under 4×$CO_2$ but drier in G1.

### 3.1.3 Zonal wind stress

Changes in zonal wind stress are directly dependent on and interact with ENSO amplitude (Guilyardi, 2006), ENSO period (Zelle et al., 2005; Capotondi et al., 2006), and ZSSTG (Hu and Fedorov, 2016). A positive feedback loop between zonal wind stress, SST, and thermocline depth influences the evolution of ENSO (Philip and van Oldenborgh, 2006). A decrease in the strength of the trade winds is concurrent with a flattening of the thermocline, a reduction of upwelling in the eastern Pacific, and increased SST in the eastern relative to the western equatorial Pacific, thus resulting in further weakening of the trade winds (Collins et al., 2010). We use the zonal wind stress index, Westerly Wind Bursts (WWBs), and Easterly Wind Bursts (EWBs) to study the wind stress over the tropical Pacific. The zonal wind stress index is defined as the wind stress averaged over the equatorial tropical Pacific (5$^o$ N-5$^o$ S; 120$^o$ E-80$^o$ W). Although here not explicitly diagnosed through daily data, WWBs and EWBs are contained respectively in the positive and negative values of this wind stress index (see Hu and Fedorov, 2016). As the duration of WWBs is 5 to 40 days (Gebbie et al., 2007), the monthly mean data of westerly wind stress includes a monthly average of these bursts.

We find that the zonal wind stress is significantly reduced over most parts of the tropical Pacific, especially over the Niño3 region in both 4×CO$_2$ and G1 (Fig. 3a-e), in agreement with the reduced zonal SST gradients in both scenarios (Fig. 1). The zonal wind stress weakens by 31% and 10% in 4×CO$_2$ and G1 (statistically significant at 99% cl; Fig. 4a), respectively. We also see a considerable weakening of zonal wind stress over the Niño3 region, both under 4×CO$_2$ and G1. The strength of WWBs increases by 13% under G1 relative to piControl (99 % cl), while the EWBs decrease in strength by 7% (99% cl). In comparison, the strength of both the WWBs and EWBs is reduced (99% cl) under 4×CO$_2$, by 33% and 28%, respectively. The strong WWBs are more closely linked to positive SST anomalies than negative SST anomalies (Cai et al., 2015a) and thus are likely to increase the frequency of extreme El Niño events (Hu and Fedorov 2016) in G1, which is important with regards to the mechanistic interpretation of the ENSO changes below.

### 3.1.4 Zonal and meridional sea surface temperature gradients

The ZSSTG between western and eastern equatorial Pacific is one of the characteristic features of the equatorial tropical Pacific. The ZSSTG is weak during an El Niño and strong during La Niña events (Latif et al., 2009). The ZSSTG is calculated as the difference between SST in the western Pacific Warm Pool (5$^o$ N-5$^o$ S; 100$^o$ E-126$^o$ E) and eastern equatorial Pacific (Niño3 region: 5$^o$ N-5$^o$ S; 160$^o$ E-150$^o$ W). The zonal SST gradient is reduced both in 4xCO$_2$ and G1 (Fig. 4b, 99% cl), but the reduction is smaller in G1 (11%) than in 4xCO$_2$ (62%). The reduced zonal SST asymmetry in 4×CO$_2$ and G1 is consistent with the weakening of the trade winds and zonal wind stress, as noted in Sect. 3.1.3. The weakening of trade winds can result in reduced upwelling in the eastern equatorial Pacific, and east to west surface currents (Collins et al., 2010), leading to an increase in El Niño events. Our results under 4xCO$_2$ are in agreement with Coats and Karnauskas (2017), who using several climate models found a weakening of the ZSSTG under the RCP8.5 scenario.

MSSTG is calculated as the SST averaged over the off-equatorial region (5$^o$ N-10$^o$ N; 150$^o$ W-90$^o$ W) minus SST averaged over the equatorial region (2.5$^o$ N-2.5$^o$ S; 150$^o$ W-90$^o$ W) (Cai et al., 2014). Reversal of sign or weakening of the MSSTG has been observed during extreme El Niño events, as the ITCZ moves over the equator (e.g., Cai et al., 2014). Overall there is a change in sign and reduction of MSSTG in 4×CO$_2$ (~-111%, 99% cl) and only a slight decrease in G1 (~-9%, 99% cl) (Fig. S3, and Table S2). The decrease in strength of MSSTG is an indication that extreme El Niño events are expected to increase (Cai et al., 2014) under solar geoengineering. The weakening of the MSSTG is qualitatively in agreement with previous studies under increased GHG forcings (e.g., Cai et al., 2014; Wang et al., 2017).

### 3.1.5 Thermocline

Previous studies (e.g., Vecchi and Soden, 2007; Yeh et al., 2009) revealed shoaling as well as a reduction in the east-west tilt of the equatorial Pacific thermocline under increased GHG scenarios. A decrease in thermocline depth and slope is a dynamical response to reduced zonal wind stress. Shoaling of the equatorial Pacific thermocline can result in positive SST

anomalies in the eastern tropical Pacific, which in turn can affect the formation of El Niño (Collins et al., 2010).

Thermocline depth here is defined as the depth of the 20 $^{o}$C (for piControl and G1), and 24 $^{o}$C (for 4×$CO_2$) isotherms averaged between 5$^{o}$ N and 5$^{o}$ S, following Phillip and van Oldenborgh, (2006). Due to surface warming in GHG scenarios, the 20 $^{o}$C isotherm deepens (Yang and Wang et al., 2009), and this must be compensated by using a warmer isotherm (24 $^{o}$C) as a metric in the 4×$CO_2$ case.

 In 4x$CO_2$, the tropical Pacific thermocline depth (24 $^{o}$C isotherm) shoals by 22% (99% cl, Fig. 4c), as expected from similar experiments (Vecchi and Soden, 2007; Yeh et al., 2009). However, there is no statistically significant change in the mean thermocline depth in G1. In 4x$CO_2$, most likely the weakened easterlies (as noticed in Sect. 3.1.3; e.g., Yeh et al., 2009, Wang et al., 2017) and greater ocean temperature stratification due to increased surface warming (see Sect. 4 and Cai et al., 2018) lead to a significant shoaling of the thermocline across the western and central equatorial Pacific. In contrast, relatively little change takes place between 130$^{o}$ W and 90$^{o}$ W. In a CMIP3 multimodel (SRESA1B scenario) ensemble, Yeh et al. (2009) found a more profound deepening of the thermocline in this part of the eastern equatorial Pacific; however, for example, Nowack et al. (2017) did not find such changes under 4x$CO_2$ (cf. their Fig. S9). One possible explanation for this behaviour is the competing effects of upper-ocean warming (which deepens the thermocline) and the weakening of westerly zonal wind stress, causing thermocline shoaling (see Kim et al. 2011a).

### 3.1.6 Vertical velocity and Walker circulation

Under normal conditions, there is strong atmospheric upwelling over the western equatorial Pacific, SPCZ, and ITCZ. In contrast, the relatively cold and dry eastern Pacific is dominated by atmospheric downwelling. This process, as simulated in HadCM3L, can be seen in maps of Omega500-100 (Fig. 5a). The region of ascent over the SPCZ and ITCZ moves equatorward in 4×$CO_2$ (Fig 5b), consistent with the increase in SST and precipitation over the equatorial region (Fig. 1d and 2d). The convective centre also moves towards the Niño3 region and centres at ~150$^{o}$ W. While these changes in spatial patterns of atmospheric divergence and convergence are found to be corrected for G1 (Fig. 5c), significant differences in the strength of the atmospheric circulation remain, which in turn are coupled to the aforementioned changes in atmospheric stability. Specifically, both for 4×$CO_2$ and G1, upwelling decreases over the Warm Pool, but increases in the central Pacific and the eastern part of the Niño3 region (Fig. 5d-e). This picture is consistent with changes in the spatial extent and a weakening of the tropical PWC (Fig. 6a-c). In 4x$CO_2$, the weakening and shifting of circulation patterns are consistent with multimodel results reported by Bayr et al. (2014) under GHG forcing. While mitigated, the PWC weakening found in G1 remains highly statistically significant (99% cl; Fig. 6d-e).

## 3.2 ENSO amplitude and frequency

In Sect. 3.1, we described a variety of coupled, and highly significant changes in the tropical Pacific mean state, such as the weakening of zonal and meridional SST gradients, zonal wind stress, and PWC. It is well-known that such changes can affect ENSO variability. This section discusses various metrics used to characterise ENSO variability and unfolds how they change in $4xCO_2$ and G1. Specifically, we investigate the amplitude of ENSO, changes in amplitude asymmetry between El Niño and La Niña events, and ENSO frequency.

### 3.2.1 ENSO amplitude

To characterise changes in ENSO, this study uses two separate indices for two different regions, because extreme warm and cold events are not mirror images of each other (Cai et al., 2015b). The Niño3 (Niño4) index is employed for studying characteristics of El Niño (La Niña) events in the eastern (central) Pacific region. ENSO amplitude is defined as the standard deviation of SST anomalies in a given ENSO region (e.g., Philip and van Oldenborgh 2006; Nowack et al., 2017). The maximum amplitude of warm events is defined as the maximum positive ENSO anomaly during the entire time series analysed (Gabriel and Robock, 2015). Cold events are defined similarly, but using the maximum negative ENSO anomaly.

In $4\times CO_2$, both eastern and central Pacific ENSO amplitudes undergo a statistically significant decrease (47 and 64%, respectively, at 99% cl, Table 1-2). The maximum amplitude of warm events in the eastern Pacific and cold events in the central Pacific are also significantly reduced (57% and 36% at 99% cl, respectively; Table 3-4). Previous studies found that climate models produced mixed responses (both increases and decreases in amplitude) in terms of how ENSO amplitude change with global warming (see Latif et al., 2009; Collins et al., 2010; Vega-Westhoff and Sriver, 2017). However, Cai et al. (2018) found an intermodel consensus, for models capable of simulating ENSO diversity, for strengthening of ENSO amplitude under A2, RCP4.5, and RPC8.5 transient scenarios. In contrast, in G1, the eastern Pacific ENSO amplitude gets strengthened (9% at 99% cl), and no statistically significant change is noticed in the central Pacific ENSO amplitude.

Further, the maximum amplitude of cold events is strengthened in the central Pacific (20% at 99% cl), but no statistically significant change occurs in the eastern Pacific. A validation of these changes in ENSO amplitude using the E- and C-indices, as these indices represent SST anomalies similar to those of Niño3 and Niño4 index (Cai et al., 2015a), yields indeed very identical results (see Table 1-4). Thus, our simulations imply that significant changes can occur in ENSO events under solar geoengineering. Mechanistically, it is self-evident that these changes might be linked to the tropical Pacific SST overcooling of ca. 0.30 $^{o}$C and the substantial SST gradient changes under G1 relative to piControl.

However, the use of standard deviations to define ENSO amplitude is suboptimal, because amplitudes of El Niño and La Niña events are asymmetric, i.e., in general, El Niño events are stronger than La Niña events (An and Jin, 2004; Schopf and Burgman, 2006; Ohba and Ueda, 2009; Ham, 2017). The relative strength of ENSO warm and cold events can be measured by

the skewness of SST over the ENSO regions (Vega-Westhoff and Sriver, 2017). Following Ham (2017), we investigate the asymmetry in the amplitude of El Niño and La Niña events by comparing the skewness of detrended Niño3 SST anomalies in piControl with $4\times CO_2$ and G1.

We find that, relative to piControl, the Niño3 SST skewness is reduced both in $4\times CO_2$ (190% at 99% cl) and G1 (65% at 99% cl) (Table 5). The E-Index also indicates reduced skewness under both $4\times CO_2$ (85%) and G1 (28%) at 99% cl. The reduced skewness is further illustrated in maps showing differences in skewness between $4\times CO_2$ and G1 with piControl (Fig. S4). Over the eastern equatorial Pacific, the SSTs are transformed from positively to negatively skewed under $4\times CO_2$ (Fig. S4b). Our results qualitatively agree with Ham (2017), who found a 40% reduction in ENSO amplitude asymmetry using several CMIP5 models in the RCP4.5 scenario. In G1 (Fig. S4e), the skewness of SSTs is reduced over the eastern equatorial Pacific, whereas it strengthens over the central equatorial Pacific region (at 99% cl). The strengthening of skewness over the central equatorial Pacific is also consistent with increased C-Index skewness (66% at 99% cl) under G1 relative to piControl. Thus, due to the concurrent strengthening of the maximum amplitude of cold events and reduction in the asymmetry of SST skewness, the intensity of cold events is predicted to increase compared to warm events under solar geoengineering.

### 3.2.2 El Niño frequency

We choose a threshold value of rainfall for defining extreme El Niño events based on the work of Cai et al., (2014, 2017), who chose averaged DJF Niño3 total rainfall exceeding 5 mm day$^{-1}$ for this threshold based on observations. However, as pointed out by Cai et al. (2017), trends in Niño3 rainfall are mainly driven by two factors: (1) the change in the mean state of the tropical Pacific and (2) the change in frequency of extreme El Niño events. Therefore, since we want to focus on the changes in the extremes, we need to remove contribution (1) from the raw Niño3 time series. We, therefore, fit a quadratic polynomial to the time series of rainfall data from which all extreme El Niño events (DJF total rainfall > 5 mm day$^{-1}$) have been excluded and then subtract this trend from the raw Niño3 rainfall time series. Linearly detrending the rainfall time series produces similar results. Note that under piControl (observations), total rainfall of 5 mm day$^{-1}$ is ~85[th] (~93[rd]) percentile in detrended Niño3 rainfall time series. Wang et al. (2020) termed events with rainfall > 5 mm day$^{-1}$ as extreme convective El Niño events.

With detrended Niño3 total rainfall exceeding 5 mm day$^{-1}$ as an extreme, three extreme and seven moderate El Niño events can be identified from the historical record between 1979 and 2017 (Fig. 7a). A statistically significant increase of 526% (99% cl) in extreme El Niño events can be seen under $4\times CO_2$ (939 events) relative to piControl (150 events) (Fig. 7b-c). The geoengineering of climate (G1) largely offsets the increase in extreme El Niño frequency under $4\times CO_2$ (Fig. 7d), however, compared to piControl, still a 17% increase in extremes and a 12% increase in the total number of El Niño events (moderate plus extreme) can be seen at 95% cl. Thus, an El Niño event occurring every ~3.3-yr under preindustrial conditions occurs every ~2.9-yr under solar geoengineered conditions.

A threshold of detrended Niño3 total rainfall of 5 mm day$^{-1}$ recognises events as extremes even when the MSSTG is positive and stronger, especially under 4×CO$_2$, which plausibly means that ITCZ might not shift over the equator for strong convection to occur during such extremes. The El Niño event of 2015 is a typical example of such events. We test our results with a more strict criterion by choosing only those events as extremes, which have characteristics similar to that of 1982 and 1997 El Niño events (i.e., Niño3 rainfall > 5 mm day$^{-1}$ and MSSTG < 0). We declare events having characteristics similar to that of the 2015 event as moderate El Niño events (Fig. S5). Based on this method, we find a robust increase in the number of extreme El Niño events both in 4×CO$_2$ (924%) and G1 (61%) at 99% cl. We also performed the same analysis by linearly detrending the rainfall time series and find similar results (Fig. S6).

An alternative approach to quantifying extreme El Niño events is based on Niño3 SST index > 1.75 s.d. as an extreme event threshold (Cai et al., 2014). We note that using this definition, no statistically significant change in the number of extreme El Niño events is detected in G1 (61 events), whereas they reduced from 57 in piControl to zero events in 4×CO$_2$ highlighting the dependency of specific results on the precise definition of El Niño events used. However, relative to piControl, Niño3 SST index indicates a statistically significant increase (decrease) of 12% (46%) in the frequency of the total number of El Niño events (Niño3 SST index > 0.5 s.d.) (Table S3) in G1 (4×CO$_2$). Further, we examine the change in extreme El Niño events using E-Index > 1.5 s.d. (see Cai et al., 2018) as a threshold. The SST based E-Index identifies 79, 147, and 93 extreme El Niño events in piControl, 4×CO$_2$, and G1, respectively. Thus using E-Index, extreme El Niño events increase by 86% (99% cl) and 17% (missing 95% cl by three events) in 4×CO$_2$ and G1, respectively. Based on the E-index definition, we see a statistically significant increase in the total number of El Niño events in 4×CO$_2$ (107%) and no statistically significant change in G1 (Table S3). Note that Wang et al. (2020) showed that extreme El Niño events having E-Index > 1.5 s.d. can still happen even if the Niño3 rainfall is not greater than 5 mm day$^{-1}$ (cf. Figure 2 in Wang et al., 2020).

We highlight that both in 4×CO$_2$ and solar geoengineered climate, more weak and reversed MSSTG events occur relative to piControl (Fig. S3). More frequent reversals of MSSTG result in a more frequent establishment of strong convection in the eastern equatorial Pacific. According to Cai et al. (2014), more frequent convection over the eastern tropical Pacific increases the sensitivity of rainfall by 25% to positive SST anomalies. Further, in Sect. 3.1.3, we found that WWBs (EWBs) are 13% (7%) stronger (weaker) than in piControl, which also favours a higher frequency of El Niño events in G1. Thus, we conclude that changes in the tropical Pacific mean state; in particular weakening of temperature gradients (MSSTG and ZSSTG), changes in zonal wind stress, and convection over the tropical Pacific (and consistent weakening of the PWC) are the plausible causes of increased frequency of extreme El Niño events under G1.

### 3.2.3 La Niña frequency

During La Niña events, the ZSSTG, the PWC, and atmospheric convection in the western Pacific are stronger than on average. Here, we present plots of Niño4 vs ZSSTG for

piControl, 4×CO$_2$, and G1 (Fig. 8a-c). In 4×CO$_2$, extreme La Nina events are reduced to zero relative to piControl, and a statistically significant (99% cl) decrease occurs in moderate, weak, and total number (sum of extreme, moderate and weak events) of La Niña events. Our findings are inconsistent to those of Cai et al. (2015b) who found nearly doubling of extreme La Nina events under increased GHG forcing. We see a statistically significant (95% cl) increase in extreme La Niña events in G1. The number of extreme La Niña events increases by 32% (61 events) in G1 relative to piControl (46 events). Thus, an extreme La Niña event occurs every ~22 years in piControl and every ~16 years in G1.

The increased number of extreme El Niño events provides a possible mechanism for increased frequency of La Niña events, as they result in more heat discharge events causing cooling, hence providing conducive conditions for increased occurrence of La Niña events (Cai et al., 2015a, 2015b). In addition, the ocean becomes 4% more stratified under G1 relative to piControl (Fig. 15e, Table S7). The increased vertical ocean stratification in the central equatorial Pacific steers cooling in the Niño4 region and, hence, can cause more frequent strong positive ZSSTG anomalies (Fig. S9c and S10b) resulting in an increased number of extreme La Niña events (see also Cai et al., 2015b).

## 3.3 Spatial characteristics of ENSO

In Sect. 3.2, we showed that overall and maximum ENSO event amplitudes generally strengthened under G1, while the amplitude asymmetry between warm and cold events is significantly reduced. In this section, we present composite anomalies, i.e. the average patterns of all El Niño and La Niña events. These composites provide process-based evidence for the strengthening (weakening) of extreme La Niña (El Niño) events in G1. We show that the PWC, SST, and composite rainfall anomalies are strengthened for extreme La Niña events, while they are weakened for extreme El Niño events under G1. For composite analysis, extreme El Niño events are selected with Niño3 rainfall > 5 mm day$^{-1}$ and MSSTG < 0 (Fig. S5) because it gives a more robust estimate as all events show a reversal of MSSTG and more vigorous convection.

### 3.3.1 Weakening of extreme El Niño events in G1

The broad spatial patterns of composite SST (Fig. 9), rainfall (Fig. 10), and PWC (Fig. 11) anomalies for the extreme and total number of El Niño events in G1 are very similar to those of piControl. During extreme El Niño events, in G1, we find reduced SST (Fig. 9e) and rainfall anomalies (Fig. 10e) over the eastern and western equatorial Pacific with a consistent weakening of the eastern and western branch of PWC (Fig. 11e). We also note reduced SST (Fig. 9f) and rainfall (Fig. 10f) anomalies over the western Pacific in agreement with a weakening of western branch of PWC (Fig. 11f) for the total number of El Niño events in G1. Thus, in general, extreme El Niño events tend to be weaker in G1 than in piControl. We conclude that, in our simulations, extreme El Niño events are more frequent but slightly less intense in a solar geoengineered climate than in preindustrial conditions. We further confirm this with a histogram of detrended Niño3 SST anomalies (Fig. S7a). Though more frequent positive Niño3 SST anomalies occur under G1 (between 1 and 3 $^{\circ}$C), the mean Niño3 SST

anomaly is weaker in G1 (1.95 $^{\circ}$C) than in piControl (2.23 $^{\circ}$C) at 99% cl. Thus, the strength of extreme El Niño events is reduced by ~12% in G1 compared to piControl. However, no statistically significant shift in histograms of Niño3 SST anomalies is detected for the total number of El Niño events (Fig. S7b).

### 3.3.2 Strengthening of La Niña events in G1

The broad spatial patterns of composite SST (Fig 12a-d), rainfall (Fig. 13a-d) and PWC (14a-d) anomalies for the extreme and total number of La Niña events are similar under G1 and piControl. During the extreme and total number of La Niña events, the negative SST and rainfall anomalies, and both east and west branch of PWC are strengthened indicating an overall intensification of La Niña events in G1 relative piControl. We note that most of the stronger negative SST anomalies occur over the eastern equatorial Pacific. We confirm strengthening of La Niña events by plotting histograms of detrended Niño3 SST anomalies for the extreme (piControl: -1.45 $^{\circ}$C; G1: -1.68 $^{\circ}$C) and the total number of La Niña events (piControl: -1.03 $^{\circ}$C; G1: -1.22 $^{\circ}$C) based on the Niño4 SST index (Fig. S7c-d). Thus, we conclude that the strength of extreme (total number of) La Niña events is increased by ~16% (~18%) in G1 compared to piControl.

### 4 Mechanisms behind the changes in ENSO variability

### 4.1 Under greenhouse gas forcing

The reduced ENSO amplitude under $4\times CO_2$ is mainly caused by stronger hf and weaker BJ feedback relative to piControl (Fig. 15a-b, and Table S5-6). More rapid warming over the eastern than western equatorial Pacific regions reduces the SST asymmetry between western and eastern Pacific (Fig. 1d), resulting in the weakening of ZSSTG (Fig. 4b) that significantly weakens the zonal winds stress (Fig. 4a) and hence PWC (Fig. 6b, d, see Bayr et al., 2014). The overall reduction of zonal wind stress reduces the BJ feedback, which, in turn, can weaken the ENSO amplitude. Climate models show an inverse relationship between hf feedback and ENSO amplitude (Lloyd et al., 2009, 2011; Kim and Jin, 2011b). The increased hf feedback might be the result of enhanced clouds due to strengthened convection (Fig. 5b, d) and stronger evaporative cooling in response to enhanced SSTs under $4\times CO_2$ (Knutson and Manabe, 1994; Kim and Jin, 2011b). Kim and Jin (2011a, b) found intermodel consensus on the strengthening of hf feedback in CMIP3 models under enhanced GHG warming scenario (Ferret and Collins, 2019). Further, we see increased ocean stratification under $4\times CO_2$ (Fig. 15d and Table S7). A more stratified ocean is associated with an increase in both the El Niño events and amplitude in the eastern Pacific (Wang et al., 2020). It can also modify the balance between feedback processes (Dewitte et al., 2013). Enhanced stratification may also cause negative temperature anomalies in the central to the western Pacific through changes in thermocline tilt (Dewitte et al., 2013). Since the overall ENSO amplitude decreases in our $4xCO_2$ simulation, we, thus, conclude that the ocean stratification mechanisms cannot be the dominant factor here, but that hf and BJ feedbacks must more than cancel out the effect of ocean stratification on ENSO amplitude. Bjerknes feedback is a multi-component process (e.g., Kim and Jin, 2011a), where some components may increase

and some may decrease under the influence of external forcing. For instance, increased upper ocean stratification tends to enhance the Bjerknes feedback, likely through coupling between the wind and thermocline. However, this study represents the Bjerknes feedback solely on the coupling between wind and SST, a caveat of this analysis.

The increased frequency of extreme El Niño events under $4\times CO_2$ is due to change in the mean position of the ITCZ (Fig. S2), causing frequent reversals of MSSTG (Fig. S3), and eastward extension of the western branch of PWC (Fig. 6), which both result in increased rainfall over the eastern Pacific (see Wang et al., 2020). This is due to greater east equatorial than off-equatorial Pacific warming (see Cai et al., 2020), which shifts the mean position of ITCZ towards the equator (Fig. S2). Simultaneously more rapid warming of the eastern than western equatorial Pacific reduces the ZSSTG, and hence zonal wind stress, as also evident from the weakening and shift of the PWC (Fig. 6) and increased instances of negative ZSSTG anomalies (Fig. S9). Ultimately, this leads to more frequent vigorous convection over the Niño3 region (Fig. 5d), and enhanced rainfall (Fig. 2d, S8). Therefore, despite the weakening of the ENSO amplitude under $4\times CO_2$, rapid warming of the eastern equatorial Pacific causes frequent reversals of meridional and zonal SST gradients, resulting in an increased frequency of extreme El Niño events (see also Cai et al., 2014; Wang et al., 2020).

We note that under GHG forcing, HadCM3L does not simulate an increase in the frequency of extreme La Niña events as found by Cai et al. (2015b) using CMIP5 models. However, it does show an increase in the total number of La Niña events (Table S4). In a multimodel ensemble mean, Cai et al. (2015b) found that the western Pacific warms more rapidly than the central Pacific under increased GHG forcing, resulting in strengthening of the zonal SST gradient between these two regions. Strengthening of this zonal SST gradient and increased vertical upper ocean stratification provide conducive conditions for increased frequency of extreme La Niña events (Cai et al., 2015b). One reason why we do not see an increase in the frequency of central Pacific extreme La Niña events might be that HadCM3L does not simulate more rapid warming of the western Pacific compared to the central Pacific as noticed by Cai et al. (2015b) (compare our Fig. 1d with Fig. 3b in Cai et al., 2015b), hence, as stronger zonal SST gradient does not develop, across the equatorial Pacific, as needed for extreme La Niña events to occur (see Fig. S9a, c and S10).

## 4.2 Under solar geoengineering

G1 over cools the upper ocean layers, whereas the GHG-induced warming in the lower ocean layers is not entirely offset, thus increasing ocean stratification (Fig. 15). The increased stratification boosts atmosphere-ocean coupling (see Cai et al., 2018), which favours enhanced westerly wind bursts (Fig. 4a) (e.g., Capotondi et al., 2018) to generate stronger SST anomalies over the eastern Pacific (Wang et al., 2020). The larger cooling of the western Pacific than the eastern Pacific can also enhance westerly wind bursts reinforcing the BJ feedback and hence SST anomalies in the eastern Pacific. We conclude that increased ocean stratification, along with stronger BJ feedback, is the most likely mechanism behind the overall strengthening of ENSO amplitude under G1.

The increased frequency of extreme El Niño events under G1 can be linked to the changes in MSSTG and ZSSTG (see Cai et al., 2014, and Fig. S3, S9). The eastern off-equatorial Pacific cools more than the eastern equatorial regions, providing relatively more conducive conditions for convection to occur through a shift of ITCZ over to the Niño3 region (Fig. 1e). At the same time, the larger cooling of the western equatorial Pacific than of the eastern equatorial Pacific reduces the ZSSTG and convective activity over the western Pacific, which leads to a weakening of the western branch of PWC (Fig. 6e). Hence we see reduced rainfall over the western Pacific and enhanced rainfall from the Niño3 to the central Pacific region (Fig 2e). These mean state changes, strengthening of convection between ~140$^{o}$ W and ~150$^{o}$ E, and more reversals of the MSSTG and ZSSTG (Fig. S3) result in an increased number of extreme El Niño events in G1 than in piControl (Fig. 7).

## 5 Discussion and conclusions

In this paper, we have analysed the impact of abruptly increased GHG forcing ($4\times CO_2$), and solar geoengineering (G1), on the tropical Pacific mean climate and ENSO extremes. Previous solar geoengineering studies did not show any statistically significant change in the PWC (e.g., Guo et al., 2018) or ENSO frequency and amplitude (e.g., Gabriel and Robock 2015). However, those results were strongly limited by the length of the respective simulations, which made changes challenging to detect, given the high tropical Pacific climate variability. This limitation has been overcome here by using long (1000-year) climate model simulations, carried out with HadCM3L. The longer record makes it possible to detect even relatively small changes between the preindustrial and G1 scenarios within the chosen model system.

To conclude, solar geoengineering can compensate many of the GHG-induced changes in the tropical Pacific, but, importantly, not all of them. In particular, controlling the downward shortwave flux cannot correct one of the climate system's most dominant modes of variability, i.e., ENSO, wholly back to preindustrial conditions. The ENSO feedbacks (Bjerkness and heat flux) and more stratified ocean temperatures may induce ENSO to behave differently under G1 than under piControl and $4\times CO_2$. Different meridional distributions of shortwave and longwave forcings (e.g., Nowack et al., 2016) resulting in the surface ocean overcooling, and residual warming of the deep ocean are the plausible reasons for the solar geoengineered climate not reverting entirely to the preindustrial state.

The changes in ENSO feedbacks and more stratified ocean temperatures under both $4\times CO_2$ and G1 can also affect the eastern and central Pacific ENSO variability differently. For instance, more stratified ocean and enhanced BJ feedback in G1 strengthens the eastern Pacific ENSO amplitude but not central Pacific ENSO amplitude (Table 1-2). Similarly, the enhanced hf and weaker BJ feedback in $4\times CO_2$ results in a more substantial reduction in central Pacific ENSO amplitude than eastern Pacific ENSO amplitude (Table 1-2). In the current model system, we expect that changes in tropical Pacific mean state and feedback process, both under $4\times CO_2$ and G1, may impact the occurrence ratio of central Pacific El Niño (La Niña) to eastern Pacific El Niño (La Niña) (e.g., Yeh et al., 2009), which requires further detailed analysis.

Finally, we note that this is a single model study, and more studies are needed to show the robustness and model-dependence of any results discussed here, e.g. using long-term multimodel ensembles from GeoMIP6 (Kravitz et al., 2015), once the data are released. The long-term Stratospheric Aerosol Geoengineering Large Ensemble (GLENS; Tilmes et al., 2018) data can also be explored to investigate ENSO variability under geoengineering.

We summarise our key findings as follows:

1. The warming over the tropical Pacific under increased GHG forcing ($4\times CO_2$) is overcompensated under solar sunshade geoengineering (G1), resulting, by design, in tropical mean overcooling of approximately 0.3 $^o$C. This overcooling is more pronounced in the western tropical Pacific and SPCZ than in the eastern Pacific under the G1 scenario.

2. The reduced SST and rainfall asymmetry between the warm pool and the cold tongue, seen under $4\times CO_2$, is mostly corrected in G1, but regionally important differences remain relative to preindustrial conditions. The tropical Pacific is 5% wetter in $4\times CO_2$, whereas it is 5% drier in G1 relative to piControl. In particular, solar geoengineering results in decreased rainfall over the warm pool, SPCZ, and ITCZ and increased rainfall over the central and eastern equatorial Pacific.

3. The preindustrial median position of ITCZ (154$^o$ W-82$^o$ W; 7.5$^o$ N) changes significantly under $4\times CO_2$ and moves over the equator (154$^o$ W-82$^o$ W; 0$^o$). G1 restores the ITCZ to its preindustrial position (154$^o$ W-82$^o$ W; 7.5$^o$ N).

4. The increased GHG forcing results in 31% reduction in zonal wind stress over the tropical Pacific. G1 fails to compensate this reduction entirely and results in weakening the zonal wind stress by 10% with a 13% (7%) increase (decrease) in WWBs (EWBs), thus providing more conducive conditions for El Niño extremes.

5. Under solar geoengineering, both ZSSTG and MSSTG are reduced by 11% and 9%, respectively. More frequent reversal of MSSTG occurs in G1 relative to piControl.

6. In $4\times CO_2$, the thermocline flattens over the tropical Pacific, and G1 recovers its preindustrial condition.

7. The PWC becomes weaker both under $4\times CO_2$ and G1 scenarios.

8. The increased GHG forcing results in a weakening of ENSO amplitude, whereas solar geoengineering strengthens it relative to preindustrial climate. The maximum amplitude of cold events is enhanced under G1.

9. The reduced ENSO amplitude under $4\times CO_2$ is mainly due to enhanced hf feedback, whereas the increase under G1 is mainly caused by enhanced BJ feedback and ocean stratification.

10. The ENSO amplitude asymmetry between warm and cold events is reduced under G1 relative to piControl.

11. The frequency of extreme El Niño events increases by 61% in G1 relative to piControl. Further, the frequency of the total number of El Niño events also increases by 12%. Thus, an El Niño event occurring every ~3.3-yr under preindustrial conditions occurs every ~2.9-yr under solar geoengineered climate. The reason for the

occurrence of more extreme El Niño events under G1 is more frequent reversals of MSSTG compared to piControl.

12. The frequency of extreme La Niña events increases by 32% under G1 relative to piControl. Thus, an extreme La Niña event occurring every ~22-yr in piControl occurs every ~16-yr in G1.

*Author contribution.* Long Cao developed the model code and performed the simulations. Abdul Malik formulated the research questions, defined the methodology with the help of all co-authors, and performed the scientific analysis. Abdul Malik prepared the manuscript with contributions from all co-authors.

*Competing interests.* The authors declare that they have no conflict of interest.

*Data availability.* Data are available upon request from Long Cao (longcao@zju.edu.cn).

**Acknowledgements**

The Swiss National Science Foundation supported this work under the grant EarlyPostdoc. Mobility (P2BEP2_175255). Peer J. Nowack was funded through an Imperial College Research Fellowship. The authors thank the referees for their comments and suggestions, which have much helped us to improve our manuscript. GPCP Precipitation and NCEP Reanalysis data were provided by the NOAA/OAR/ESRL PSD, Boulder, Colorado, USA, from their Web site at https://www.esrl.noaa.gov/psd/. The ERA5 data was downloaded from https://cds.climate.copernicus.eu/cdsapp#!/home.

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

# Figures and Figure Captions

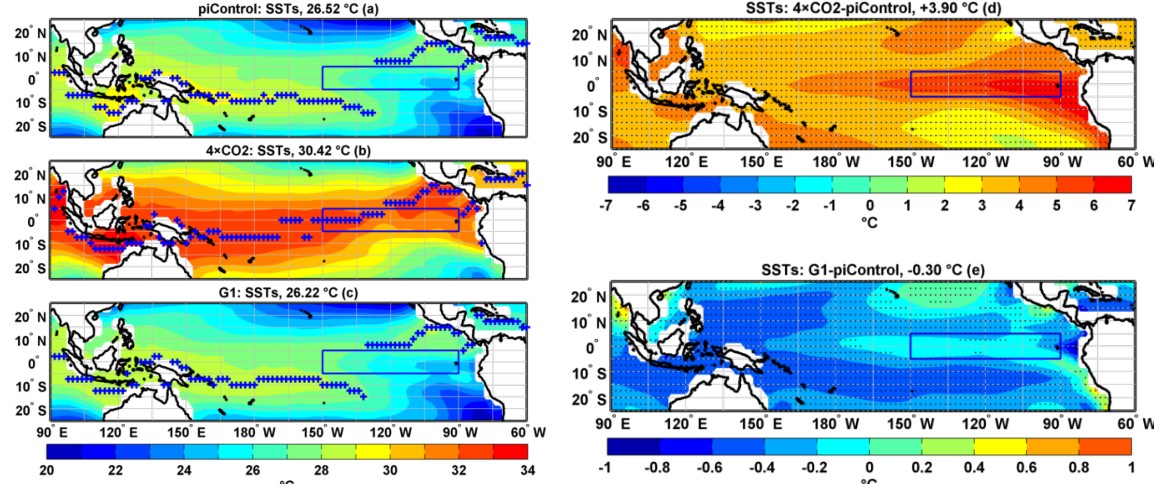

**Figure 1.** Tropical Pacific SST mean DJF climatology (a) piControl (b) 4×CO$_2$ (c) G1 (d) difference 4×CO$_2$-piControl and (e) difference G1-piControl. The blue plus sign in a-c indicates latitudes with maximum SSTs. Stipples indicate grid points where the difference is statistically significant at 99% cl using a non-parametric Wilcoxon rank-sum test. The box in the eastern Pacific identifies the Niño3 region. The numbers in a-c represent a mean temperature in the corresponding simulation, and numbers in d-e represent an area-averaged difference of piControl with 4×CO$_2$ and G1, respectively, in the tropical Pacific region (25$^o$ N-25$^o$ S; 90$^o$ E-60$^o$ W).

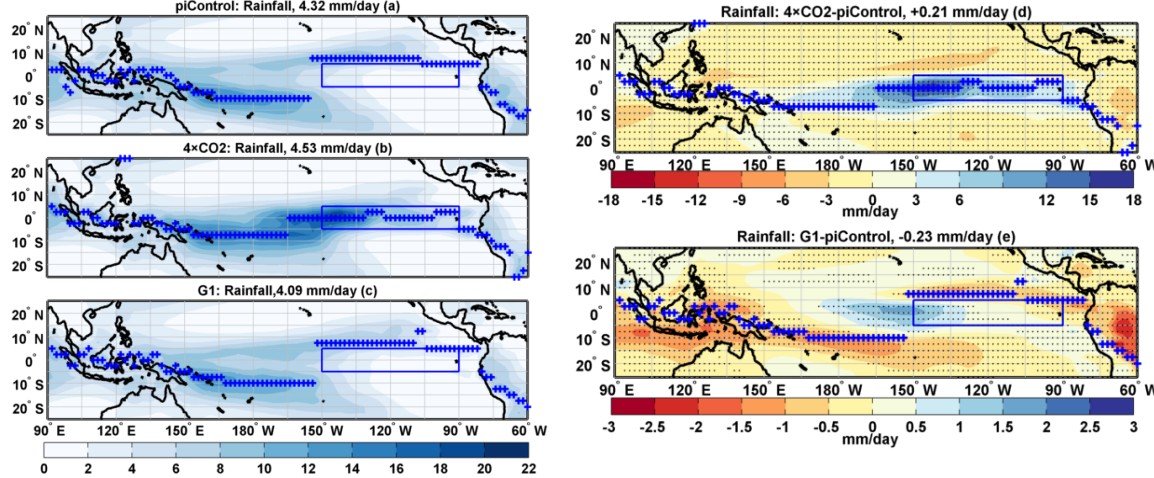

**Figure 2.** Tropical Pacific rainfall mean DJF climatology (a) piControl (b) $4\times CO_2$ (c) G1 (d) difference: $4\times CO_2$-piControl; the blue plus signs indicate the position of ITCZ under $4\times CO_2$ and (e) difference: G1-piControl; the blue plus signs indicate the position of ITCZ under G1. In a-c, the blue plus signs indicate the position of ITCZ for the corresponding experiment. Stipples indicate grid points where the difference is statistically significant at 99% cl using a non-parametric Wilcoxon rank-sum test. The numbers in a-c represent mean rainfall in the corresponding simulation, and numbers in d-e represent an area-averaged difference of piControl with $4\times CO_2$ and G1, respectively, in the tropical Pacific region ($25^{\circ}$ N-$25^{\circ}$ S; $90^{\circ}$ E-$60^{\circ}$ W).

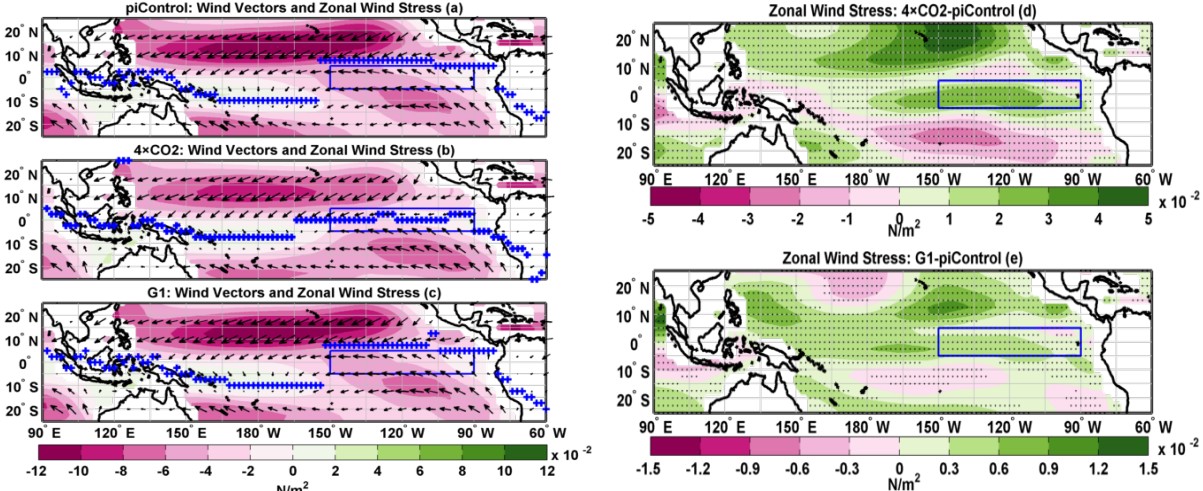

**Figure 3.** Tropical Pacific zonal wind stress mean DJF climatology (a) piControl (b) $4\times CO_2$ (c) G1 (d) difference: $4\times CO_2$-piControl and (e) difference: G1-piControl. Black arrows indicate the direction of 10 m wind. The blue plus sign in a-c indicates latitudes with maximum rainfall. Stipples indicate grid points where the difference is statistically significant at 99% cl using a non-parametric Wilcoxon rank-sum test.

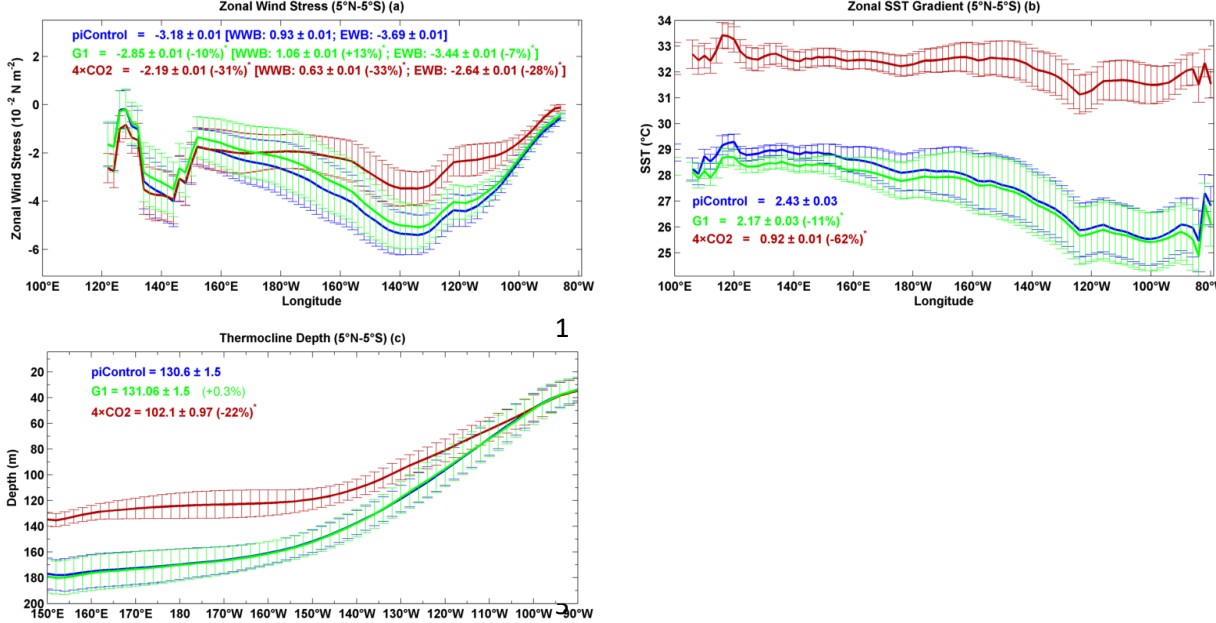

Figure 4. DJF mean climatology of (a) zonal wind stress, (b) zonal SST gradient, and (c) thermocline depth. Error bars indicate ±1 s.d. calculated over the simulated period. Numbers with an asterisk indicate that the percentage change is statistically significant at 99% cl.

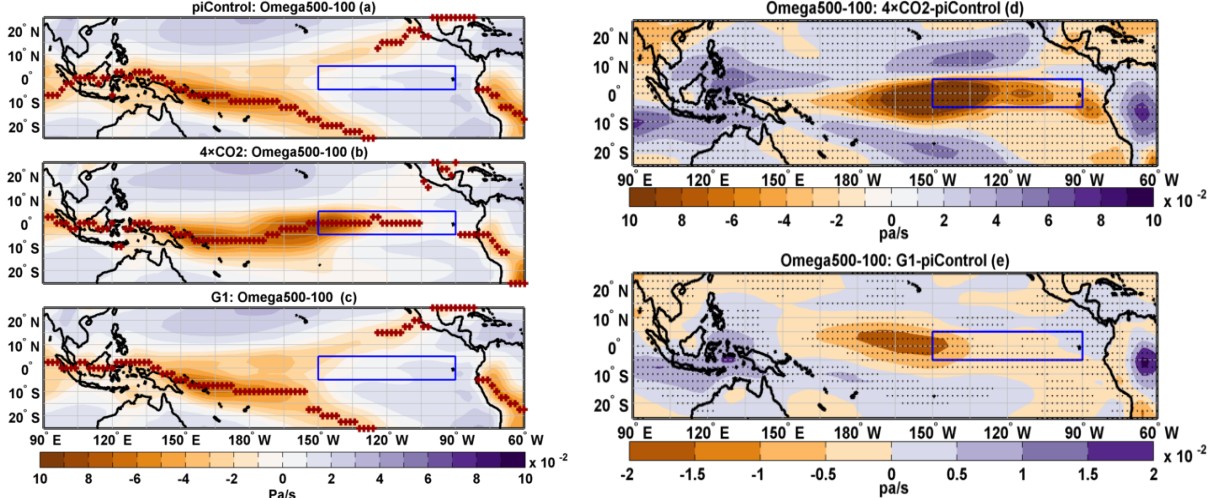

Figure 5. Tropical Pacific mean DJF climatology of vertical velocity averaged between 500- and 100-hPa (Omega500-100) (a) piControl (b) 4×CO$_2$ (c) G1 (d) difference: 4×CO$_2$-piControl and (e) difference: G1-piControl. In a-c, the brown plus sign indicates latitudes where maximum upwelling occurs. Stipples indicate grid points where the difference is statistically significant at 99% cl using a non-parametric Wilcoxon rank-sum test.

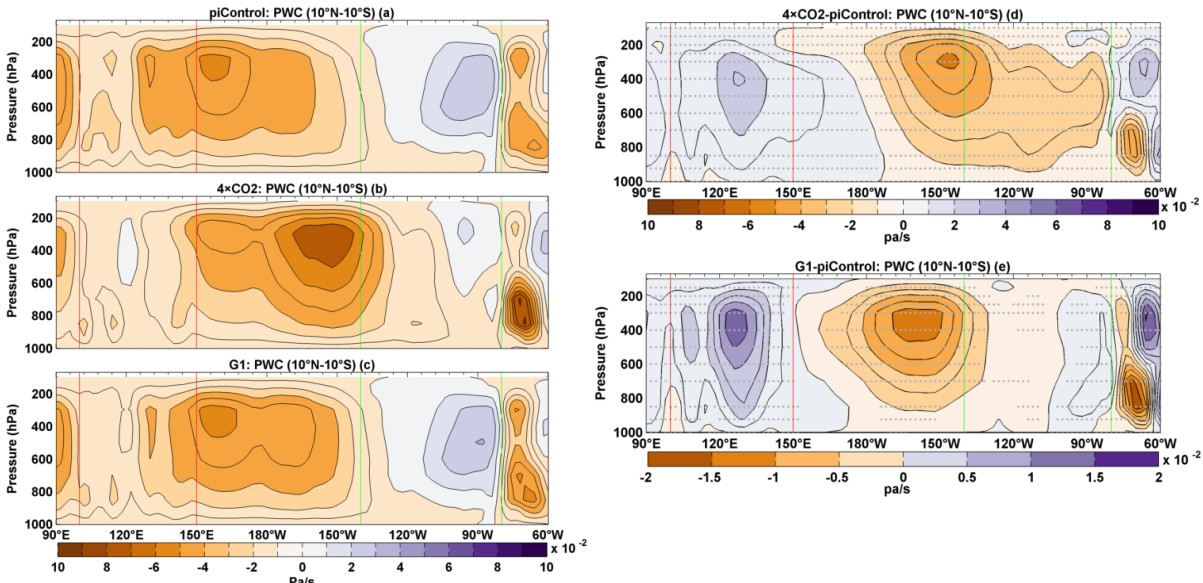

**Figure 6.** Mean DJF climatology of tropical Pacific Walker Circulation averaged over 90$^o$ E-60$^o$ W and 10$^o$ N-10$^o$ S (a) piControl (b) 4×CO$_2$ (c) G1 (d) difference: 4×CO$_2$-piControl and (e) difference: G1-piControl. Green (red) vertical lines show the longitudinal spread of the eastern (western) Pacific. Stipples indicate grid points where the difference is statistically significant at 99% cl using a non-parametric Wilcoxon rank-sum test.

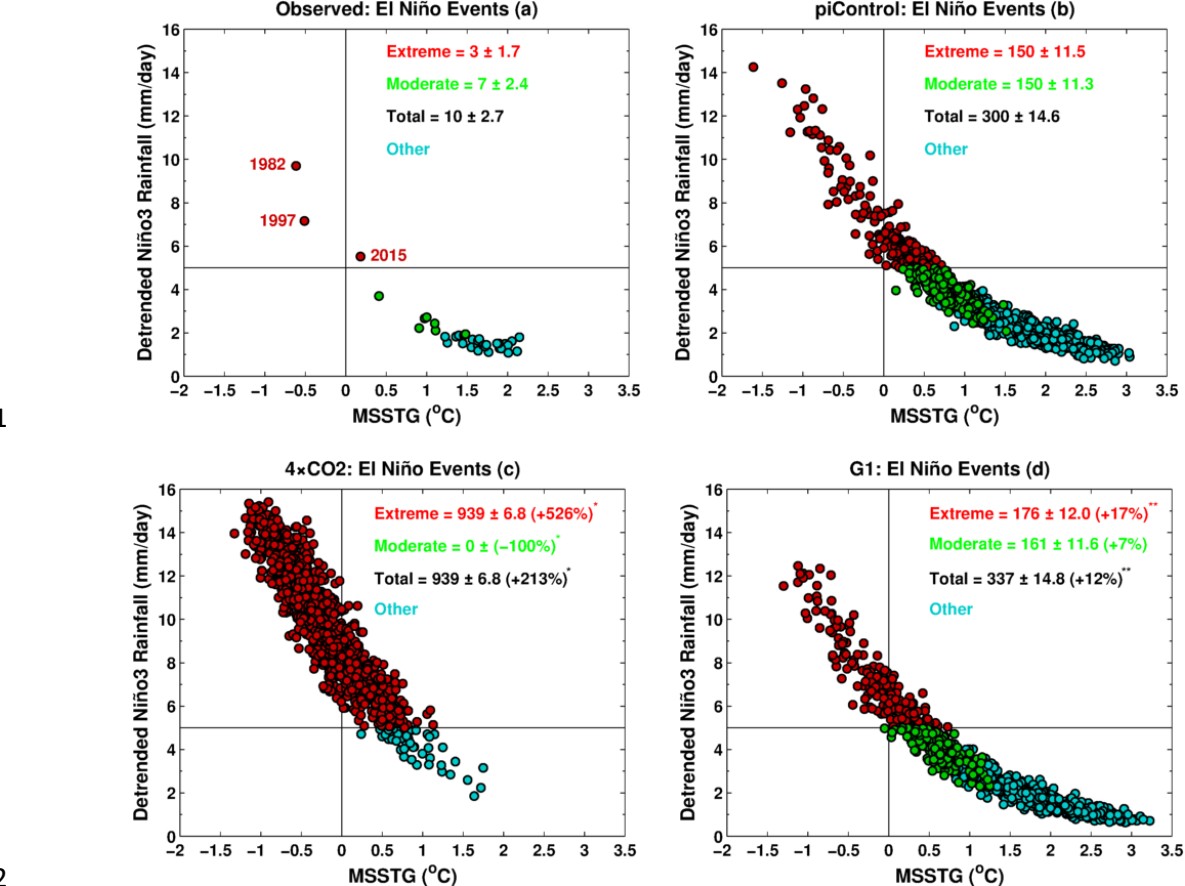

**Figure 7.** Relationship between MSSTG and Niño3 rainfall for (a) observations (b) piControl
(c) 4×CO$_2$, and (d) G1. A solid black horizontal line indicates a threshold value of 5 mm day$^-$
$^1$. See text for the definition of extreme, moderate, and total El Niño events. A single (double)
asterisk indicates that the change in frequency, relative to piControl, is statistically significant
at 99% (95%) cl. Numbers with a ± symbol indicate s.d. calculated with 10,000 bootstrap
realisations. Following Cai et al. (2014), a non-ENSO related trend has been removed from
the rainfall time series.

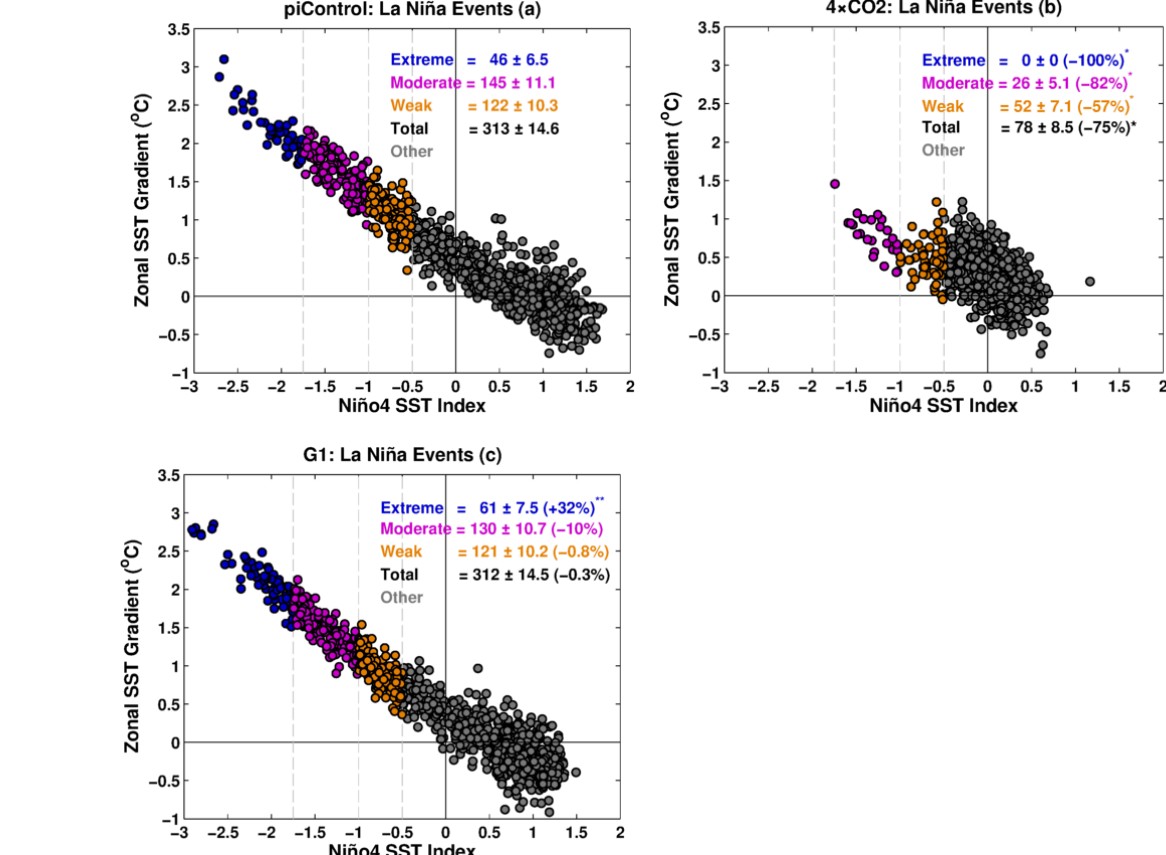

**Figure 8.** Relationship between ZSSTG and Niño4 SST index for (a) piControl (b) 4×CO$_2$ and (c) G1. Dashed grey vertical lines indicate threshold values of -1.75, -1, and -0.5 s.d. See text for the definition of extreme, moderate, weak, and total La Niña events. A single (double) asterisk indicates that the change in frequency is statistically significant at 99% (95%) cl. Numbers with a ± symbol indicate s.d. calculated with 10,000 bootstrap realisations.

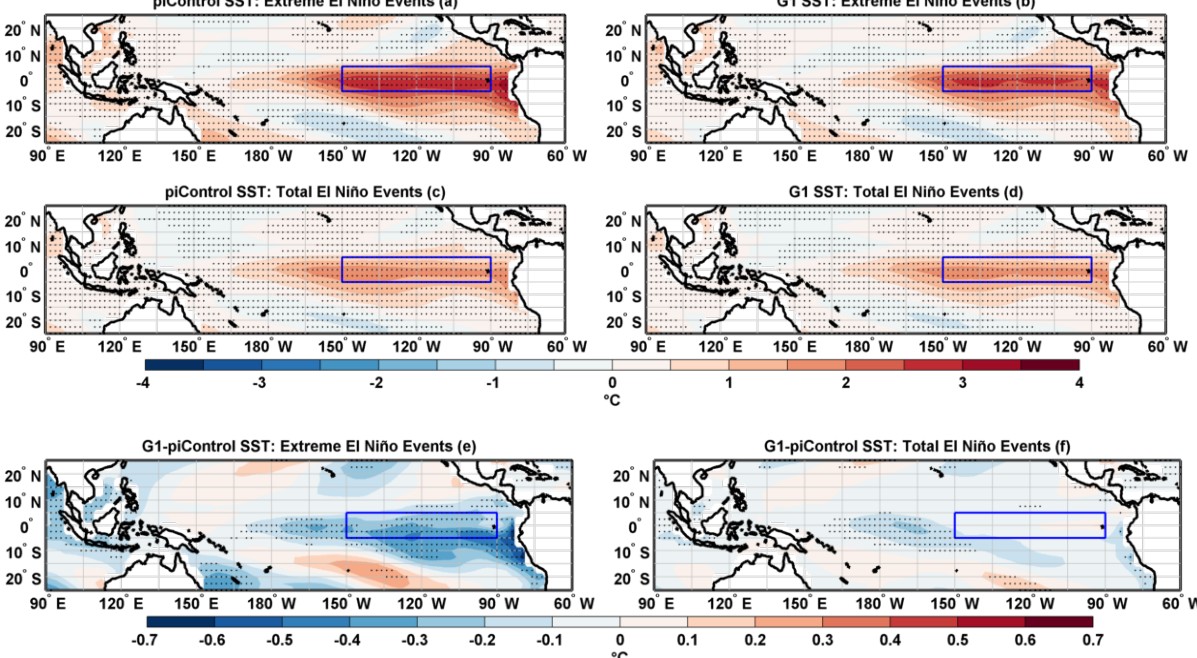

**Figure 9.** Composites of SST anomalies for extreme El Niño events in (a) piControl and (b) G1. Composites of SST anomalies for the total number of El Niño events in (c) piControl and (d) G1. Composite differences (G1-piControl) of SST anomalies for (e) extreme El Niño events and (f) total number of El Niño events. Stipples indicate grid points with statistical significance at 99% cl using a non-parametric Wilcoxon rank-sum test. The blue box in the eastern Pacific identifies the Niño3 region.

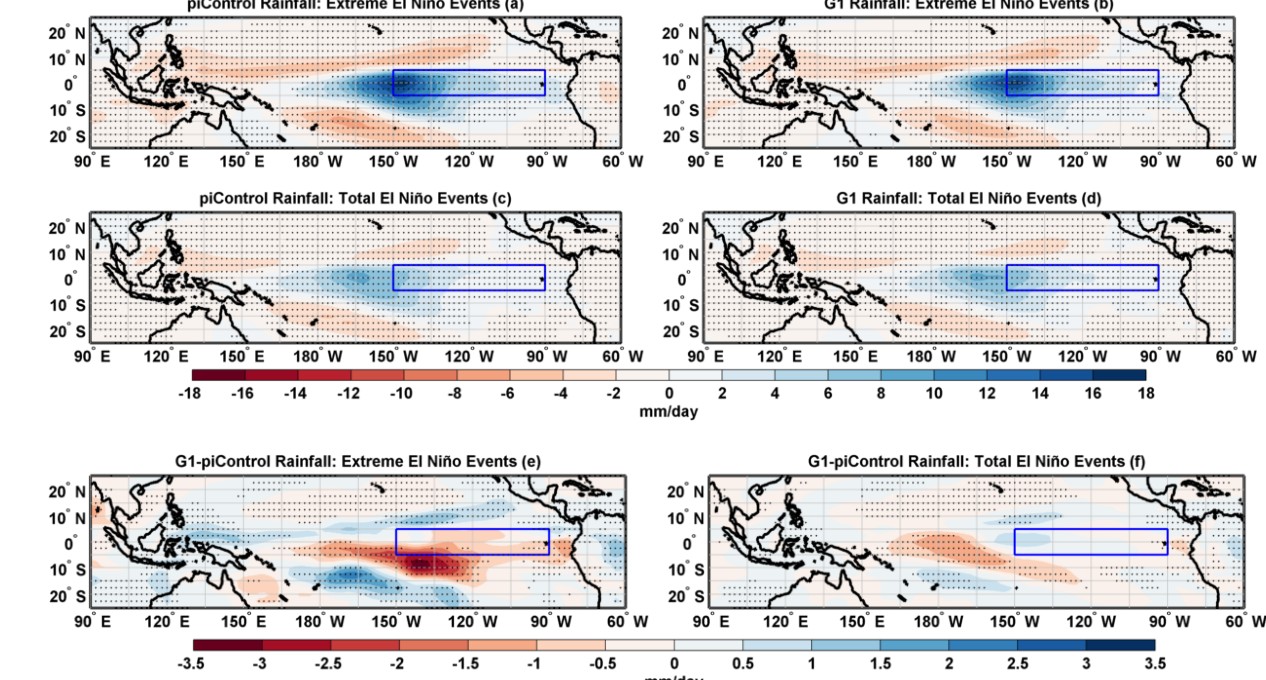

**Figure 10.** Composites of rainfall anomalies for extreme El Niño events in (a) piControl and (b) G1. Composites of rainfall anomalies for the total number of El Niño events in (c) piControl and (d) G1. Composite differences (G1-piControl) of rainfall anomalies for (e) extreme El Niño events and (f) total number of El Niño events. Stipples in a-d and f (e) indicate grid points with statistical significance at 99 (95) % cl using a non-parametric Wilcoxon rank-sum test. The blue box in the eastern Pacific identifies the Niño3 region.

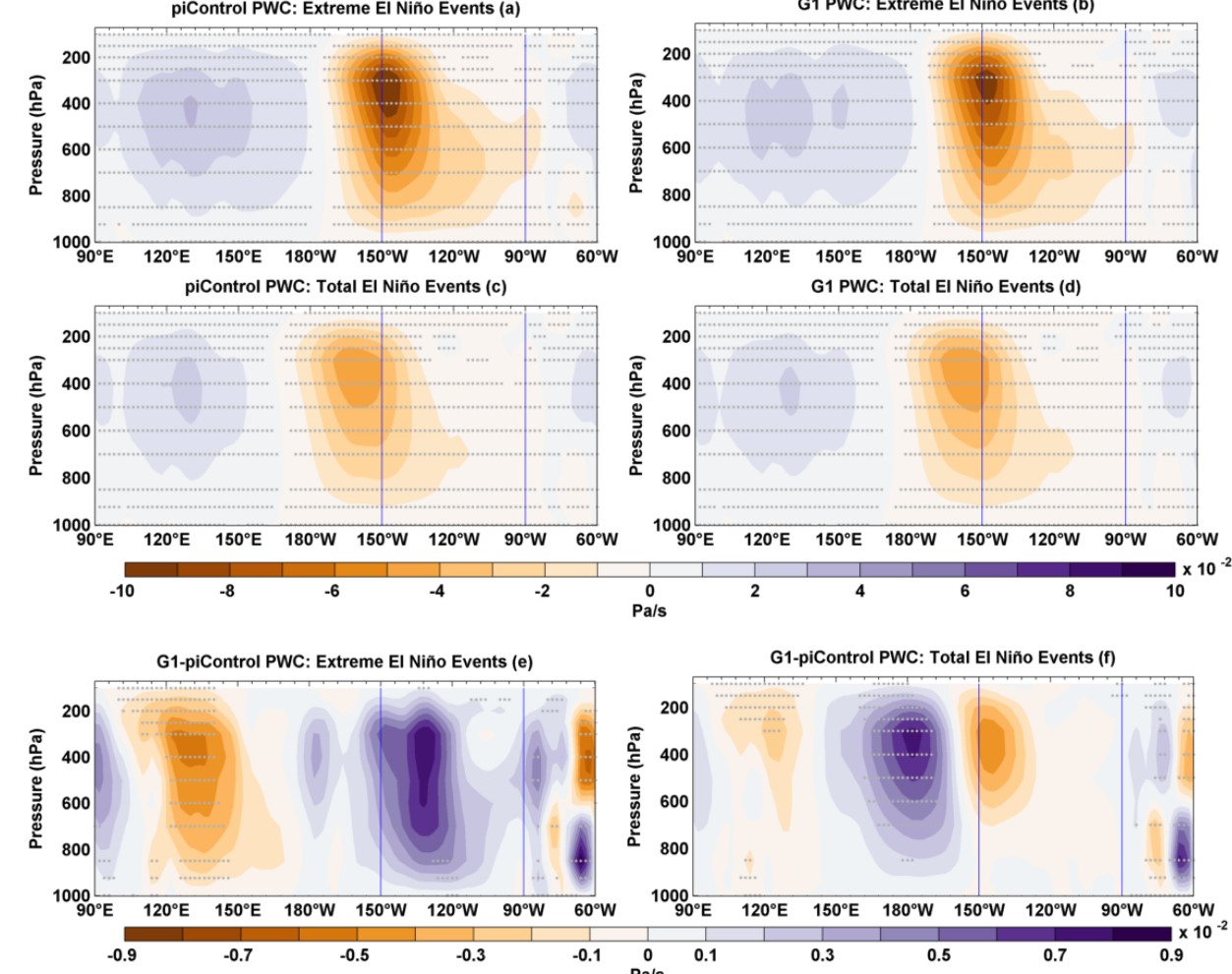

**Figure 11.** Composites of PWC anomalies for extreme El Niño events in (a) piControl and (b) G1. Composites of PWC anomalies for the total number of El Niño events in (c) piControl and (d) G1. Composite differences (G1-piControl) of PWC for (e) extreme El Niño events and (f) total number of El Niño events. Stipples indicate grid points with statistical significance at 99% cl using a non-parametric Wilcoxon rank-sum test. The blue vertical lines indicate the Niño3 region.

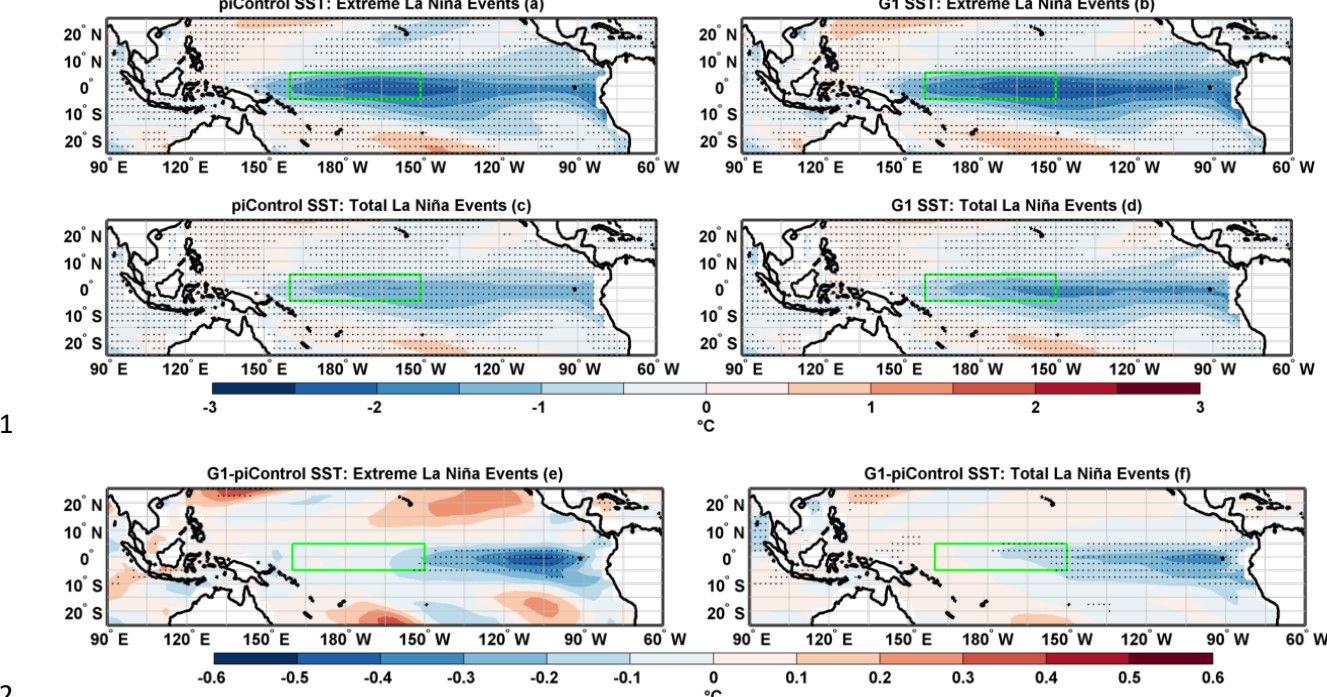

**Figure 12.** Composites of SST anomalies for extreme La Niña events in (a) piControl and (b) G1. Composites of SST for the total number of La Niña events in (c) piControl and (d) G1. Composite differences (G1-piControl) of SST for (e) extreme La Niña events and (f) the total number of La Niña events. Stipples indicate grid points with statistical significance at 99% cl using a non-parametric Wilcoxon rank-sum test. The green box indicates the Niño4 region.

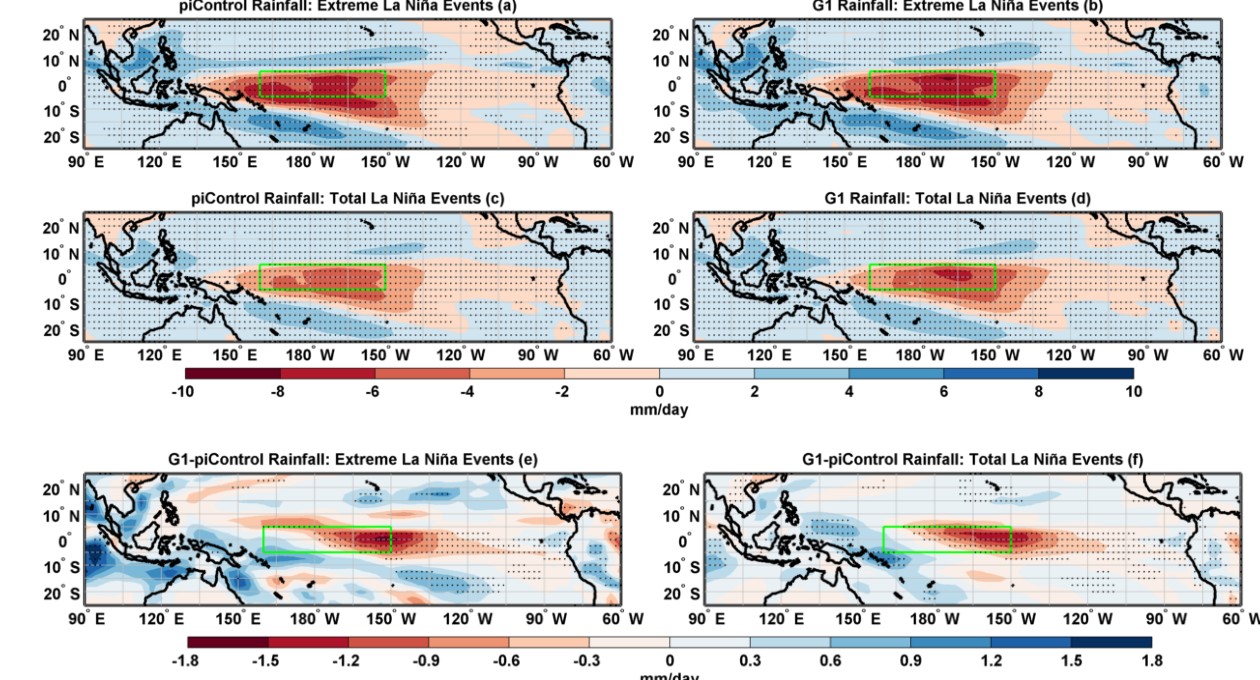

**Figure 13.** Composites of rainfall anomalies for extreme La Niña events in (a) piControl and (b) G1. Composites of rainfall anomalies for the total number of La Niña events in (c) piControl and (d) G1. Composite differences (G1-piControl) of rainfall for (e) extreme La Niña events and (f) the total number of La Niña events. Stipples indicate grid points with statistical significance at 99% cl using a non-parametric Wilcoxon rank-sum test. The green box indicates the Niño4 region.

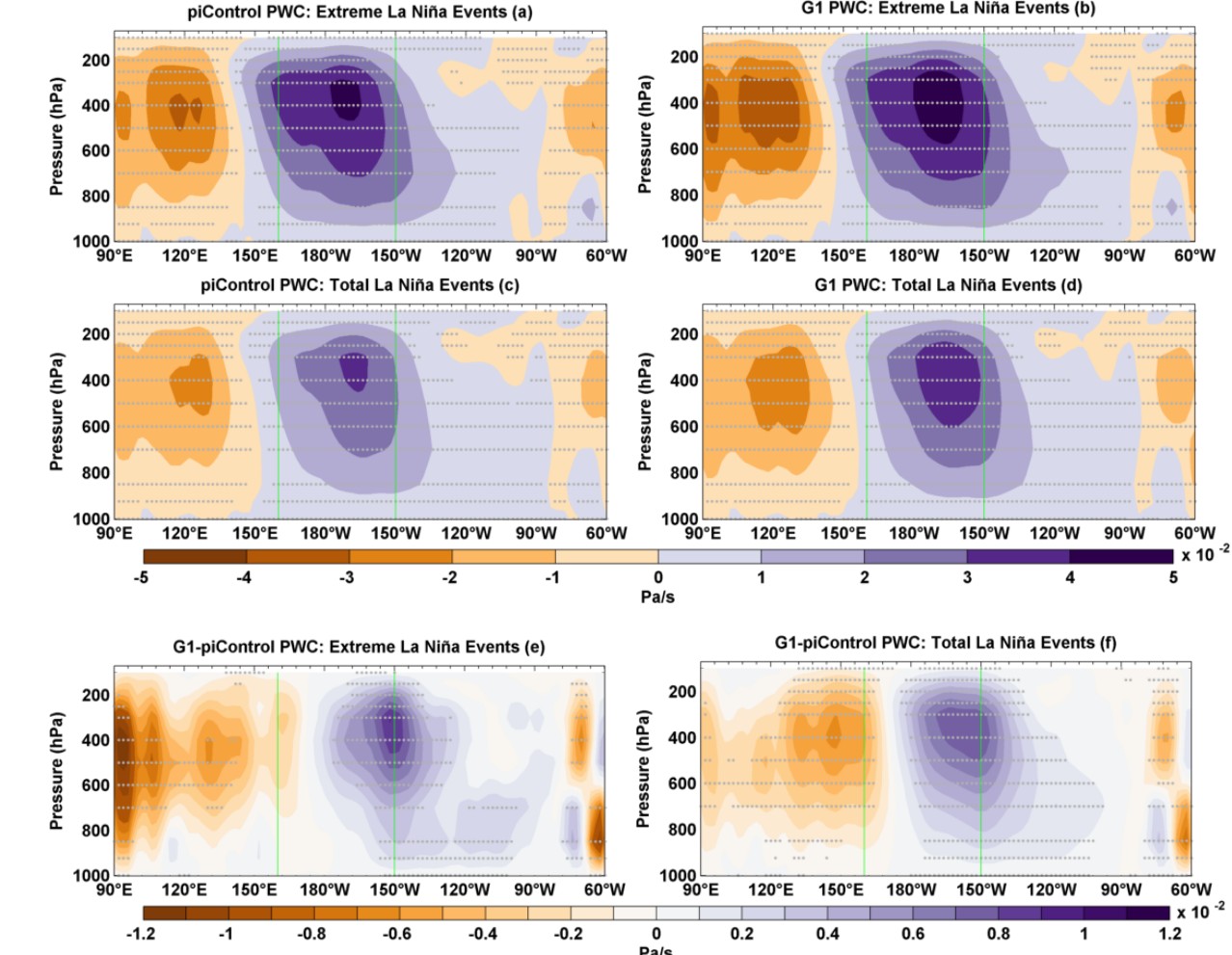

**Figure 14.** Composites of PWC anomalies for extreme La Niña events in (a) piControl and (b) G1. Composites of PWC for the total number of La Niña events in (c) piControl and (d) G1. Composite differences (G1-piControl) of PWC anomalies for (e) extreme La Niña events and (f) the total number of La Niña events. Stipples indicate grid points with statistical significance at 99% cl using a non-parametric Wilcoxon rank-sum test. The green vertical lines indicate the Niño4 region.

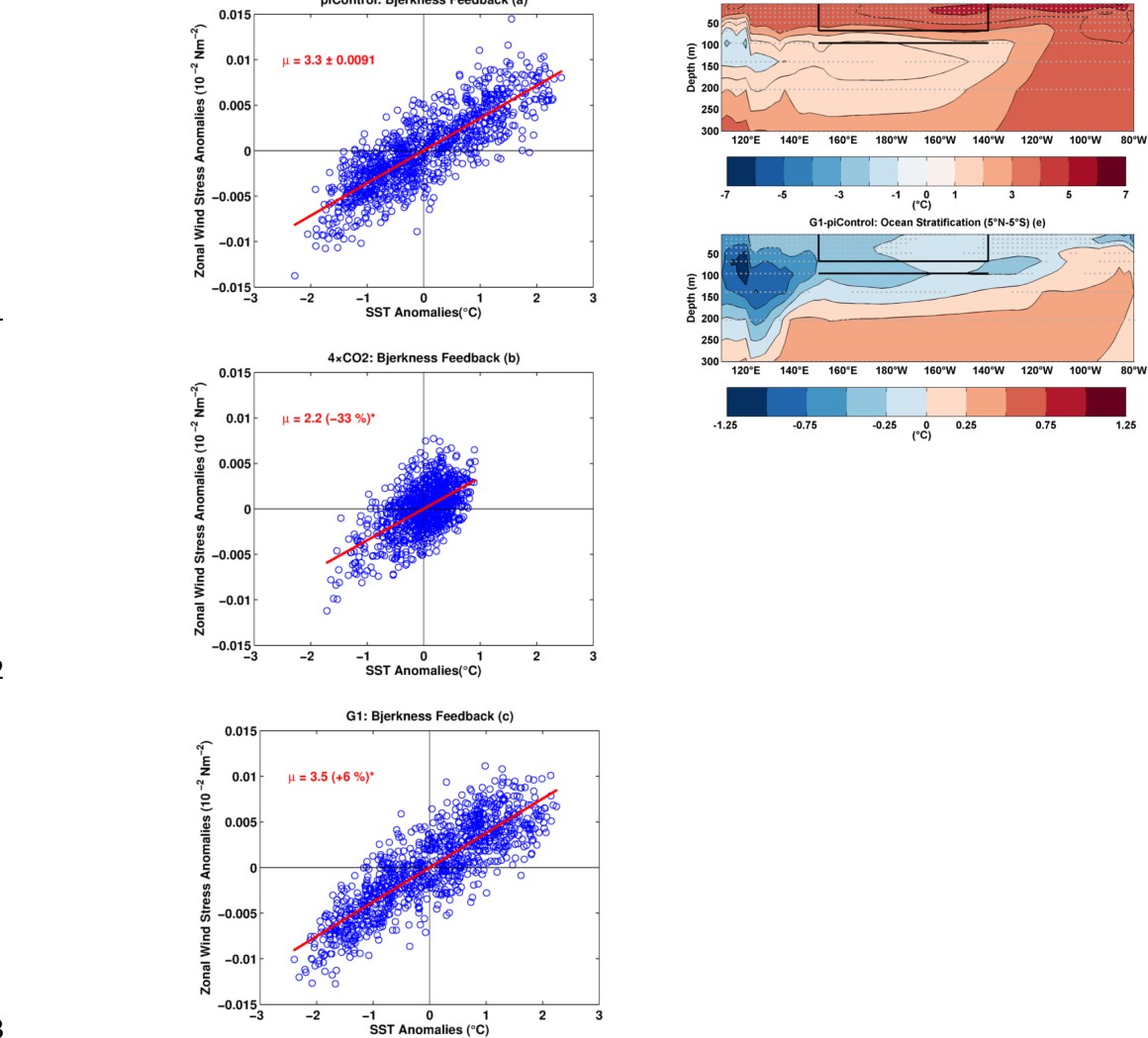

**Figure 15.** BJ feedback ($\mu$; $10^{-2}$ $Nm^{-2}/^oC$) for (a) piControl (b) $4\times CO_2$, and (c) G1. The value with ± sign indicates s.d. of $\mu$ after 10,000 bootstrap realisations. An asterisk indicates statistical significance at 99% cl. Mean change in ocean temperature, (d) $4\times CO_2$-piControl, and (e) G1-piControl. The black box shows the area averaging region for upper ocean temperature, and the black line shows the lower layer used for calculation of stratification as a difference of upper and lower layer. Stipples indicate grid points with statistical significance at 99% cl using a non-parametric Wilcoxon rank-sum test.

1    **Tables and Table Captions**

2    **Table 1.** Eastern Pacific ENSO amplitude

| Experiment | Amplitude ($^o$C) | Difference w.r.t. piControl ($^o$C) | Std. Dev. 10,000 Realizations ($^o$C) | ~ Change w.r.t. piControl (%) |
|---|---|---|---|---|
| **piControl** | 1.04 [1.03] | | 0.0213 [0.03] | |
| **4×CO₂** | 0.55 [0.85] | -0.49 [-0.18] | | -47* [-17*] |
| **G1** | 1.13 [1.13] | 0.09 [0.1] | | +9* [+10**] |

3    Key: Niño3 [E-Index]; *99% cl; **95% cl

5    **Table 2.** Central Pacific ENSO amplitude

| Experiment | Amplitude ($^o$C) | Difference w.r.t. piControl ($^o$C) | Std. Dev. 10,000 Realizations ($^o$C) | ~ Change w.r.t. piControl (%) |
|---|---|---|---|---|
| **piControl** | (0.78) [0.85] | | (0.0132) [0.0167] | |
| **4×CO₂** | (0.28) [0.53] | (-0.50) [-0.32] | | (-64*) [-38*] |
| **G1** | (0.79) [0.83] | (0.01) [0.03] | | (+1) [-3] |

6    Key: (Niño4) [C-Index]; *99% cl; **95% cl

8    **Table 3.** Maximum amplitude of warm events

| Experiment | Amplitude ($^o$C) | Difference w.r.t. piControl ($^o$C) | Std. Dev. 10,000 Realizations ($^o$C) | ~ Change w.r.t. piControl (%) |
|---|---|---|---|---|
| **piControl** | 2.97 [4.59] | | 0.0687 [0.2342] | |
| **4×CO₂** | 1.29 [3.65] | -1.68 [-0.94] | | -57* [-21*] |
| **G1** | 2.85 [4.33] | -0.12 [-0.26] | | -4 [-6] |

9    Key: Niño3 [E-Index]; *99% cl; **95% cl

11    **Table 4**. Maximum amplitude of cold events

| Experiment | Amplitude ($^o$C) | Difference w.r.t. piControl ($^o$C) | Std. Dev. 10,000 Realizations ($^o$C) | ~ Change w.r.t. piControl (%) |
|---|---|---|---|---|
| **piControl** | (-2.13) [-2.47] | | (0.0459) [0.1452] | |
| **4×CO₂** | (-1.37) [-2.17] | (-0.76) [-0.30] | | (-36*) [-12*] |
| **G1** | (-2.55) [-2.90] | (0.42) [0.43] | | (+20*) [+17*] |

12    Key: (Niño4) [C-Index]; *99% cl; **95% cl

14    **Table 5.** Niño3 SST skewness

| Experiment | Skewness | Difference w.r.t. piControl | Std. Dev. 10,000 Realizations | ~ Change w.r.t. piControl (%) |
|---|---|---|---|---|
| **piControl** | 0.52* | | 0.0542 | |
| **4×CO₂** | -0.47* | -0.99 | | -190* |
| **G1** | 0.18* | -0.34 | | -65* |

15    Key: *99% cl; **95% cl