# Peer review of "Tropical Pacific Climate Variability under Solar Geoengineering: Impacts on ENSO 1"

_Atmospheric Chemistry and Physics, 2018_

## Referee Comment (RC1) · Anonymous Referee #1 · 15 Feb 2019

This study investigates what would happen to ENSO variability if geoengineering is applied. To do this, it employs millennial integration of a coupled climate model under pre-industrial, 4xCO2, and geoengineering (G1) scenarios. The very long integration is key to detecting statistically significant changes in ENSO, which weren't found by previous studies using short data of about 50 years. This study extends the analysis from previous G1-based studies to look into extreme ENSO which has been projected to increase in frequency under business as usual emission scenarios by Cai et al. The topic is an important one given the global impact of ENSO, and the paper is informative. However it would benefit from a careful revision as there are various instances that are not clear. Comments are provided below for the authors' considerations.

Main criticism is that as it stands this paper sounds more like throwing out results,

[Figure]

from a single model, without giving it further perspectives on the mechanisms. For instance, it is not clear exactly why the modeled ENSO changed from 4xCO2 to G1 in this model? Is it because of the air-sea heat fluxes act more less as a damping in the eastern equatorial Pacific associated with the mean state change in G1? More interestingly, why G1 does not recover many of the climatic states of piControl? Initial thought would be the ocean state never fully recovers. But as stated in the paper the change in thermocline depth is not statistically different between G1 and piControl. I don't think I came across a plot of subsurface temperature, e.g., depth-longitude differences between 4xCO2 and G1 vs piControl. Perhaps while the thermocline depth statistics do not change, there are still changes in the subsurface ocean temperatures in certain areas.

Nonetheless this leads to the question: How large are the differences in mean state and ENSO statistics between G1 and piControl state in comparison to the internal variability in piControl? For example P9, L20-21: the reduction in MSSTG is 9% in G1, is this this substantial compared internal variability in piControl and to that seen during an El Nino?

In many of the plots showing differences between experiments and piControl, the confidence level was set to 90%. Given the long time series of the model output, it should be increased to 95% or even 99%. This would perhaps show more regions in G1 where the differences are not significantly different from piControl.

The conclusion section could provide the reader with a little perspective on whether it is worth it to do the geoengineering solution in the context of projected increase in extreme ENSO activity. A relevant paper to help the discussion: Trenberth KE, Dai A (2007). Geophys Res Lett 34:L15702. doi: 10.1029/2007GL030524

Other specific major comments:

P11, L36: Picking a result on one model sounds rather odd as we know that the change in ENSO amplitude varies widely across models (e.g., Collins et al. 2010). In a recent

study by Cai et al. (2018, Nature, https://www.nature.com/articles/s41586-018-0776-9), however, there seems to be a stronger inter-model agreement on the increase in ENSO amplitude in models that are able to simulate ENSO flavors (see their Extended Data Fig. 8b), as implied in the PC1-PC2 space. So does the HadCM3L model capture the nonlinear relationship between PC1 and PC2 as observed? Here PC1 and PC2 refer to the first and second eigenmodes of tropical Pacific SST (see their Fig. 1). Also, it is relevant to discuss the results of Cai et al. (2018) in 1st paragraph of Page 3.

P7, L10: make clear the results are in *qualitative* agreement with previous studies. Not all of the cited studies are based on 4xCO2.

P.7, L13: some studies argue against the use of "El Nino-like" term in describing the mean-state change under greenhouse forcing (e.g., Collins et al. 2010; see also Xie et al. 2010 https://journals.ametsoc.org/doi/10.1175/2009JCLI3329.1). Cautionary is needed to avoid confusions. A relevant reference on the mean state change: diNezio et al https://journals.ametsoc.org/doi/full/10.1175/2009JCLI2982.1

Fig. 2d, e: title of the figure states +0.21 mm/day, -0.23 mm/day. Please explain in the caption that those numbers correspond to the area average difference between experiment and piControl in the tropical Pacific (state domain).

P9, L22-24: This sentence needs a rework. Avoid the word "observe" on model analysis (models are not observations). I think Wang et al. (2017) was referring to zonal temperature gradient between the maritime continent and central Pacific, not eastern Pacific. The difference is not significant in RCP2.6, but should be significant in RCP8.5 (Cai et al. 2015, Nature Climate Change on extreme La Nina).

Fig. 7: Please indicate clearly in the caption that the timeseries have been detrended with non ENSO related trend removed following Cai et al. (2017). Otherwise it would create confusion as other studies show that the 2015/16 Nino3 rainfall is close to the 5 mm/day threshold and is thus classified as an extreme El Nino (Santoso et al. 2017). In panel c, d, it must be rainfall anomalies that are shown because there are negative

rainfall values, so wouldn't the 4 or 3 mm/day threshold be applied here? Panel a and b also have negative rainfall values. Please double check.

P12, L28-31: under 4xCO2 the rainfall skewness is dramatically reduced. Does that mean there are less extreme El Nino based on the rainfall definition? If so, this does not seem consistent with the PPE results of Cai et al. (2014) using the same model.

P13, L28-39: The characterization of extreme La Nina is based on Nino4 (Cai et al. 2015), so it is not clear how Nino3 and Nino3.4 indices are used here to infer changes in extreme La Nina.

Figure presentation

Fig. 1e, some areas look white (e.g., eastern equatorial Pacific which is supposed to be approx. -0.2C p7, L9) while the colorbar does not have white on it.

Figure 10: the color limit does not seem correct, which shows much larger values in e, f G1-piControl than the composite anomalies themselves in panels a-d.

The colorbar of Fig. 2, right panel especially is not ideal. It is hard to immediately see which are positive or negative without referring to the colorbar.

Might be best to have the same color scale for comparing the results of 4xCO2 – piControl vs G1 – piControl. This is to convey the message the difference is much smaller for G1 – piControl than for 4xCO2.

Minor points

Page 4, L34: that sentence is due to Cai et al. (2014).

P4, L35: delete "the northern part of" – the ITCZ is located north of equator, and that rainfall band moves equatorward during strong El Nino events.

P5, L23: "ggradients"

P6, L2: extreme El Ninos are not resulting in just "anomalous rainfall" but unusually

large rainfall in the eastern equatorial Pacific.

P6, L 35: "depicts this SSTasymmetry between the western and eastern equatorial Pacific well (Fig. 1a)." – not clear since the observed counterpart is not presented.

P8, L10: "problem" – not clear, in what way it is a problem?

P9, L19: repetitive: El Nino being stronger than La Nina already implies asymmetric amplitude.

P9, L29: the shoaling of thermocline is also due to increased stratification associated with surface intensified warming in response to greenhouse forcing.

P9, L32-36: why not use the maximum of vertical temperature gradient as a proxy of thermocline depth for all scenarios?

P14, L6-7: for extreme El Nino events, are the PWC, SST, and rainfall anomalies strengthened as well?

P14, L23-25: this must be referring to the difference between G1 and piControl. Please make that clear.

---

## Short Comment (SC1) · 27 Mar 2019

I analyzed ENSO in HadCM3 and HadCM3L during my PhD (in work that was never published) and found that the ENSO SST timeseries in HadCM3 were fairly similar to observations whereas the HadCM3L series looked nothing like the observed series. I was therefore surprised to see that this paper used HadCM3L and described the ENSO performance as good. Does HadCM3L actually have a good representation of ENSO?

"HadCM3L is capable of reproducing present-day ENSO periodicity, teleconnection patterns, and amplitude (Collins et al., 2001)." The paper the authors cite, actually reports results for HadCM3 and not HadCM3L, i.e. a model with double the ocean resolution than the one they use here:

https://link.springer.com/article/10.1007%2Fs003820000094

Has ENSO been analyzed in HadCM3L before? Has its performance been validated? Seeing a comparison between the simulated and observed ENSO timeseries would provide a basic test of ENSO performance (see here: https://www.esrl.noaa.gov/psd/enso/dashboard.html), though I'm sure there are other relevant tests that could gauge HadCM3L's simulation of ENSO.

---

## Referee Comment (RC2) · Anonymous Referee #2 · 27 Apr 2019

Review Comments: This manuscript tries to investigate the impacts of solar geoengineering strategies on our climate, especially the interannual variability in the tropical Pacific, referring to ENSO phenomenon. In detail, the mean state of tropical Pacific, ENSO intensity and ENSO frequency are analyzed by comparing the results from piControl, 4×CO2 and G1 simulations. This work emphasizes that the results from 1000 years long simulation which is much longer than previous used, can be more significant to detect the ENSO changes. Highly based on the definition in Cai et al. (2014), the extreme ENSO increases in G1 compared to piControl. Obviously, the analyses are very detailed in describing the results, including the changes of sea surface temperature, precipitation, zonal wind stress, ZSSTG, MSSTG, thermocline, PWC, ENSO amplitude, ENSO frequency, extreme ENSO. However, major revision is needed considering

unclear physical mechanisms involved in this paper.

Major comments: 1. To study the ENSO changes under solar geoengineering, the results are all based on one single model HadCM3L. In Cai et al. (2014) and Collins et al. (2001), the model they used is HadCM3. I admit that HadCM3L and HadCM3 are identical in most aspects, but there are still differences between these two simulations. The differences should be mentioned in this study because the HadCM3L may not be skillful in reproducing ENSO variabilities, and thus the sentence in P4 L32-33 may not be completely correct. I suggest that the ENSO simulated in HadCM3L should be addressed first, regarding its magnitude and pattern. For instance, the EOF analyses can be carried out on the piControl simulations. It will help us to have a general idea of how capable the HadCM3L is in simulating the ENSO and its diversity, and what's the biases compared with observations. As pointed out in Cai et al. (2018), the magnitude and location of ENSO events are inconsistent among models. The averaged SSTA in a fixed box to measure the intensity of ENSO can be tricky. A look at the ENSO pattern in HadCM3L can also facilitate a better ENSO extreme definition, i.e. the Nino indices may not be best to define ENSO intensity. At least, a glimpse of the Figure 8 reveals that ENSO simulation is not good enough, especially the shape, maximum location and horseshoe-shaped cold SSTA in the western Pacific during El Nino events. 2. The change of extreme ENSO under solar geoengineering is a major concern in this study. This paper shows adequate results to uncovering the phenomenon that may happen but lacks the investigations on underlying mechanisms. The magnitude of ENSO is mainly driven by the positive and negative feedbacks involving air-sea interactions. In the manuscript, the major atmospheric and oceanic components are depicted, such as the thermocline, zonal wind stress and zonal SST gradient. A clear physical process is needed to understand how ENSO can be modified in G1 and $4\times CO_2$. The Bjerknes feedback, thermocline feedback and heat flux feedback can be evaluated under different scenarios. This may be helpful to illustrate why ENSO in G1 can be modified even though the thermocline, zonal SST gradient and zonal wind stress are not well separated in G1 and piControl. Also, it's necessary to go deeper into the reason why

Interactive
comment

the responses of El Nino and La Nina are different for magnitude change and same for frequency change. 3. This manuscript pays a lot of efforts on how mean state of tropical Pacific might be modified under $4\times CO2$ and G1. A connection between mean state change and ENSO change is simply built by using the previously proposed conclusions, i.e. the reduction of MSSTG in both $4\times CO2$ and G1 indicate increase of extreme El Nino. However, more detailed explanations should be reviewed before applying this theory.

Minor comments: 1. In P11, L24, the calculation of skewness of SST should be clarified in the context. 2. In P9, L22-24 and P11, L36-39, the independent paragraphs seem abrupt for the context. Better to immerse in the other paragraphs. 3. In P12, L6-10, please clarify why quadratic trend to the time series of rainfall data should be excluded. 4. In P13, L13-15, the central Pacific El Nino is not mentioned in the introduction. Also, the question backs to the major comment 1. The HadCM3L may not be able to capture ENSO diversity. 5. In Figure 4c, why the thermocline depth is not significantly changed over the eastern Pacific. If this is the case, is it due to the choice of 24 isotherms? 6. The significance level is 90% for differences between G1, $4\times CO2$ and piControl. How about 95% or even 99%? Will the significant regions be much less? 7. In P23, the height of color bars for figures can be smaller to enlarge the main part of figures. In Figure 2 d & e, symmetric colors are better to represent the negative and positive shadings. 8. In Figure 6 d & e, it's better to set the color bar range with the same ratio as in Figure 5 d & e.

---

## Author Comment (AC1) · 29 Aug 2020

Major Points

1)

It is not clear exactly why the modeled ENSO changed from 4xCO2 to G1 in this model? Is it because of the air-sea heat fluxes act more less as a damping in the eastern equatorial Pacific associated with the mean state change in G1? More interestingly, why G1 does not recover many of the climatic states of piControl? Initial thought would be the ocean state never fully recovers. But as stated in the paper the change in thermocline depth is not statistically different between G1 and piControl. I don't think I came across a plot of subsurface temperature, e.g., depth-longitude differences between 4xCO2

and G1 vs piControl. Perhaps while the thermocline depth statistics do not change, there are still changes in the subsurface ocean temperatures in certain areas.

Reply:

In the revised manuscript, we have calculated ENSO feedbacks, Bjerknes and heat flux, and ocean stratification to explain the mechanisms for change in ENSO. We have added Section 4 elaborating on the mechanism for change in ENSO under both 4xCO2 and G1. (See section 4, from page 17 and line 1 to page 18 and line 29). Specifically we write:

[revised manuscript text omitted]

2)

Nonetheless this leads to the question: How large are the differences in mean state and ENSO statistics between G1 and piControl state in comparison to the internal variability in piControl? For example P9, L20-21: the reduction in MSSTG is 9% in G1, is this substantial compared internal variability in piControl and to that seen during an El Nino?

Reply:

We have shown that the 9 % change in MSSTG under G1 is statistically significant (99
% confidence level) relative to piControl using both Bootstrap resampling and a non-parametric Wilcoxon rank-sum test. The increase in the frequency of extreme El Niño events is due to more frequent reversals of MSSTG (Fig. S3 and Table S2). In the revised manuscript, we have tested the change in frequency under both $4\times CO_2$ and G1, relative to piControl, first by using rainfall > 5 mm day-1 as a threshold for extreme El Niño events and then selecting only those events for which rainfall > 5 mm day-1 and MSSTG < 0. Both methods show a statistically significant increase in extreme El Niño events. Choosing extreme events having MSSTG < 0 assures that strong convection has established over the Niño3 region during the extreme. Further, we have shown the histograms of MSSTG for all samples and exclusively for extreme El Niño events, which indicate more frequent reversals of MSSTG both under $4\times CO_2$ and G1 relative to piControl. In the revised manuscript, we have incorporated the following changes:

Overall there is a change in sign and reduction of MSSTG in $4\times CO_2$ ($\sim$-111 %, 99 % cl) and only decrease in G1 ($\sim$-9 %, 99 % cl) (Fig. S3, and Table S2). (See section 3.1.4, page11, lines 17-19)

A threshold of detrended Niño3 total rainfall of 5 mm day-1 recognizes events as extremes even when the MSSTG is positive and stronger, especially under $4\times CO_2$, which plausibly means that ITCZ might not shift over the equator for strong convection to occur during such extremes. The El Niño event of 2015 is a typical example of such events. We test our results with a more strict criterion by choosing only those events as extremes, which have characteristics similar to that of 1982 and 1997 El Niño events (i.e., Niño3 rainfall > 5 mm day-1 and MSSTG < 0). We declare events having characteristics similar to that of the 2015 event as moderate El Niño events (Fig. S5). Based on this method, we find a robust increase in the number of extreme El Niño events both in $4\times CO_2$ (924 %) and G1 (61 %) at 99 % cl. (See section 3.2.2, page14, lines 26-34)

3)

In many of the plots showing differences between experiments and piControl, the confidence level was set to 90%. Given the long time series of the model output, it should be increased to 95% or even 99%. This would perhaps show more regions in G1 where the differences are not significantly different from piControl.

Reply:

All statistics have been recalculated either with a 95 % or 99 % confidence level. See the manuscript with tracked changes.

4)

The conclusion section could provide the reader with a little perspective on whether it is worth it to do the geoengineering solution in the context of projected increase in extreme ENSO activity. A relevant paper to help the discussion: Trenberth KE, Dai A 2007). Geophys Res Lett 34:L15702. doi: 10.1029/2007GL030524

Reply:

In the revised manuscript (see section 5, page 19, lines 1-14), we have included the following paragraphs/statements:

To conclude, solar geoengineering can compensate many of the GHG-induced changes in the tropical Pacific, but, importantly, not all of them. In particular, controlling the downward shortwave flux cannot correct one of the climate system's most dominant modes of variability, i.e., ENSO, wholly back to preindustrial conditions. The ENSO feedbacks (Bjerkness and heat flux) and more stratified ocean temperatures may induce ENSO to behave differently under G1 than under piControl and $4\times CO2$. Different meridional distributions of shortwave and longwave forcings (e.g., Nowack et al., 2016) resulting in the surface ocean overcooling, and residual warming of the deep ocean are the plausible reasons for the solar geoengineered climate not reverting entirely to the preindustrial state. However, we note that this is a single model study, and more studies are needed to show the robustness and model-dependence of any results discussed here, e.g. using long-term multimodel ensembles from GeoMIP6 (Kravitz et

al., 2015), once the data are released. The long-term Stratospheric Aerosol Geoengineering Large Ensemble (GLENS; Tilmes et al., 2018) data can also be explored to investigate ENSO variability under geoengineering.

5)

P11, L36: Picking a result on one model sounds rather odd as we know that the change in ENSO amplitude varies widely across models (e.g., Collins et al. 2010). In a recent study by Cai et al. (2018, Nature, https://www.nature.com/articles/s41586-018-0776-9), however, there seems to be a stronger inter-model agreement on the increase in ENSO amplitude in models that are able to simulate ENSO flavors (see their Extended Data Fig. 8b), as implied in the PC1-PC2 space. So does the HadCM3L model capture the nonlinear relationship between PC1 and PC2 as observed? Here PC1 and PC2 refer to the first and second eigenmodes of tropical Pacific SST (see their Fig. 1). Also, it is relevant to discuss the results of Cai et al. (2018) in 1st paragraph of Page 3.

Reply:

Regarding the change in amplitude, we refer to other studies in the revised manuscript and include the following paragraphs/statements:

Previous studies found that climate models produced mixed responses (both increases and decreases in amplitude) in terms of how ENSO amplitude change with global warming (see Latif et al. 2009; Collins et al. 2010; Vega-Westhoff and Sriver 2017). However, Cai et al. (2018) found an intermodel consensus, for models capable of reproducing ENSO diversity, for strengthening of ENSO amplitude under A2, RCP4.5, and RPC8.5 transient scenarios. (See section 3.2.1, page 13, lines 6-11)

We have included a separate section (2.4) under the title "ENSO representation in HadCM3L" which discusses the HadCM3L capability to simulate ENSO diversity as described by Cai et al. (2018). We have incorporated the following paragraphs/statements in the revised manuscript:

Before employing HadCM3L for studying ENSO variability under 4×CO2, and G1, we evaluate its piControl simulation against present-day observational data. (See section 2.4, page 6, lines 40-41)

Further, we have included the following paragraphs (see section 2.4, page 7, and line 14 to next page line 21):

[revised manuscript text omitted]

We conclude that HadCM3L has a reasonable skill for studying long-term ENSO variability and its response to solar geoengineering. However, we also highlight the need

for and hope to motivate future modelling studies that will help identify model dependencies in the ENSO response.

We discuss the results of Cai et al. (2018) as follows:

As diagnosed from Sea Surface Temperature (SST) indices in state-of-the-art AOGCMs, there was no intermodel consensus about change in frequency of ENSO events and amplitude in a warming climate (Vega-Westhoff and Sriver 2017; Yang et al., 2018) until Cai et al. (2018) used SST indices based on Principal Component Analysis (PCA). (See section 1, page 2, and line 41 to next page line 4)

However, Cai et al. (2018) later found robust evidence of a consistent increase in El Niño amplitude in the subset of CMIP5 climate models, which were capable of reproducing both eastern and central Pacific ENSO modes. (See section 1, page 3, line 11-14)

Please see Supplementary Fig. S1 and Tables S5-S6 as well.

6)

P7, L10: make clear the results are in *qualitative* agreement with previous studies. Not all of the cited studies are based on 4xCO2.

Reply:

We check our results and categorically mention that our results qualitatively agree with previous studies. Thus we add the following change:

Our SST results under 4xCO2 qualitatively agree with previous studies (Liu et al., 2005; van Oldenborgh et al., 2005; Collins et al., 2010; Vecchi and Wittenberg et al., 2010; Cai et al., 2015a; Huang and Ying et al., 2015; Luo et al., 2015; Kohyama et al., 2017; Nowack et al., 2017). (See section 3.1.1, page 9, line 9-12)

7)

P.7, L13: some studies argue against the use of "El Nino-like" term in describing the mean-state change under greenhouse forcing (e.g., Collins et al. 2010; see also Xie et al. 2010 https://journals.ametsoc.org/doi/10.1175/2009JCLI3329.1). Cautionary is needed to avoid confusions. A relevant reference on the mean state change: diNezio et al https://journals.ametsoc.org/doi/full/10.1175/2009JCLI2982.1.

Reply:

We have deleted the term "El Nino-like" from the revised manuscript and have replaced it with appropriate words like "a significant mean warming" or "a warming state" (See section 1, page 3, lines 18-19; and section 3.1.1 page 8, line 37)

8)

Fig. 2d, e: title of the figure states +0.21 mm/day, -0.23 mm/day. Please explain in the caption that those numbers correspond to the area average difference between experiment and piControl in the tropical Pacific (state domain).

Reply:

The following change is made in the caption of Fig. 1:

The numbers in a-c represent a mean temperature in the corresponding simulation, and numbers in d-e represent an area-averaged difference of piControl with $4 \times CO_2$ and G1, respectively, in the tropical Pacific region (25o N-25o S; 90o E-60o W). (See page 28, lines 8-11)

The following change is made in the caption of Fig. 2:

The numbers in a-c represent mean rainfall in the corresponding simulation, and numbers in d-e represent an area-averaged difference of piControl with $4 \times CO_2$ and G1, respectively, in the tropical Pacific region (25o N-25o S; 90o E-60o W). (See page 29, lines 7-10)

9)

P9, L22-24: This sentence needs a rework. Avoid the word "observe" on model analysis (models are not observations). I think Wang et al. (2017) was referring to zonal temperature gradient between the maritime continent and central Pacific, not eastern Pacific. The difference is not significant in RCP2.6, but should be significant in RCP8.5 (Cai et al. 2015, Nature Climate Change on extreme La Nina).

Reply:

The use of word "observed" for modelled data has been replaced with appropriate words in the revised manuscript. The reference of Wang et al. (2017) for weakening of ZSSTG has also been removed from the revised manuscript. Instead we add the following statements:

Our results under 4xCO2 are in agreement with Coats and Karnauskas (2017), who using several climate models found a weakening of the ZSSTG under the RCP8.5 scenario.(see section 3.1.4, page 11, line 11-13)

The weakening of the MSSTG is qualitatively in agreement with previous studies under increased GHG forcings (e.g., Cai et al., 2014; Wang et al., 2017). (See section 3.1.4, page 11, lines 21-22)

10)

Fig. 7: Please indicate clearly in the caption that the timeseries have been detrended with non ENSO related trend removed following Cai et al. (2017). Otherwise it would create confusion as other studies show that the 2015/16 Nino3 rainfall is close to the 5 mm/day threshold and is thus classified as an extreme El Nino (Santoso et al. 2017). In panel c, d, it must be rainfall anomalies that are shown because there are negative rainfall values, so wouldn't the 4 or 3 mm/day threshold be applied here? Panel a and b also have negative rainfall values. Please double check.

Reply:

In the captions, we have added the following text:

Following Cai et al. (2014), a non-ENSO related trend has been removed from the rainfall time series. (See Fig. 7, page 32, lines 8-9; and Fig.S5-S6)

In Fig. 7 and Fig. S5-6, revised manuscript, we have shown total rainfall after subtracting the non-ENSO related trend as described by Cai et al. (2017). In the previous manuscript, we subtracted the non-ENSO related trend, including the intercept term; therefore, negative values were present, and it's been corrected now.

11)

P12, L28-31: under 4xCO2 the rainfall skewness is dramatically reduced. Does that mean there are less extreme El Nino based on the rainfall definition? If so, this does not seem consistent with the PPE results of Cai et al. (2014) using the same model.

Reply:

In the revised manuscript, we have included the analysis for 4×CO2. We show that extreme El Niño events increase under 4×CO2 using metrics based on rainfall and E-index (See section 3.2.2). The climate regime under 4×CO2 is substantially different from that of piControl (See Fig. S8). The comparison of piControl and 4×CO2 is not simple as mean rainfall, despite zero skewness, significantly shifts to a higher value (9.8 mm day-1) under 4×CO2. We have added the following text in the revised manuscript:

[revised manuscript text omitted]

12)

P13, L28-39: The characterization of extreme La Nina is based on Nino4 (Cai et al. 2015), so it is not clear how Nino3 and Nino3.4 indices are used here to infer changes in extreme La Nina.

Reply:

We have deleted inferences based on Nino3 and Nino3.4 in section 3.2.3 of the revised manuscript.

Figure presentation

13)

Fig. 1e, some areas look white (e.g., eastern equatorial Pacific which is supposed to be approx. -0.2C p7, L9) while the colorbar does not have white on it.

Reply:

We have reproduced Fig. 1e with a different color bar, and visibility of colors has improved in the revised manuscript.

14)

Figure 10: the color limit does not seem correct, which shows much larger values in e, f G1-piControl than the composite anomalies themselves in panels a-d.

Reply:

We have corrected the color limits in Fig. 10.

15)

The colorbar of Fig. 2, right panel especially is not ideal. It is hard to immediately see which are positive or negative without referring to the colorbar.

Reply:

In the revised manuscript, we have reproduced Fig. 2 with a diverging color bar.

16)

Might be best to have the same color scale for comparing the results of 4xCO2 – piControl vs G1 –piControl. This is to convey the message the difference is much smaller for G1 – piControl than for 4xCO2.

Reply:

The differences under G1-piControl are small; if we use the same color bar for 4xCO2-piControl and G1-piControl, most of the information is suppressed for G1-piControl.

Therefore we have used two different color bars.

Minor points

17)

Page 4, L34: that sentence is due to Cai et al. (2014).

Reply:

We have cited Cai et al. (2014) in the revised manuscript. (See section 2.4, page 7, line 2)

18)

P4, L35: delete "the northern part of" – the ITCZ is located north of equator, and that rainfall band moves equatorward during strong El Nino events.

Reply:

We have deleted "the northern part of" in the revised manuscript.

19)

P5, L23: "ggradients"

Reply:

Corrected in the revised manuscript. (See section 2.3, page 5, line 24)

20)

P6, L2: extreme El Ninos are not resulting in just "anomalous rainfall" but unusually large rainfall in the eastern equatorial Pacific.

Reply:

We have deleted the word anomalous and modified the text as follows:

.... Niño3 region resulting in rainfall higher than 5mm day-1 (Cai et al., 2014). (See section 2.3, page 6, lines 1-2)

21)

P6, L 35: "depicts this SSTasymmetry between the western and eastern equatorial Pacific well (Fig. 1a)." – not clear since the observed counterpart is not presented.

Reply:

In the text, we have cited a reference for comparing the piControl SST asymmetry with an observational dataset. We have modified the version as follows:

The piControl simulation (Fig. 1a) reproduces the SST asymmetry between the western and eastern equatorial Pacific well (cf. Fig 1a in Vecchi and Wittenberg 2010). (See section 3.1.1, page 8, lines 34-36)

22)

P8, L10: "problem" – not clear, in what way it is a problem?

Reply:

The word "problem" has been deleted in the revised manuscript. We have modified the text as follows:

That is, while the relative additional rainfall asymmetry between the western and eastern Pacific in $4\times CO_2$ is mostly resolved in G1, the tropical Pacific is overall wetter under $4\times CO_2$ but drier in G1. (See section 3.1.2, page 10, lines 13-15)

23)

P9, L19: repetitive: El Nino being stronger than La Nina already implies asymmetric amplitude.

Reply:

In the revised manuscript, we have modified text as follows:

However, the use of standard deviations to define ENSO amplitude is suboptimal, because amplitudes of El Niño and La Niña events are asymmetric, i.e., in general, El Niño events are stronger than La Niña events (An and Jin 2004; Schopf and Burgman 2006; Ohba and Ueda 2009; Ham 2017). (See section 3.2.1, page 13, lines 22-25)

24)

P9, L29: the shoaling of thermocline is also due to increased stratification associated with surface intensified warming in response to greenhouse forcing.

Reply:

We have added the following text in the revised manuscript:

In 4xCO2, most likely the weakened easterlies (as noticed in Sect. 3.1.3; e.g., Yeh et al., 2009, Wang et al., 2017) and greater ocean temperature stratification due to increased surface warming (see Sect. 4 and Cai et al., 2018) lead to a significant shoaling of the thermocline across the western and central equatorial Pacific. In contrast, relatively little change takes place between 130o W and 90o W. In a CMIP3 multimodel (SRESA1B scenario) ensemble, Yeh et al. (2009) found a more profound deepening of the thermocline in this part of the eastern equatorial Pacific; however, for example, Nowack et al. (2017) did not find such changes under 4xCO2 (cf. their Fig. S9). One possible explanation for this behaviour is the competing effects of upper-ocean warming (which deepens the thermocline) and the weakening of westerly zonal wind stress, causing thermocline shoaling (see Kim et al. 2011a). (See section 3.1.5, from page 11 and line 37 to next page line 8)

25)

P9, L32-36: why not use the maximum of vertical temperature gradient as a proxy of thermocline depth for all scenarios?

Reply:

In the revised manuscript, we have included a map for ocean stratification; we think it can provide some details on this. Further, model ocean vertical resolution (13 levels) is not very high to calculate maximum vertical temperature gradient.

26)

P14, L6-7: for extreme El Nino events, are the PWC, SST, and rainfall anomalies strengthened as well?

Reply:

For extreme El Niño events, the PWC, SST, and rainfall anomalies are weakened. We have rectified the text as follows:

These composites provide process-based evidence for the strengthening (weakening) of extreme La Niña (El Niño) events in G1. We show that the PWC, SST, and composite rainfall anomalies are strengthened for extreme La Niña events, while they are weakened for extreme El Niño events under G1. (See section 3.3, page, 16, lines 5-8)

27)

P14, L23-25: this must be referring to the difference between G1 and piControl. Please make that clear.

Reply:

In the revised manuscript, we have modified text as follows:

During extreme El Niño events, in G1, we find reduced SST (Fig. 9e) and rainfall anomalies (Fig. 10e) over the eastern and western equatorial Pacific with a consistent weakening of the eastern and western branch of PWC (Fig. 11e). (See section 3.3.1, page 16, lines 15-17)

NB. Please see attached Supplementary for figure captions.

Please also note the supplement to this comment:
https://acp.copernicus.org/preprints/acp-2018-1312/acp-2018-1312-AC1-
supplement.pdf
* * *
[Figure]

[Figure]

**Fig. 1.** Figure 7.

[Figure]

**Fig. 2.** Figure 15.

[Figure]

**Fig. 3.** Figure S1

[Figure]

**Fig. 4.** Figure S3

**Observed: El Niño Events (a)**

Extreme = 2 ± 1.4
Moderate = 1 ± 0.9
Weak = 7 ± 2.4
Total = 10 ± 2.7
Other

1982
1997
2015

**piControl: El Niño Events (b)**

Extreme = 62 ± 7.6
Moderate = 88 ± 9.0
Weak = 150 ± 11.3
Total = 300 ± 14.6
Other

**4×CO2: El Niño Events (c)**

Extreme = 635 ± 15.0 (+924%)*
Moderate = 304 ± 14.5 (+245)*
Weak = 0
Total = 939 ± 6.8 (+213%)*
Other

**G1: El Niño Events (d)**

Extreme = 100 ± 9.4 (+61%)*
Moderate = 76 ± 8.4 (−14%)
Weak = 161 ± 11.6 (+7%)
Total = 337 ± 14.8 (+12%)**
Other

**Fig. 5.** Figure S5

[Figure]

[Figure]

**Fig. 6.** Figure S6

[Figure]

[Figure]

**Fig. 7.** Figure S8

[Figure]

**Fig. 8.** Figure S9

[Figure]

**Fig. 9.** Figure S10

**Table S2.** Meridional SST Gradient (MSSTG)

| Experiment | Mean (ºC) | Difference w.r.t. piControl (ºC) | Std. Dev. 10,000 Realizations (ºC) | ~ Change w.r.t. piControl (%) |
|---|---|---|---|---|
| piControl | 1.38* | | 0.0265 | |
| 4×CO₂ | -0.15* | -1.53 | | -111* |
| G1 | 1.25* | -0.13 | | -9* |

Key: *99 % cl; **95 % cl

**Fig. 10.** Table S2

**Table S3.** Total number of El Niño events (SST > 0.5 s.d.)

| Experiment | No. of Events | Difference w.r.t. piControl | Std. Dev. 10,000 Realizations | ~ Change w.r.t. piControl (%) |
|---|---|---|---|---|
| piControl | 300 [300] | | 14.6 [14.6] | |
| 4×CO₂ | 161 [565] | 139 [265] | | -46* [+88*] |
| G1 | 337 [337] | 37 [37] | | +12** [+12**] |

Key: Niño3 [E-Index]; *99 % cl; **95 % cl

**Fig. 11.** Table S3

**Table S5.** Mean DJF Heat Flux (hf) Feedback

| Experiment | hf feedback or Damping Coefficient (Wm$^{-2}$/°C) | Difference w.r.t. piControl (Wm$^{-2}$/°C) | Std. Dev. 10,000 Realizations (Wm$^{-2}$/°C) | ~ Change w.r.t. piControl (%) |
|---|---|---|---|---|
| ERA5 | -14.59 | | | |
| piControl | -14.70 | | 0.52 | |
| 4×CO$_2$ | -21.90 | +7.19 | | +48* |
| G1 | -14.85 | +0.15 | | +1.0 |

*99% cl; **95% cl; Calculation period: ERA5 (41-yrs); HadCM3L (990-yrs)

**Fig. 12.** Table S5

**Table S6.** Mean DJF Bjerknes (BJ) Feedback

| Experiment | BJ feedback ($10^{-2}$ Nm$^{-2}$/°C) | Difference w.r.t. piControl ($10^{-2}$ Nm$^{-2}$/°C) | Std. Dev. 10,000 Realizations (Wm$^{-2}$/°C) | ~ Change w.r.t. piControl (%) |
|---|---|---|---|---|
| ERA5 | 3.3 | | | |
| piControl | 3.3 | | 0.0091 | |
| 4×CO$_2$ | 2.2 | -1.1 | | -33* |
| G1 | 3.5 | +0.2 | | +6* |

*99% cl; **95% cl; Calculation period: ERA5 (41-yrs); HadCM3L (990-yrs)

**Fig. 13.** Table S6

**Table S7.** Mean DJF Ocean Stratification

| Experiment | Stratification (°C) | Difference w.r.t. piControl (°C) | Std. Dev. 10,000 Realizations (°C) | ~ Change w.r.t. piControl (%) |
|---|---|---|---|---|
| piControl | 2.28* | | 0.0331 | |
| 4×CO₂ | 5.06* | +2.78 | | +122* |
| G1 | 2.37* | +0.09 | | +4** |

*99% cl; **95% cl

**Fig. 14.** Table S7

**Supplement:**

**Figure Captions**

**Figure 7.** *Relationship between MSSTG and Niño3 rainfall for (a) observations (b) piControl (c) 4×CO$_2$, and (d) G1. A solid black horizontal line indicates a threshold value of 5 mm day$^{-1}$. See text for the definition of extreme, moderate, and total El Niño events. A single (double) asterisk indicates that the change in frequency, relative to piControl, is statistically significant at 99 % (95 %) cl. Numbers with a ± symbol indicate s.d. calculated with 10,000 bootstrap realizations. Following Cai et al. (2014), a non-ENSO related trend has been removed from the rainfall time series.*

**Figure 15.** *BJ feedback (μ; 10$^{-2}$ Nm$^{-2}$/$^{\circ}$C) for (a) piControl (b) 4×CO$_2$, and (c) G1. The value with ± sign indicates s.d. of μ after 10,000 bootstrap realizations. An asterisk indicates statistical significance at 99 % cl. Mean change in ocean temperature, (d) 4×CO$_2$-piControl, and (e) G1-piControl. The black box shows the area averaging region for upper ocean temperature, and the black line shows the lower layer used for calculation of stratification as a difference of upper and lower layer. Stipples indicate grid points with statistical significance at 99 % cl using a non-parametric Wilcoxon rank-sum test.*

**Figure S1.** *ENSO diversity and nonlinear relationship between PCs. First monthly principal pattern, EOF1, for (a) ERA5 and (b, c) piControl. Second monthly principal pattern, EOF2, for (d) ERA5 and (e, f) piControl. DJF EP pattern for (g) ERA5 and (h, i) piControl. DJF CP pattern for (j) ERA5 and (k, l) piControl. The nonlinear relationship between PC1 and PC2 for (m) ERA5 and (n, o) piControl. The blue box indicates the Niño3 (Niño4) region in a-c, and g-I (d-f and j-l). The left and the middle panel shows EOF analysis over the 41 years of ER5 (1979-2019) and piControl. The right panel shows EOF analysis over 990-year of piControl.*

**Figure S3.** *Histogram of MSSTG for piControl, 4×CO$_2$, and G1 for all samples (a) and for extreme El Niño events. The values are plotted at the centre of each bin with an interval of 0.5 $^{\circ}$C. Blue, red, and green vertical lines indicate climatological mean values of MSSTG under piControl (1.38 $^{\circ}$C), 4×CO$_2$ (-0.15 $^{\circ}$C), and G1 (1.25 $^{\circ}$C), respectively. H = 1 indicates that the shift in the mean is statistically significant at 99 % cl using a non-parametric Wilcoxon rank-sum test.*

**Figure S5.** *Relationship between MSSTG and quadratically detrended Niño3 rainfall for (a) observations (b) piControl (c) 4×CO$_2$, and (d) G1. The solid black horizontal line indicates a threshold of 5 mm day$^{-1}$. A single (double) asterisk indicates that the change in frequency, relative to piControl, is statistically significant at 99 % (95 %) cl. Numbers with a ± symbol indicate s.d. calculated with 10,000 bootstrap realizations. Following Cai et al. (2014), a non-ENSO related trend has been removed from the rainfall time series. Events are classified as: Extreme (Niño3 rainfall > 5 mm day$^{-1}$ and MSSTG < 0), moderate (Niño3 rainfall > 5 mm day$^{-1}$ and MSSTG > 0), weak (Standardized Niño3 SSTs > 0.5 $^{\circ}$C and Niño3 rainfall < 5 mm day$^{-1}$), total is sum of extreme, moderate, and weak events.*

**Figure S6.** *Relationship between MSSTG and linearly detrended Niño3 rainfall for (a) observations (b) piControl (c) 4×CO$_2$, and (d) G1. The solid black horizontal line indicates a threshold of 5 mm day$^{-1}$. A single (double) asterisk indicates that the change in frequency, relative to piControl, is statistically significant at 99 % (95 %) cl. Numbers with a ± symbol indicate s.d. calculated with 10,000 bootstrap realizations. Following Cai et al. (2014), a non-ENSO related trend has been removed from the rainfall time series. Events are classified as: Extreme (Niño3 rainfall > 5 mm day$^{-1}$ and MSSTG < 0), moderate (Niño3 rainfall > 5 mm day$^{-1}$ and MSSTG > 0), weak (Standardized Niño3 SSTs > 0.5 $^{\circ}$C and Niño3 rainfall < 5 mm day$^{-1}$), total is sum of extreme, moderate, and weak events.*

**Figure S8.** *Histogram of Niño3 rainfall for piControl, 4×CO$_2$, and G1. The values are plotted at the centre of each bin with an interval of 1 mm day$^{-1}$. Blue, red, and green vertical lines indicate climatological mean values of Niño3 rainfall under piControl (2.9 mm day$^{-1}$), 4×CO$_2$ (9.8 mm day$^{-1}$), and G1 (3.2 mm day$^{-1}$), respectively. H = 1 indicates that the shift in the mean is statistically significant at 99 (95) % cl for 4×CO$_2$ (G1) using the non-parametric Wilcoxon rank-sum test. The grey vertical line show threshold of 5 mm day$^{-1}$.*

***Figure S9.*** *Histogram of ZSSTG anomalies for (a) all samples, (b) extreme El Niño events only, and (c) extreme La Niña events only. The values are plotted at the centre of each bin with an interval of 0.5 $^{o}$C. In a blue, red, and green solid vertical lines indicate climatological median ZSSTG under piControl (0.07 $^{o}$C), $4 \times CO_2$ (-1.54 $^{o}$C), and G1 (-0.28 $^{o}$C), respectively, for all samples. In b, blue, red, and green dashed vertical lines indicate climatological median ZSSTG under piControl (-1.83 $^{o}$C), $4 \times CO_2$ (-1.71 $^{o}$C), and G1 (-1.96 $^{o}$C), respectively, for extreme El Niño events. In c, blue, and green dashed vertical lines indicate climatological median ZSSTG under piControl (1.37 $^{o}$C) and G1 (1.52 $^{o}$C), respectively, for extreme La Niña events. H = 1 indicates that using a non-parametric Wilcoxon rank-sum test, the shift in the median is statistically significant at 99 (95) % cl in a (b). H = 0 means that the shift in the median is not statistically significant. The ZSSTG is defined as the difference between SST in the Maritime continent (5$^{o}$ N-5$^{o}$ S; 100$^{o}$ E-126$^{o}$ E) and eastern equatorial Pacific (Niño3 region: 5$^{o}$ N-5$^{o}$ S, 150$^{o}$ W-90$^{o}$W). The anomalies are calculated relative to piControl.*

***Figure S10.*** *Histogram of ZSSTG anomalies for (a) all samples and (b) extreme La Niña events only. The values are plotted at the centre of each bin with an interval of 0.5 $^{o}$C. Blue, red, and solid green lines indicate climatological median ZSSTG under piControl (-0.14 $^{o}$C), $4 \times CO_2$ (-1.37 $^{o}$C), and G1 (-0.40 $^{o}$C), respectively, for all samples. Blue, red, and green dash-dotted lines indicate climatological median ZSSTG under piControl (0.84 $^{o}$C), $4 \times CO_2$ (-0.03 $^{o}$C), and G1 (0.72 $^{o}$C), respectively, for all La Niña events. In b, blue, red, and green dashed lines indicate climatological median ZSSTG under piControl (1.52 $^{o}$C) and G1 (3.35 $^{o}$C), respectively, for extreme La Niña events. H = 1 indicates that the shift in the median is statistically significant at 99 % cl using the non-parametric Wilcoxon rank-sum test. The ZSSTG is defined as the difference between SST in the Maritime continent (5$^{o}$ N-5$^{o}$ S; 100$^{o}$ E-126$^{o}$ E) and central equatorial Pacific (Niño4 region: 5$^{o}$ N-5$^{o}$ S, 160$^{o}$ E-150$^{o}$ W) (Cai et al., 2015). The anomalies are calculated relative to piControl.*

---

## Author Comment (AC2) · 29 Aug 2020

The authors thank to both referees for their comments and suggestions, which have greatly helped us to improve our manuscript. Below, we reply point-by-point, highlighting the changes we have implemented. The primary concern of the referees was the evaluation of the climate model capability to simulate ENSO variability, and the lack of detailed explanations on possible mechanisms responsible for changes in ENSO both under $4\times CO_2$ and solar geoengineering (G1). In the revised manuscript, we therefore put a strong emphasis on model evaluation and are able to confirm the necessary model skill (section 2.4). We also provide an entirely new section (section 4) on possible mechanisms behind the changes in ENSO extremes and ENSO amplitudes.

[Figure]

Major Points

1)

To study the ENSO changes under solar geoengineering, the results are all based on one single model HadCM3L. In Cai et al. (2014) and Collins et al. (2001), the model they used is HadCM3. I admit that HadCM3L and HadCM3 are identical in most aspects, but there are still differences between these two simulations. The differences should be mentioned in this study because the HadCM3L may not be skillful in reproducing ENSO variabilities, and thus the sentence in P4 L32-33 may not be completely correct. I suggest that the ENSO simulated in HadCM3L should be addressed first, regarding its magnitude and pattern. For instance, the EOF analyses can be carried out on the piControl simulations. It will help us to have a general idea of how capable the HadCM3L is in simulating the ENSO and its diversity, and what's the biases compared with observations. As pointed out in Cai et al. (2018), the magnitude and location of ENSO events are inconsistent among models. The averaged SSTA in a fixed box to measure the intensity of ENSO can be tricky. A look at the ENSO pattern in HadCM3L can also facilitate a better ENSO extreme definition, i.e. the Nino indices may not be best to define ENSO intensity. At least, a glimpse of the Figure 8 reveals that ENSO simulation is not good enough, especially the shape, maximum location and horseshoe-shaped cold SSTA in the western Pacific during El Nino events.

Reply:

We have deleted the text referring to P4 and L32-33 in the revised manuscript. In the revised manuscript we have evaluated the model skill for reproducing ENSO diversity following Cai et al. (2018). The HadCM3L belongs to the family of HadCM3 models; the only difference between HadCM3 and HadCM3L is lower ocean resolution. We have included a separate section on model evaluation. We have also mentioned the apparent biases in HadCM3L compared to observations. We find that HadCM3L has a reasonable skill to simulate ENSO and can be employed for the current study. The

HadCM3L simulates the sea surface temperature maximum anomaly pattern over the Niño3 region. In the revised manuscript, we have made the following additions:

HadCM3L stems from the family of HadCM3 climate models; the only difference is lower ocean resolution (HadCM3: 1.25o × 1.25o; Valdes et al., 2017). (see section 2.1, page 4, lines 25-27)

Before employing HadCM3L for studying ENSO variability under 4×CO2, and G1, we evaluate its piControl simulation against present-day observational data. (see section 2.4, page 6, lines 40-41)

Further, we have included the following paragraphs (section 2.4, page 7, and line 14 to next page line 21):

[revised manuscript text omitted]

We conclude that HadCM3L has a reasonable skill for studying long-term ENSO variability and its response to solar geoengineering. However, we also highlight the need for and hope to motivate future modelling studies that will help identify model dependencies in the ENSO response.

Please see Supplementary Fig. S1 and Tables S5-S6 as well.

2)

The change of extreme ENSO under solar geoengineering is a major concern in this study. This paper shows adequate results to uncovering the phenomenon that may happen but lacks the investigations on underlying mechanisms. The magnitude of ENSO is mainly driven by the positive and negative feedbacks involving air-sea interactions. In the manuscript, the major atmospheric and oceanic components are depicted, such as the thermocline, zonal wind stress and zonal SST gradient. A clear physical process is needed to understand how ENSO can be modified in G1 and 4_CO2. The Bjerknes feedback, thermocline feedback and heat flux feedback can be evaluated under different scenarios. This may be helpful to illustrate why ENSO in G1 can be modified even though the thermocline, zonal SST gradient and zonal wind stress are not well separated in G1 and piControl. Also, it's necessary to go deeper into the reason why the responses of El Nino and La Nina are different for magnitude change and same for frequency change.

Reply:

In the revised manuscript, we have calculated ENSO feedbacks, Bjerknes and heat flux, and ocean stratification to explain the mechanisms for change in ENSO. We have added Section 4 elaborating on the mechanism for change in ENSO under both 4xCO2

and G1. (See section 4, from page 17 and line 1 to page 18 and line 29). Specifically we write:

[revised manuscript text omitted]

3)

This manuscript pays a lot of efforts on how mean state of tropical Pacific might be modified under 4_CO2 and G1. A connection between mean state change and ENSO change is simply built by using the previously proposed conclusions, i.e. the reduction of MSSTG in both 4_CO2 and G1 indicate increase of extreme El Nino. However, more detailed explanations should be reviewed before applying this theory.

Reply:

In the revised manuscript, we have tested the change in frequency under both 4×CO2 and G1, relative to piControl, first by using rainfall > 5 mm day-1 as a threshold for extreme El Niño events and then selecting only those events for which rainfall > 5 mm day-1 and MSSTG < 0. Both methods show a statistically significant increase in extreme El Niño events. Choosing extreme events having MSSTG < 0 assures that strong convection has established over the Niño3 region during the extreme. Further, we have shown the histograms of MSSTG for all samples and exclusively for extreme

El Niño events, which indicate more frequent reversals of MSSTG both under 4×CO2 and G1 relative to piControl. See also the discussion on mechanism now presented in Sect. 4 and included in a response above. In the revised manuscript, we have further incorporated the following changes:

A threshold of detrended Niño3 total rainfall of 5 mm day-1 recognizes events as extremes even when the MSSTG is positive and stronger, especially under 4×CO2, which plausibly means that ITCZ might not shift over the equator for strong convection to occur during such extremes. The El Niño event of 2015 is a typical example of such events. We test our results with a more strict criterion by choosing only those events as extremes, which have characteristics similar to that of 1982 and 1997 El Niño events (i.e., Niño3 rainfall > 5 mm day-1 and MSSTG < 0). We declare events having characteristics similar to that of the 2015 event as moderate El Niño events (Fig. S5). Based on this method, we find a robust increase in the number of extreme El Niño events both in 4×CO2 (924 %) and G1 (61 %) at 99 % cl. (Section 3.2.2, page14, lines 26-34)

Minor Points

1)

In P11, L24, the calculation of skewness of SST should be clarified in the context.

Reply:

In the revised manuscript we have made the following changes:

Skewness is a measure of asymmetry around the mean of the distribution (see eq. S1). Positive skewness means that in given data distribution, the tail of the distribution is spread out towards high positive values, and vice versa (Ghandi et al., 2016). (See section 2.4, page 7, lines 2-5)

(See Supplementary, page 13)

$S = [1/(n\text{-}1)] \left( \sum\nolimits\_{\ominus n} (X\_i - X \ominus) \right) / \sigma\hat{}3$ . . . . . . . . . . . . . . . . . . . . . . . . . . . . (S1; Ghandi

et al., 2016)

Where

S = skewness n = sample size X_i = sample ith observation X Ì̆Ě = sample mean $\sigma\hat{\ }3$ = sample standard deviation

2)

In P9, L22-24 and P11, L36-39, the independent paragraphs seem abrupt for the context. Better to immerse in the other paragraphs.

Reply:

We have edited and merged the text with other paragraphs as follows:

The weakening of the MSSTG is qualitatively in agreement with previous studies under increased GHG forcings (e.g., Cai et al., 2014; Wang et al., 2017). (see section 3.1.4, page 11, lines 21-22)

Previous studies found that climate models produced mixed responses (both increases and decreases in amplitude) in terms of how ENSO amplitude change with global warming (see Latif et al. 2009; Collins et al. 2010; Vega-Westhoff and Sriver 2017). However, Cai et al. (2018) found an intermodel consensus, for models capable of reproducing ENSO diversity, for strengthening of ENSO amplitude under A2, RCP4.5, and RPC8.5 transient scenarios. (see section 3.2.1, page 13, lines 6-11)

3)

In P12, L6-10, please clarify why quadratic trend to the time series of rainfall data should be excluded.

Reply:

In the revised manuscript, the text has been edited as follows:

To study changes in El Niño frequency, we first need to define what constitutes an El

Niño event. We here define extreme El Niño events as episodes when monthly-mean DJF Niño3 total rainfall exceeds 5 mm day-1, following the threshold definition by Cai et al. (2014). However, as pointed out by Cai et al. (2017), trends in Niño3 rainfall are mainly driven by two factors: (1) the change in the mean state of the tropical Pacific and (2) the change in frequency of extreme El Niño events. Therefore, since we want to focus on the changes in the extremes, we need to remove contribution (1) from the raw Niño3 time series. We, therefore, fit a quadratic polynomial to the time series of rainfall data from which all extreme El Niño events (DJF total rainfall > 5 mm day-1) have been excluded and then subtract this trend from the raw Niño3 rainfall time series. Linearly detrending the rainfall time series produces similar results. (See section 3.2.2, page 14, lines 4-14)

4)

In P13, L13-15, the central Pacific El Nino is not mentioned in the introduction. Also, the question backs to the major comment 1. The HadCM3L may not be able to capture ENSO diversity.

Reply:

We have deleted the referred text from the revised manuscript.

5)

In Figure 4c, why the thermocline depth is not significantly changed over the eastern Pacific. If this is the case, is it due to the choice of 24 isotherms?

Reply:

In a CMIP3 multimodel (SRESA1B scenario) ensemble, Yeh et al. (2009) showed a deepening of the thermocline in the eastern equatorial Pacific; however, Nowack et al. (2017) did not find any change under 4xCO2. Both studies defined thermocline using a maximum vertical temperature gradient. Thus, we believe that no-significant-change in the eastern Pacific is not due to the choice of 24 oC isotherm, but rather due to a

cancellation of competing effects on thermocline depth. In the revised manuscript, we have therefore added the following text:

In 4xCO2, most likely the weakened easterlies (as noticed in Sect. 3.1.3; e.g., Yeh et al., 2009, Wang et al., 2017) and greater ocean temperature stratification due to increased surface warming (see Sect. 4 and Cai et al., 2018) lead to a significant shoaling of the thermocline across the western and central equatorial Pacific. In contrast, relatively little change takes place between 130o W and 90o W. In a CMIP3 multimodel (SRESA1B scenario) ensemble, Yeh et al. (2009) found a more profound deepening of the thermocline in this part of the eastern equatorial Pacific; however, for example, Nowack et al. (2017) did not find such changes under 4xCO2 (cf. their Fig. S9). One possible explanation for this behaviour is the competing effects of upper-ocean warming (which deepens the thermocline) and the weakening of westerly zonal wind stress, causing thermocline shoaling (see Kim et al. 2011a). (see section 3.1.5, from page 11 and line 37 to next page line 8)

6)

The significance level is 90% for differences between G1, 4_CO2 and piControl. How about 95% or even 99%? Will the significant regions be much less?

Reply:

All statistics have been recalculated either with a 95 % or 99 % confidence level. See the manuscript with track changes.

7)

In P23, the height of color bars for figures can be smaller to enlarge the main part of figures. In Figure 2 d & e, symmetric colors are better to represent the negative and positive shadings.

Reply:

**[ACPD](ACPD)**

Interactive
comment

In the revised manuscript, all figures have been re-plotted with relatively small and diverging color bars.

8)

In Figure 6 d & e, it's better to set the color bar range with the same ratio as in Figure 5 d & e.

Reply:

In the revised manuscript, both figures are re-plotted with the same color range.

NB. Please see supplementary for figure captions.

Please also note the supplement to this comment:
https://acp.copernicus.org/preprints/acp-2018-1312/acp-2018-1312-AC2-supplement.pdf

**Observed: El Niño Events (a)**

Extreme = 3 ± 1.7
Moderate = 7 ± 2.4
Total = 10 ± 2.7
Other

1982
1997
2015

**piControl: El Niño Events (b)**

Extreme = 150 ± 11.5
Moderate = 150 ± 11.3
Total = 300 ± 14.6
Other

**4×CO2: El Niño Events (c)**

Extreme = 939 ± 6.8 (+526%)*
Moderate = 0 ± (−100%)*
Total = 939 ± 6.8 (+213%)*
Other

**G1: El Niño Events (d)**

Extreme = 176 ± 12.0 (+17%)**
Moderate = 161 ± 11.6 (+7%)
Total = 337 ± 14.8 (+12%)**
Other

Axis labels: Detrended Niño3 Rainfall (mm/day) vs MSSTG ($^o$C)

**Fig. 1.** Figure 7.

[Figure]

**Fig. 2.** Figure 15.

[Figure]

**Fig. 3.** Figure S1

[Figure]

**Fig. 4.** Figure S3

**Observed: El Niño Events (a)**

Extreme = 2 ± 1.4

Moderate = 1 ± 0.9

Weak = 7 ± 2.4

Total = 10 ± 2.7

Other

1982

1997

2015

**piControl: El Niño Events (b)**

Extreme = 62 ± 7.6

Moderate = 88 ± 9.0

Weak = 150 ± 11.3

Total = 300 ± 14.6

Other

**4×CO2: El Niño Events (c)**

Extreme = 635 ± 15.0 (+924%)*

Moderate = 304 ± 14.5 (+245)*

Weak = 0

Total = 939 ± 6.8 (+213%)*

Other

**G1: El Niño Events (d)**

Extreme = 100 ± 9.4 (+61%)*

Moderate = 76 ± 8.4 (−14%)

Weak = 161 ± 11.6 (+7%)

Total = 337 ± 14.8 (+12%)**

Other

**Fig. 5.** Figure S5

**Observed: El Niño Events (a)**

Extreme = 2 ± 1.4
Moderate = 1 ± 0.9
Weak = 6 ± 2.2
Total = 9 ± 2.6
Other

1982
1997
2015

**piControl: El Niño Events (b)**

Extreme = 62 ± 7.6
Moderate = 74 ± 8.3
Weak = 163 ± 11.6
Total = 299 ± 11.6
Other

**4×CO2: El Niño Events (c)**

Extreme = 635 ± 15.0 (+924%)*
Moderate = 317 ± 14.7 (+328)*
Weak = 0
Total = 952 ± 5.9 (+218%)*
Other

**G1: El Niño Events (d)**

Extreme = 100 ± 9.4 (+61%)*
Moderate = 74 ± 8.3
Weak = 165 ± 8.0 (+1.2%)
Total = 339 ± 14.9 (+13%)**
Other

**Fig. 6.** Figure S6

[Figure]

**Fig. 7.** Figure S8

[Figure]

**Fig. 8.** Figure S9

[Figure]

**Fig. 9.** Figure S10

**Table S2.** Meridional SST Gradient (MSSTG)

| Experiment | Mean (ºC) | Difference w.r.t. piControl (ºC) | Std. Dev. 10,000 Realizations (ºC) | ~ Change w.r.t. piControl (%) |
|---|---|---|---|---|
| piControl | 1.38* | | 0.0265 | |
| 4×CO₂ | -0.15* | -1.53 | | -111* |
| G1 | 1.25* | -0.13 | | -9* |

Key: *99 % cl; **95 % cl

**Fig. 10.** Table S2

**Table S3.** Total number of El Niño events (SST > 0.5 s.d.)

| Experiment | No. of Events | Difference w.r.t. piControl | Std. Dev. 10,000 Realizations | ~ Change w.r.t. piControl (%) |
|---|---|---|---|---|
| piControl | 300 [300] | | 14.6 [14.6] | |
| 4×$CO_2$ | 161 [565] | 139 [265] | | -46* [+88*] |
| G1 | 337 [337] | 37 [37] | | +12** [+12**] |

Key: Niño3 [E-Index]; *99 % cl; **95 % cl

**Fig. 11.** Table S3

**Table S5.** Mean DJF Heat Flux (hf) Feedback

| Experiment | hf feedback or Damping Coefficient (Wm$^{-2}$/°C) | Difference w.r.t. piControl (Wm$^{-2}$/°C) | Std. Dev. 10,000 Realizations (Wm$^{-2}$/°C) | ~ Change w.r.t. piControl (%) |
|---|---|---|---|---|
| ERA5 | -14.59 | | | |
| piControl | -14.70 | | 0.52 | |
| 4×CO$_2$ | -21.90 | +7.19 | | +48* |
| G1 | -14.85 | +0.15 | | +1.0 |

*99% cl; **95% cl; Calculation period: ERA5 (41-yrs); HadCM3L (990-yrs)

**Fig. 12.** Table S5

**Table S6.** Mean DJF Bjerknes (BJ) Feedback

| Experiment | BJ feedback ($10^{-2}$ Nm$^{-2}$/°C) | Difference w.r.t. piControl ($10^{-2}$ Nm$^{-2}$/°C) | Std. Dev. 10,000 Realizations (Wm$^{-2}$/°C) | ~ Change w.r.t. piControl (%) |
|---|---|---|---|---|
| ERA5 | 3.3 | | | |
| piControl | 3.3 | | 0.0091 | |
| 4×CO$_2$ | 2.2 | -1.1 | | -33* |
| G1 | 3.5 | +0.2 | | +6* |

*99% cl; **95% cl; Calculation period: ERA5 (41-yrs); HadCM3L (990-yrs)

**Fig. 13.** Table S6

**Table S7.** Mean DJF Ocean Stratification

| Experiment | Stratification (ºC) | Difference w.r.t. piControl (ºC) | Std. Dev. 10,000 Realizations (ºC) | ~ Change w.r.t. piControl (%) |
|---|---|---|---|---|
| piControl | 2.28* | | 0.0331 | |
| 4×CO₂ | 5.06* | +2.78 | | +122* |
| G1 | 2.37* | +0.09 | | +4** |

*99% cl; **95% cl

**Fig. 14.** Table S7

**Supplement:**

**Figure Captions**

**Figure 7.** *Relationship between MSSTG and Niño3 rainfall for (a) observations (b) piControl (c) 4×CO$_2$, and (d) G1. A solid black horizontal line indicates a threshold value of 5 mm day$^{-1}$. See text for the definition of extreme, moderate, and total El Niño events. A single (double) asterisk indicates that the change in frequency, relative to piControl, is statistically significant at 99 % (95 %) cl. Numbers with a ± symbol indicate s.d. calculated with 10,000 bootstrap realizations. Following Cai et al. (2014), a non-ENSO related trend has been removed from the rainfall time series.*

**Figure 15.** *BJ feedback (μ; 10$^{-2}$ Nm$^{-2}$/$^{\circ}$C) for (a) piControl (b) 4×CO$_2$, and (c) G1. The value with ± sign indicates s.d. of μ after 10,000 bootstrap realizations. An asterisk indicates statistical significance at 99 % cl. Mean change in ocean temperature, (d) 4×CO$_2$-piControl, and (e) G1-piControl. The black box shows the area averaging region for upper ocean temperature, and the black line shows the lower layer used for calculation of stratification as a difference of upper and lower layer. Stipples indicate grid points with statistical significance at 99 % cl using a non-parametric Wilcoxon rank-sum test.*

**Figure S1.** *ENSO diversity and nonlinear relationship between PCs. First monthly principal pattern, EOF1, for (a) ERA5 and (b, c) piControl. Second monthly principal pattern, EOF2, for (d) ERA5 and (e, f) piControl. DJF EP pattern for (g) ERA5 and (h, i) piControl. DJF CP pattern for (j) ERA5 and (k, l) piControl. The nonlinear relationship between PC1 and PC2 for (m) ERA5 and (n, o) piControl. The blue box indicates the Niño3 (Niño4) region in a-c, and g-I (d-f and j-l). The left and the middle panel shows EOF analysis over the 41 years of ER5 (1979-2019) and piControl. The right panel shows EOF analysis over 990-year of piControl.*

**Figure S3.** *Histogram of MSSTG for piControl, 4×CO$_2$, and G1 for all samples (a) and for extreme El Niño events. The values are plotted at the centre of each bin with an interval of 0.5 $^{\circ}$C. Blue, red, and green vertical lines indicate climatological mean values of MSSTG under piControl (1.38 $^{\circ}$C), 4×CO$_2$ (-0.15 $^{\circ}$C), and G1 (1.25 $^{\circ}$C), respectively. H = 1 indicates that the shift in the mean is statistically significant at 99 % cl using a non-parametric Wilcoxon rank-sum test.*

**Figure S5.** *Relationship between MSSTG and quadratically detrended Niño3 rainfall for (a) observations (b) piControl (c) 4×CO$_2$, and (d) G1. The solid black horizontal line indicates a threshold of 5 mm day$^{-1}$. A single (double) asterisk indicates that the change in frequency, relative to piControl, is statistically significant at 99 % (95 %) cl. Numbers with a ± symbol indicate s.d. calculated with 10,000 bootstrap realizations. Following Cai et al. (2014), a non-ENSO related trend has been removed from the rainfall time series. Events are classified as: Extreme (Niño3 rainfall > 5 mm day$^{-1}$ and MSSTG < 0), moderate (Niño3 rainfall > 5 mm day$^{-1}$ and MSSTG > 0), weak (Standardized Niño3 SSTs > 0.5 $^{\circ}$C and Niño3 rainfall < 5 mm day$^{-1}$), total is sum of extreme, moderate, and weak events.*

**Figure S6.** *Relationship between MSSTG and linearly detrended Niño3 rainfall for (a) observations (b) piControl (c) 4×CO$_2$, and (d) G1. The solid black horizontal line indicates a threshold of 5 mm day$^{-1}$. A single (double) asterisk indicates that the change in frequency, relative to piControl, is statistically significant at 99 % (95 %) cl. Numbers with a ± symbol indicate s.d. calculated with 10,000 bootstrap realizations. Following Cai et al. (2014), a non-ENSO related trend has been removed from the rainfall time series. Events are classified as: Extreme (Niño3 rainfall > 5 mm day$^{-1}$ and MSSTG < 0), moderate (Niño3 rainfall > 5 mm day$^{-1}$ and MSSTG > 0), weak (Standardized Niño3 SSTs > 0.5 $^{\circ}$C and Niño3 rainfall < 5 mm day$^{-1}$), total is sum of extreme, moderate, and weak events.*

**Figure S8.** *Histogram of Niño3 rainfall for piControl, 4×CO$_2$, and G1. The values are plotted at the centre of each bin with an interval of 1 mm day$^{-1}$. Blue, red, and green vertical lines indicate climatological mean values of Niño3 rainfall under piControl (2.9 mm day$^{-1}$), 4×CO$_2$ (9.8 mm day$^{-1}$), and G1 (3.2 mm day$^{-1}$), respectively. H = 1 indicates that the shift in the mean is statistically significant at 99 (95) % cl for 4×CO$_2$ (G1) using the non-parametric Wilcoxon rank-sum test. The grey vertical line show threshold of 5 mm day$^{-1}$.*

***Figure S9.*** *Histogram of ZSSTG anomalies for (a) all samples, (b) extreme El Niño events only, and (c) extreme La Niña events only. The values are plotted at the centre of each bin with an interval of 0.5 $^{o}$C. In a blue, red, and green solid vertical lines indicate climatological median ZSSTG under piControl (0.07 $^{o}$C), $4 \times CO_2$ (-1.54 $^{o}$C), and G1 (-0.28 $^{o}$C), respectively, for all samples. In b, blue, red, and green dashed vertical lines indicate climatological median ZSSTG under piControl (-1.83 $^{o}$C), $4 \times CO_2$ (-1.71 $^{o}$C), and G1 (-1.96 $^{o}$C), respectively, for extreme El Niño events. In c, blue, and green dashed vertical lines indicate climatological median ZSSTG under piControl (1.37 $^{o}$C) and G1 (1.52 $^{o}$C), respectively, for extreme La Niña events. H = 1 indicates that using a non-parametric Wilcoxon rank-sum test, the shift in the median is statistically significant at 99 (95) % cl in a (b). H = 0 means that the shift in the median is not statistically significant. The ZSSTG is defined as the difference between SST in the Maritime continent (5$^{o}$ N-5$^{o}$ S; 100$^{o}$ E-126$^{o}$ E) and eastern equatorial Pacific (Niño3 region: 5$^{o}$ N-5$^{o}$ S, 150$^{o}$ W-90$^{o}$W). The anomalies are calculated relative to piControl.*

***Figure S10.*** *Histogram of ZSSTG anomalies for (a) all samples and (b) extreme La Niña events only. The values are plotted at the centre of each bin with an interval of 0.5 $^{o}$C. Blue, red, and solid green lines indicate climatological median ZSSTG under piControl (-0.14 $^{o}$C), $4 \times CO_2$ (-1.37 $^{o}$C), and G1 (-0.40 $^{o}$C), respectively, for all samples. Blue, red, and green dash-dotted lines indicate climatological median ZSSTG under piControl (0.84 $^{o}$C), $4 \times CO_2$ (-0.03 $^{o}$C), and G1 (0.72 $^{o}$C), respectively, for all La Niña events. In b, blue, red, and green dashed lines indicate climatological median ZSSTG under piControl (1.52 $^{o}$C) and G1 (3.35 $^{o}$C), respectively, for extreme La Niña events. H = 1 indicates that the shift in the median is statistically significant at 99 % cl using the non-parametric Wilcoxon rank-sum test. The ZSSTG is defined as the difference between SST in the Maritime continent (5$^{o}$ N-5$^{o}$ S; 100$^{o}$ E-126$^{o}$ E) and central equatorial Pacific (Niño4 region: 5$^{o}$ N-5$^{o}$ S, 160$^{o}$ E-150$^{o}$ W) (Cai et al., 2015). The anomalies are calculated relative to piControl.*

---

## Author Response (AR1)

[revised manuscript text omitted]

4      manuscript. Below, we reply point-by-point, highlighting the changes we have implemented. The primary
5      concern of the referees was the evaluation of the climate model capability to simulate ENSO variability, and the
6      lack of detailed explanations on possible mechanisms responsible for changes in ENSO both under $4\times CO_2$ and
7      solar geoengineering (G1). In the revised manuscript, we therefore put a strong emphasis on model evaluation
8      and are able to confirm the necessary model skill (section 2.4). We also provide an entirely new section (section
9      4) on possible mechanisms behind the changes in ENSO extremes and ENSO amplitudes.

**Referee #1**

**Major Points**

**1)**

**It is not clear exactly why the modeled ENSO changed from 4xCO2 to G1 in this model? Is it because of the air-sea heat fluxes act more less as a damping in the eastern equatorial Pacific associated with the mean state change in G1? More interestingly, why G1 does not recover many of the climatic states of piControl? Initial thought would be the ocean state never fully recovers. But as stated in the paper the change in thermocline depth is not statistically different between G1 and piControl. I don't think I came across a plot of subsurface temperature, e.g., depth-longitude differences between 4xCO2 and G1 vs piControl. Perhaps while the thermocline depth statistics do not change, there are still changes in the subsurface ocean temperatures in certain areas.**

In the revised manuscript, we have calculated ENSO feedbacks, Bjerknes and heat flux, and ocean stratification to explain the mechanisms for change in ENSO. We have added Section 4 elaborating on the mechanism for change in ENSO under both 4xCO$_2$ and G1. (See section 4, from page 17 and line 1 to page 18 and line 29). Specifically we write:

[revised manuscript text omitted]

34   *$^o$C), respectively. H = 1 indicates that the shift in the mean is statistically significant at 99 % cl using a non-*
35   *parametric Wilcoxon rank-sum test.*

[Figure]

**Figure S9.** *Histogram of ZSSTG anomalies for (a) all samples, (b) extreme El Niño events only, and (c) extreme La Niña events only. The values are plotted at the centre of each bin with an interval of 0.5 $^o$C. In a blue, red, and green solid vertical lines indicate climatological median ZSSTG under piControl (0.07 $^o$C), 4×$CO_2$ (-1.54 $^o$C), and G1 (-0.28 $^o$C), respectively, for all samples. In b, blue, red, and green dashed vertical lines indicate climatological median ZSSTG under piControl (-1.83 $^o$C), 4×$CO_2$ (-1.71 $^o$C), and G1 (-1.96 $^o$C), respectively, for extreme El Niño events. In c, blue, and green dashed vertical lines indicate climatological median ZSSTG under piControl (1.37 $^o$C) and G1 (1.52 $^o$C), respectively, for extreme La Niña events. H = 1 indicates that using a non-parametric Wilcoxon rank-sum test, the shift in the median is statistically significant at 99 (95) % cl in a (b). H = 0 means that the shift in the median is not statistically significant. The ZSSTG is defined as the difference between SST in the Maritime continent (5$^o$ N-5$^o$ S; 100$^o$ E-126$^o$ E) and eastern equatorial Pacific (Niño3 region: 5$^o$ N-5$^o$ S, 150$^o$ W-90$^o$W). The anomalies are calculated relative to piControl.*

[Figure]

**Figure S10.** *Histogram of ZSSTG anomalies for (a) all samples and (b) extreme La Niña events only. The values are plotted at the centre of each bin with an interval of 0.5 °C. Blue, red, and solid green lines indicate climatological median ZSSTG under piControl (-0.14 °C), 4×CO₂ (-1.37 °C), and G1 (-0.40 °C), respectively, for all samples. Blue, red, and green dash-dotted lines indicate climatological median ZSSTG under piControl (0.84 °C), 4×CO₂ (-0.03 °C), and G1 (0.72 °C), respectively, for all La Niña events. In b, blue, red, and green dashed lines indicate climatological median ZSSTG under piControl (1.52 °C) and G1 (3.35 °C), respectively, for extreme La Niña events. H = 1 indicates that the shift in the median is statistically significant at 99 % cl using the non-parametric Wilcoxon rank-sum test. The ZSSTG is defined as the difference between SST in the Maritime continent (5° N-5° S; 100° E-126° E) and central equatorial Pacific (Niño4 region: 5° N-5° S, 160° E-150° W) (Cai et al., 2015). The anomalies are calculated relative to piControl.*

**Table S5.** Mean DJF Heat Flux (hf) Feedback

| Experiment | hf feedback or Damping Coefficient (Wm⁻²/°C) | Difference w.r.t. piControl (Wm⁻²/°C) | Std. Dev. 10,000 Realizations (Wm⁻²/°C) | ~ Change w.r.t. piControl (%) |
|---|---|---|---|---|
| ERA5 | -14.59 | | | |
| piControl | -14.70 | | 0.52 | |
| 4×CO₂ | -21.90 | +7.19 | | +48* |
| G1 | -14.85 | +0.15 | | +1.0 |

*99% cl; **95% cl; Calculation period: ERA5 (41-yrs); HadCM3L (990-yrs)

**Table S6.** Mean DJF Bjerknes (BJ) Feedback

| Experiment | BJ feedback (10⁻² Nm⁻²/°C) | Difference w.r.t. piControl (10⁻² Nm⁻²/°C) | Std. Dev. 10,000 Realizations (Wm⁻²/°C) | ~ Change w.r.t. piControl (%) |
|---|---|---|---|---|
| ERA5 | 3.3 | | | |
| piControl | 3.3 | | 0.0091 | |
| 4×CO₂ | 2.2 | -1.1 | | -33* |
| G1 | 3.5 | +0.2 | | +6* |

*99% cl; **95% cl; Calculation period: ERA5 (41-yrs); HadCM3L (990-yrs)

**Table S7.** Mean DJF Ocean Stratification

| Experiment | Stratification (°C) | Difference w.r.t. piControl (°C) | Std. Dev. 10,000 Realizations (°C) | ~ Change w.r.t. piControl (%) |
|---|---|---|---|---|
| piControl | 2.28* | | 0.0331 | |
| 4×CO₂ | 5.06* | +2.78 | | +122* |
| G1 | 2.37* | +0.09 | | +4** |

*99% cl; **95% cl

[Figure]

*Figure 15. BJ feedback (μ; $10^{-2}$ $Nm^{-2}/°C$) for (a) piControl (b) 4×$CO_2$, and (c) G1. The value with ± sign indicates s.d. of μ after 10,000 bootstrap realizations. An asterisk indicates statistical significance at 99 % cl. Mean change in ocean temperature, (d) 4×$CO_2$-piControl, and (e) G1-piControl. The black box shows the area averaging region for upper ocean temperature, and the black line shows the lower layer used for calculation of stratification as a difference of upper and lower layer. Stipples indicate grid points with statistical significance at 99 % cl using a non-parametric Wilcoxon rank-sum test.*

In the Discussion and conclusion (section 5, page 19, lines 1-14), we have added the following paragraph:

*To conclude, solar geoengineering can compensate many of the GHG-induced changes in the tropical Pacific, but, importantly, not all of them. In particular, controlling the downward shortwave flux cannot correct one of the climate system's most dominant modes of variability, i.e., ENSO, wholly back to preindustrial conditions. The ENSO feedbacks (Bjerkness and heat flux) and more stratified ocean temperatures may induce ENSO to behave differently under G1 than under piControl and 4×$CO_2$. Different meridional distributions of shortwave and longwave forcings (e.g., Nowack et al., 2016) resulting in the surface ocean overcooling, and residual warming of the deep ocean are the plausible reasons for the solar geoengineered climate not reverting entirely to the preindustrial state. However, we note that this is a single model study, and more studies are needed to show the robustness and model-dependence of any results discussed here, e.g. using long-term multimodel ensembles from GeoMIP6 (Kravitz et al., 2015), once the data are released. The long-term Stratospheric Aerosol Geoengineering Large Ensemble (GLENS; Tilmes et al., 2018) data can also be explored to investigate ENSO variability under geoengineering.*

**2)**

**Nonetheless this leads to the question: How large are the differences in mean state and ENSO statistics between G1 and piControl state in comparison to the internal variability in piControl? For example P9, L20-21: the reduction in MSSTG is 9% in G1, is this substantial compared internal variability in piControl and to that seen during an El Nino?**

We have shown that the 9 % change in MSSTG under G1 is statistically significant (99 % confidence level) relative to piControl using both Bootstrap resampling and a non-parametric Wilcoxon rank-sum test. The increase in the frequency of extreme El Niño events is due to more frequent reversals of MSSTG (Fig. S3 and Table S2). In the revised manuscript, we have tested the change in frequency under both $4\times CO_2$ and G1, relative to piControl, first by using rainfall > 5 mm day$^{-1}$ as a threshold for extreme El Niño events and then selecting only those events for which rainfall > 5 mm day$^{-1}$ and MSSTG < 0. Both methods show a statistically significant increase in extreme El Niño events. Choosing extreme events having MSSTG < 0 assures that strong convection has established over the Niño3 region during the extreme. Further, we have shown the histograms of MSSTG for all samples and exclusively for extreme El Niño events, which indicate more frequent reversals of MSSTG both under $4\times CO_2$ and G1 relative to piControl. In the revised manuscript, we have incorporated the following changes:

*Overall there is a change in sign and reduction of MSSTG in $4\times CO_2$ (~-111 %, 99 % cl) and only decrease in G1 (~-9 %, 99 % cl) (Fig. S3, and Table S2).* (Section 3.1.4, page11, lines 17-19)

*A threshold of detrended Niño3 total rainfall of 5 mm day$^{-1}$ recognizes events as extremes even when the MSSTG is positive and stronger, especially under $4\times CO_2$, which plausibly means that ITCZ might not shift over the equator for strong convection to occur during such extremes. The El Niño event of 2015 is a typical example of such events. We test our results with a more strict criterion by choosing only those events as extremes, which have characteristics similar to that of 1982 and 1997 El Niño events (i.e., Niño3 rainfall > 5 mm day$^{-1}$ and MSSTG < 0). We declare events having characteristics similar to that of the 2015 event as moderate El Niño events (Fig. S5). Based on this method, we find a robust increase in the number of extreme El Niño events both in $4\times CO_2$ (924 %) and G1 (61 %) at 99 % cl.* (Section 3.2.2, page14, lines 26-34)

**Table S2.** Meridional SST Gradient (MSSTG)

| Experiment | Mean (ºC) | Difference w.r.t. piControl (ºC) | Std. Dev. 10,000 Realizations (ºC) | ~ Change w.r.t. piControl (%) |
|---|---|---|---|---|
| piControl | 1.38* | | 0.0265 | |
| $4\times CO_2$ | -0.15* | -1.53 | | -111* |
| G1 | 1.25* | -0.13 | | -9* |

Key: *99 % cl; **95 % cl

[Figure]

*Figure S5. Relationship between MSSTG and quadratically detrended Niño3 rainfall for (a) observations (b) piControl (c) 4×CO$_2$, and (d) G1. The solid black horizontal line indicates a threshold of 5 mm day$^{-1}$. A single (double) asterisk indicates that the change in frequency, relative to piControl, is statistically significant at 99 % (95 %) cl. Numbers with a ± symbol indicate s.d. calculated with 10,000 bootstrap realizations. Following Cai et al. (2014), a non-ENSO related trend has been removed from the rainfall time series. Events are classified as: Extreme (Niño3 rainfall > 5 mm day$^{-1}$ and MSSTG < 0), moderate (Niño3 rainfall > 5 mm day$^{-1}$ and MSSTG > 0), weak (Standardized Niño3 SSTs > 0.5 °C and Niño3 rainfall < 5 mm day$^{-1}$), total is sum of extreme, moderate, and weak events.*

**3)**

**In many of the plots showing differences between experiments and piControl, the confidence level was set to 90%. Given the long time series of the model output, it should be increased to 95% or even 99%. This would perhaps show more regions in G1 where the differences are not significantly different from piControl.**

All statistics have been recalculated either with a 95 % or 99 % confidence level. See the manuscript with tracked changes.

**4)**

**The conclusion section could provide the reader with a little perspective on whether it is worth it to do the geoengineering solution in the context of projected increase in extreme ENSO activity. A relevant paper to help the discussion: Trenberth KE, Dai A 2007). Geophys Res Lett 34:L15702. doi: 10.1029/2007GL030524**

In the revised manuscript (section 5, page 19, lines 1-14), we have included the following paragraphs/statements:

*To conclude, solar geoengineering can compensate many of the GHG-induced changes in the tropical Pacific, but, importantly, not all of them. In particular, controlling the downward shortwave flux cannot correct one of the climate system's most dominant modes of variability, i.e., ENSO, wholly back to preindustrial conditions.*

*The ENSO feedbacks (Bjerkness and heat flux) and more stratified ocean temperatures may induce ENSO to behave differently under G1 than under piControl and $4\times CO_2$. Different meridional distributions of shortwave and longwave forcings (e.g., Nowack et al., 2016) resulting in the surface ocean overcooling, and residual warming of the deep ocean are the plausible reasons for the solar geoengineered climate not reverting entirely to the preindustrial state. However, we note that this is a single model study, and more studies are needed to show the robustness and model-dependence of any results discussed here, e.g. using long-term multimodel ensembles from GeoMIP6 (Kravitz et al., 2015), once the data are released. The long-term Stratospheric Aerosol Geoengineering Large Ensemble (GLENS; Tilmes et al., 2018) data can also be explored to investigate ENSO variability under geoengineering.*

**5)**

**P11, L36: Picking a result on one model sounds rather odd as we know that the change in ENSO amplitude varies widely across models (e.g., Collins et al. 2010). In a recent study by Cai et al. (2018, Nature, https://www.nature.com/articles/s41586-018-0776-9), however, there seems to be a stronger inter-model agreement on the increase in ENSO amplitude in models that are able to simulate ENSO flavors (see their Extended Data Fig. 8b), as implied in the PC1-PC2 space. So does the HadCM3L model capture the nonlinear relationship between PC1 and PC2 as observed? Here PC1 and PC2 refer to the first and second eigenmodes of tropical Pacific SST (see their Fig. 1). Also, it is relevant to discuss the results of Cai et al. (2018) in 1st paragraph of Page 3.**

Regarding the change in amplitude, we refer to other studies in the revised manuscript and include the following paragraphs/statements:

*Previous studies found that climate models produced mixed responses (both increases and decreases in amplitude) in terms of how ENSO amplitude change with global warming (see Latif et al. 2009; Collins et al. 2010; Vega-Westhoff and Sriver 2017). However, Cai et al. (2018) found an intermodel consensus, for models capable of reproducing ENSO diversity, for strengthening of ENSO amplitude under A2, RCP4.5, and RPC8.5 transient scenarios.* (see section 3.2.1, page 13, lines 6-11)

We have included a separate section (2.4) under the title "ENSO representation in HadCM3L" which discusses the HadCM3L capability to simulate ENSO diversity as described by Cai et al. (2018). We have incorporated the following paragraphs/statements in the revised manuscript:

*Before employing HadCM3L for studying ENSO variability under $4\times CO_2$, and G1, we evaluate its piControl simulation against present-day observational data.* (see section 2.4, page 6, lines 40-41)

Further, we have included the following paragraphs (section 2.4, page 7, and line 14 to next page line 21):

[revised manuscript text omitted]

See Supplementary Fig. S1 and Tables S5-S6

[Figure]

**Figure S1.** *ENSO diversity and nonlinear relationship between PCs. First monthly principal pattern, EOF1, for (a) ERA5 and (b, c) piControl. Second monthly principal pattern, EOF2, for (d) ERA5 and (e, f) piControl. DJF EP pattern for (g) ERA5 and (h, i) piControl. DJF CP pattern for (j) ERA5 and (k, l) piControl. The nonlinear relationship between PC1 and PC2 for (m) ERA5 and (n, o) piControl. The blue box indicates the Niño3 (Niño4) region in a-c, and g-I (d-f and j-l). The left and the middle panel shows EOF analysis over the 41 years of ER5 (1979-2019) and piControl. The right panel shows EOF analysis over 990-year of piControl.*

**Table S5.** Mean DJF Heat Flux (hf) Feedback

| Experiment | hf feedback or Damping Coefficient (Wm$^{-2}$/°C) | Difference w.r.t. piControl (Wm$^{-2}$/°C) | Std. Dev. 10,000 Realizations (Wm$^{-2}$/°C) | ~ Change w.r.t. piControl (%) |
|---|---|---|---|---|
| ERA5 | -14.59 | | | |
| piControl | -14.70 | | 0.52 | |
| 4×CO$_2$ | -21.90 | +7.19 | | +48* |
| G1 | -14.85 | +0.15 | | +1.0 |

*99% cl; **95% cl; Calculation period: ERA5 (41-yrs); HadCM3L (990-yrs)

**Table S6.** Mean DJF Bjerknes (BJ) Feedback

| Experiment | BJ feedback (10$^{-2}$ Nm$^{-2}$/°C) | Difference w.r.t. piControl (10$^{-2}$ Nm$^{-2}$/°C) | Std. Dev. 10,000 Realizations (Wm$^{-2}$/°C) | ~ Change w.r.t. piControl (%) |
|---|---|---|---|---|
| ERA5 | 3.3 | | | |
| piControl | 3.3 | | 0.0091 | |
| 4×CO$_2$ | 2.2 | -1.1 | | -33* |
| G1 | 3.5 | +0.2 | | +6* |

*99% cl; **95% cl; Calculation period: ERA5 (41-yrs); HadCM3L (990-yrs)

**6)**

**P7, L10: make clear the results are in *qualitative* agreement with previous studies. Not all of the cited studies are based on 4xCO2.**

We check our results and categorically mention that our results qualitatively agree with previous studies. Thus we add the following change:

*Our SST results under $4xCO_2$ qualitatively agree with previous studies (Liu et al., 2005; van Oldenborgh et al., 2005; Collins et al., 2010; Vecchi and Wittenberg et al., 2010; Cai et al., 2015a; Huang and Ying et al., 2015; Luo et al., 2015; Kohyama et al., 2017; Nowack et al., 2017).* (see section 3.1.1, page 9, line 9-12)

**7)**

**P.7, L13: some studies argue against the use of "El Nino-like" term in describing the mean-state change under greenhouse forcing (e.g., Collins et al. 2010; see also Xie et al. 2010 https://journals.ametsoc.org/doi/10.1175/2009JCLI3329.1). Cautionary is needed to avoid confusions. A relevant reference on the mean state change: diNezio et al https://journals.ametsoc.org/doi/full/10.1175/2009JCLI2982.1.**

We have deleted the term "El Nino-like" from the revised manuscript and have replaced it with appropriate words like *"a significant mean warming"* or *"a warming state"* (see section 1, page 3, lines 18-19; and section 3.1.1 page 8, line 37)

**8)**

**Fig. 2d, e: title of the figure states +0.21 mm/day, -0.23 mm/day. Please explain in the caption that those numbers correspond to the area average difference between experiment and piControl in the tropical Pacific (state domain).**

The following change is made in the caption of Fig. 1:

*The numbers in a-c represent a mean temperature in the corresponding simulation, and numbers in d-e represent an area-averaged difference of piControl with $4 \times CO_2$ and G1, respectively, in the tropical Pacific region ($25^o$ N-$25^o$ S; $90^o$ E-$60^o$ W).* (page 28, lines 8-11)

The following change is made in the caption of Fig. 2:

*The numbers in a-c represent mean rainfall in the corresponding simulation, and numbers in d-e represent an area-averaged difference of piControl with $4 \times CO_2$ and G1, respectively, in the tropical Pacific region ($25^o$ N-$25^o$ S; $90^o$ E-$60^o$ W).* (page 29, lines 7-10)

**9)**

**P9, L22-24: This sentence needs a rework. Avoid the word "observe" on model analysis (models are not observations). I think Wang et al. (2017) was referring to zonal temperature gradient between the maritime continent and central Pacific, not eastern Pacific. The difference is not significant in RCP2.6, but should be significant in RCP8.5 (Cai et al. 2015, Nature Climate Change on extreme La Nina).**

The use of word "observed" for modelled data has been replaced with appropriate words in the revised manuscript. The reference of Wang et al. (2017) for weakening of ZSSTG has also been removed from the revised manuscript. Instead we add the following statements:

*Our results under $4xCO_2$ are in agreement with Coats and Karnauskas (2017), who using several climate models found a weakening of the ZSSTG under the RCP8.5 scenario.* (see section 3.1.4, page 11, line 11-13)

*The weakening of the MSSTG is qualitatively in agreement with previous studies under increased GHG forcings (e.g., Cai et al., 2014; Wang et al., 2017).* (see section 3.1.4, page 11, lines 21-22)

**10)**

**Fig. 7: Please indicate clearly in the caption that the timeseries have been detrended with non ENSO related trend removed following Cai et al. (2017). Otherwise it would create confusion as other studies show that the 2015/16 Nino3 rainfall is close to the 5 mm/day threshold and is thus classified as an extreme El Nino (Santoso et al. 2017). In panel c, d, it must be rainfall anomalies that are shown because there are negative rainfall values, so wouldn't the 4 or 3 mm/day threshold be applied here? Panel a and b also have negative rainfall values. Please double check.**

In the captions, we have added the following text:

*Following Cai et al. (2014), a non-ENSO related trend has been removed from the rainfall time series.* (see Fig. 7, page 32, lines 8-9; and Fig.S5-S6)

In Fig. 7 and Fig. S5-6, revised manuscript, we have shown total rainfall after subtracting the non-ENSO related trend as described by Cai et al. (2017). In the previous manuscript, we subtracted the non-ENSO related trend, including the intercept term; therefore, negative values were present, and it's been corrected now.

**11)**

**P12, L28-31: under 4xCO2 the rainfall skewness is dramatically reduced. Does that mean there are less extreme El Nino based on the rainfall definition? If so, this does not seem consistent with the PPE results of Cai et al. (2014) using the same model.**

In the revised manuscript, we have included the analysis for $4\times CO_2$. We show that extreme El Niño events increase under $4\times CO_2$ using metrics based on rainfall and E-index (See section 3.2.2). The climate regime under $4\times CO_2$ is substantially different from that of piControl (See Fig. S8). The comparison of piControl and $4\times CO_2$ is not simple as mean rainfall, despite zero skewness, significantly shifts to a higher value (9.8 mm day$^{-1}$) under $4\times CO_2$. We have added the following text in the revised manuscript:

[revised manuscript text omitted]

**Figure S5.** *Relationship between MSSTG and quadratically detrended Niño3 rainfall for (a) observations (b) piControl (c) 4×CO$_2$, and (d) G1. The solid black horizontal line indicates a threshold of 5 mm day$^{-1}$. A single (double) asterisk indicates that the change in frequency, relative to piControl, is statistically significant at 99 % (95 %) cl. Numbers with a ± symbol indicate s.d. calculated with 10,000 bootstrap realizations. Following Cai et al. (2014), a non-ENSO related trend has been removed from the rainfall time series. Events are classified as: Extreme (Niño3 rainfall > 5 mm day$^{-1}$ and MSSTG < 0), moderate (Niño3 rainfall > 5 mm day$^{-1}$ and MSSTG > 0), weak (Standardized Niño3 SSTs > 0.5 $^o$C and Niño3 rainfall < 5 mm day$^{-1}$), total is sum of extreme, moderate, and weak events.*

[Figure]

**Figure S6.** *Relationship between MSSTG and linearly detrended Niño3 rainfall for (a) observations (b) piControl (c) 4×CO₂, and (d) G1. The solid black horizontal line indicates a threshold of 5 mm day⁻¹. A single (double) asterisk indicates that the change in frequency, relative to piControl, is statistically significant at 99 % (95 %) cl. Numbers with a ± symbol indicate s.d. calculated with 10,000 bootstrap realizations. Following Cai et al. (2014), a non-ENSO related trend has been removed from the rainfall time series. Events are classified as: Extreme (Niño3 rainfall > 5 mm day⁻¹ and MSSTG < 0), moderate (Niño3 rainfall > 5 mm day⁻¹ and MSSTG > 0), weak (Standardized Niño3 SSTs > 0.5 °C and Niño3 rainfall < 5 mm day⁻¹), total is sum of extreme, moderate, and weak events.*

[Figure]

**Figure S8.** *Histogram of Niño3 rainfall for piControl, 4×CO₂, and G1. The values are plotted at the centre of each bin with an interval of 1 mm day⁻¹. Blue, red, and green vertical lines indicate climatological mean values of Niño3 rainfall under piControl (2.9 mm day⁻¹), 4×CO₂ (9.8 mm day⁻¹), and G1 (3.2 mm day⁻¹), respectively. H = 1 indicates that the shift in the mean is statistically significant at 99 (95) % cl for 4×CO₂ (G1) using the non-parametric Wilcoxon rank-sum test. The grey vertical line show threshold of 5 mm day⁻¹.*

**Table S3.** Total number of El Niño events (SST > 0.5 s.d.)

| Experiment | No. of Events | Difference w.r.t. piControl | Std. Dev. 10,000 Realizations | ~ Change w.r.t. piControl (%) |
|---|---|---|---|---|
| piControl | 300 [300] | | 14.6 [14.6] | |
| 4×CO₂ | 161 [565] | 139 [265] | | -46* [+88*] |
| G1 | 337 [337] | 37 [37] | | +12** [+12**] |

Key: Niño3 [E-Index]; *99 % cl; **95 % cl

**12)**

**P13, L28-39: The characterization of extreme La Nina is based on Nino4 (Cai et al. 2015), so it is not clear how Nino3 and Nino3.4 indices are used here to infer changes in extreme La Nina.**

We have deleted inferences based on Nino3 and Nino3.4 in section 3.2.3 of the revised manuscript.

**Figure presentation**

**13)**

**Fig. 1e, some areas look white (e.g., eastern equatorial Pacific which is supposed to be approx. -0.2C p7, L9) while the colorbar does not have white on it.**

We have reproduced Fig. 1e with a different color bar, and visibility of colors has improved in the revised manuscript.

**14)**

**Figure 10: the color limit does not seem correct, which shows much larger values in e, f G1-piControl than the composite anomalies themselves in panels a-d.**

We have corrected the color limits in Fig. 10.

**15)**

**The colorbar of Fig. 2, right panel especially is not ideal. It is hard to immediately see which are positive or negative without referring to the colorbar.**

In the revised manuscript, we have reproduced Fig. 2 with a diverging color bar.

**16)**

**Might be best to have the same color scale for comparing the results of 4xCO2 – piControl vs G1 – piControl. This is to convey the message the difference is much smaller for G1 – piControl than for 4xCO2.**

The differences under G1-piControl are small; if we use the same color bar for $4xCO_2$-piControl and G1-piControl, most of the information is suppressed for G1-piControl. Therefore we have used two different color bars.

**Minor points**

**17)**

**Page 4, L34: that sentence is due to Cai et al. (2014).**

We have cited Cai et al. (2014) in the revised manuscript. (see section 2.4, page 7, line 2)

**18)**

**P4, L35: delete "the northern part of" – the ITCZ is located north of equator, and that rainfall band moves equatorward during strong El Nino events.**

We have deleted "the northern part of" in the revised manuscript.

**19)**

**P5, L23: "ggradients"**

Corrected in the revised manuscript. (see section 2.3, page 5, line 24)

**20)**

**P6, L2: extreme El Ninos are not resulting in just "anomalous rainfall" but unusually large rainfall in the eastern equatorial Pacific.**

We have deleted the word anomalous and modified the text as follows:

*.... Niño3 region resulting in rainfall higher than 5mm day$^{-1}$ (Cai et al., 2014).* (see section 2.3, page 6, lines 1-2)

**21)**

**P6, L 35: "depicts this SSTasymmetry between the western and eastern equatorial Pacific well (Fig. 1a)." – not clear since the observed counterpart is not presented.**

In the text, we have cited a reference for comparing the piControl SST asymmetry with an observational dataset. We have modified the version as follows:

*The piControl simulation (Fig. 1a) reproduces the SST asymmetry between the western and eastern equatorial Pacific well (cf. Fig 1a in Vecchi and Wittenberg 2010).* (see section 3.1.1, page 8, lines 34-36)

**22)**

**P8, L10: "problem" – not clear, in what way it is a problem?**

The word "problem" has been deleted in the revised manuscript. We have modified the text as follows:

*That is, while the relative additional rainfall asymmetry between the western and eastern Pacific in 4×CO₂ is mostly resolved in G1, the tropical Pacific is overall wetter under 4×CO₂ but drier in G1.* (see section 3.1.2, page 10, lines 13-15)

**23)**

**P9, L19: repetitive: El Nino being stronger than La Nina already implies asymmetric amplitude.**

In the revised manuscript, we have modified text as follows:

*However, the use of standard deviations to define ENSO amplitude is suboptimal, because amplitudes of El Niño and La Niña events are asymmetric, i.e., in general, El Niño events are stronger than La Niña events (An and Jin 2004; Schopf and Burgman 2006; Ohba and Ueda 2009; Ham 2017).* (see section 3.2.1, page 13, lines 22-25)

**24)**

**P9, L29: the shoaling of thermocline is also due to increased stratification associated with surface intensified warming in response to greenhouse forcing.**

We have added the following text in the revised manuscript:

*In 4xCO₂, most likely the weakened easterlies (as noticed in Sect. 3.1.3; e.g., Yeh et al., 2009, Wang et al., 2017) and greater ocean temperature stratification due to increased surface warming (see Sect. 4 and Cai et al., 2018) lead to a significant shoaling of the thermocline across the western and central equatorial Pacific. In contrast, relatively little change takes place between 130ᵒ W and 90ᵒ W. In a CMIP3 multimodel (SRESA1B scenario) ensemble, Yeh et al. (2009) found a more profound deepening of the thermocline in this part of the eastern equatorial Pacific; however, for example, Nowack et al. (2017) did not find such changes under 4xCO₂ (cf. their Fig. S9). One possible explanation for this behaviour is the competing effects of upper-ocean warming (which deepens the thermocline) and the weakening of westerly zonal wind stress, causing thermocline shoaling (see Kim et al. 2011a).* (see section 3.1.5, from page 11 and line 37 to next page line 8)

**25)**

**P9, L32-36: why not use the maximum of vertical temperature gradient as a proxy of thermocline depth for all scenarios?**

In the revised manuscript, we have included a map for ocean stratification; we think it can provide some details on this. Further, model ocean vertical resolution (13 levels) is not very high to calculate maximum vertical temperature gradient.

**26)**

**P14, L6-7: for extreme El Nino events, are the PWC, SST, and rainfall anomalies strengthened as well?**

For extreme El Niño events, the PWC, SST, and rainfall anomalies are weakened. We have rectified the text as follows:

*These composites provide process-based evidence for the strengthening (weakening) of extreme La Niña (El Niño) events in G1. We show that the PWC, SST, and composite rainfall anomalies are strengthened for extreme La Niña events, while they are weakened for extreme El Niño events under G1.* (see section 3.3, page, 16, lines 5-8)

**27)**

**P14, L23-25: this must be referring to the difference between G1 and piControl. Please make that clear.**

In the revised manuscript, we have modified text as follows:

*During extreme El Niño events, in G1, we find reduced SST (Fig. 9e) and rainfall anomalies (Fig. 10e) over the eastern and western equatorial Pacific with a consistent weakening of the eastern and western branch of PWC (Fig. 11e).* (see section 3.3.1, page 16, lines 15-17)

**Referee #2**

**Major Points**

**1)**

**To study the ENSO changes under solar geoengineering, the results are all based on one single model HadCM3L. In Cai et al. (2014) and Collins et al. (2001), the model they used is HadCM3. I admit that HadCM3L and HadCM3 are identical in most aspects, but there are still differences between these two simulations. The differences should be mentioned in this study because the HadCM3L may not be skillful in reproducing ENSO variabilities, and thus the sentence in P4 L32-33 may not be completely correct. I suggest that the ENSO simulated in HadCM3L should be addressed first, regarding its magnitude and pattern. For instance, the EOF analyses can be carried out on the piControl simulations. It will help us to have a general idea of how capable the HadCM3L is in simulating the ENSO and its diversity, and what's the biases compared with observations. As pointed out in Cai et al. (2018), the magnitude and location of ENSO events are inconsistent among models. The averaged SSTA in a fixed box to measure the intensity of ENSO can be tricky. A look at the ENSO pattern in HadCM3L can also facilitate a better ENSO extreme definition, i.e. the Nino indices may not be best to define ENSO intensity. At least, a glimpse of the Figure 8 reveals that ENSO simulation is not good enough, especially the shape, maximum location and horseshoe-shaped cold SSTA in the western Pacific during El Nino events.**

We have deleted the text referring to P4 and L32-33 in the revised manuscript. In the revised manuscript we have evaluated the model skill for reproducing ENSO diversity following Cai et al. (2018). The HadCM3L belongs to the family of HadCM3 models; the only difference between HadCM3 and HadCM3L is lower ocean resolution. We have included a separate section on model evaluation. We have also mentioned the apparent biases in HadCM3L compared to observations. We find that HadCM3L has a reasonable skill to simulate ENSO and can be employed for the current study. The HadCM3L simulates the sea surface temperature maximum anomaly pattern over the Niño3 region. In the revised manuscript, we have made the following additions:

*HadCM3L stems from the family of HadCM3 climate models; the only difference is lower ocean resolution (HadCM3: $1.25^o \times 1.25^o$; Valdes et al., 2017).* (see section 2.1, page 4, lines 25-27)

*Before employing HadCM3L for studying ENSO variability under $4\times CO_2$, and G1, we evaluate its piControl simulation against present-day observational data.* (see section 2.4, page 6, lines 40-41)

Further, we have included the following paragraphs (section 2.4, page 7, and line 14 to next page line 21):

[revised manuscript text omitted]

20 *We conclude that HadCM3L has a reasonable skill for studying long-term ENSO variability and its response to*
21 *solar geoengineering. However, we also highlight the need for and hope to motivate future modelling studies*
22 *that will help identify model dependencies in the ENSO response.*

23 See Supplementary Fig. S1 and Tables S5-S6

**Table S5.** Mean DJF Heat Flux (hf) Feedback

| Experiment | hf feedback or Damping Coefficient ($Wm^{-2}/°C$) | Difference w.r.t. piControl ($Wm^{-2}/°C$) | Std. Dev. 10,000 Realizations ($Wm^{-2}/°C$) | ~ Change w.r.t. piControl (%) |
|---|---|---|---|---|
| ERA5 | -14.59 | | | |
| piControl | -14.70 | | 0.52 | |
| 4×CO₂ | -21.90 | +7.19 | | +48* |
| G1 | -14.85 | +0.15 | | +1.0 |

*99% cl; **95% cl; Calculation period: ERA5 (41-yrs); HadCM3L (990-yrs)

**Table S6.** Mean DJF Bjerknes (BJ) Feedback

| Experiment | BJ feedback ($10^{-2} Nm^{-2}/°C$) | Difference w.r.t. piControl ($10^{-2} Nm^{-2}/°C$) | Std. Dev. 10,000 Realizations ($Wm^{-2}/°C$) | ~ Change w.r.t. piControl (%) |
|---|---|---|---|---|
| ERA5 | 3.3 | | | |
| piControl | 3.3 | | 0.0091 | |
| 4×CO₂ | 2.2 | -1.1 | | -33* |
| G1 | 3.5 | +0.2 | | +6* |

*99% cl; **95% cl; Calculation period: ERA5 (41-yrs); HadCM3L (990-yrs)

[Figure]

**Figure S1.** *ENSO diversity and nonlinear relationship between PCs. First monthly principal pattern, EOF1, for (a) ERA5 and (b, c) piControl. Second monthly principal pattern, EOF2, for (d) ERA5 and (e, f) piControl. DJF EP pattern for (g) ERA5 and (h, i) piControl. DJF CP pattern for (j) ERA5 and (k, l) piControl. The nonlinear relationship between PC1 and PC2 for (m) ERA5 and (n, o) piControl. The blue box indicates the Niño3 (Niño4) region in a-c, and g-I (d-f and j-l). The left and the middle panel shows EOF analysis over the 41 years of ER5 (1979-2019) and piControl. The right panel shows EOF analysis over 990-year of piControl.*

**2)**

**The change of extreme ENSO under solar geoengineering is a major concern in this study. This paper shows adequate results to uncovering the phenomenon that may happen but lacks the investigations on underlying mechanisms. The magnitude of ENSO is mainly driven by the positive and negative feedbacks involving air-sea interactions. In the manuscript, the major atmospheric and oceanic components are depicted, such as the thermocline, zonal wind stress and zonal SST gradient. A clear physical process is needed to understand how ENSO can be modified in G1 and 4_CO2. The Bjerknes feedback, thermocline feedback and heat flux feedback can be evaluated under different scenarios. This may be helpful to illustrate why ENSO in G1 can be modified even though the thermocline, zonal SST gradient and zonal wind stress are not well separated in G1 and piControl. Also, it's necessary to go deeper into the reason why the responses of El Nino and La Nina are different for magnitude change and same for frequency change.**

In the revised manuscript, we have calculated ENSO feedbacks, Bjerknes and heat flux, and ocean stratification to explain the mechanisms for change in ENSO. We have added Section 4 elaborating on the mechanism for change in ENSO under both 4xCO₂ and G1. (See section 4, from page 17 and line 1 to page 18 and line 29). Specifically we write:

[revised manuscript text omitted]

*99% cl; **95% cl; Calculation period: ERA5 (41-yrs); HadCM3L (990-yrs)

**Table S6.** Mean DJF Bjerknes (BJ) Feedback

| Experiment | BJ feedback (10⁻² Nm⁻²/°C) | Difference w.r.t. piControl (10⁻² Nm⁻²/°C) | Std. Dev. 10,000 Realizations (Wm⁻²/°C) | ~ Change w.r.t. piControl (%) |
|---|---|---|---|---|
| ERA5 | 3.3 | | | |
| piControl | 3.3 | | 0.0091 | |
| 4×CO₂ | 2.2 | -1.1 | | -33* |
| G1 | 3.5 | +0.2 | | +6* |

*99% cl; **95% cl; Calculation period: ERA5 (41-yrs); HadCM3L (990-yrs)

**Table S7.** Mean DJF Ocean Stratification

| Experiment | Stratification (°C) | Difference w.r.t. piControl (°C) | Std. Dev. 10,000 Realizations (°C) | ~ Change w.r.t. piControl (%) |
|---|---|---|---|---|
| piControl | 2.28* | | 0.0331 | |
| 4×CO₂ | 5.06* | +2.78 | | +122* |
| G1 | 2.37* | +0.09 | | +4** |

*99% cl; **95% cl

[Figure]

*Figure 15. BJ feedback ($\mu$; $10^{-2}$ $Nm^{-2}/{}^\circ C$) for (a) piControl (b) $4\times CO_2$, and (c) G1. The value with $\pm$ sign*
*indicates s.d. of $\mu$ after 10,000 bootstrap realizations. An asterisk indicates statistical significance at 99 % cl.*
*Mean change in ocean temperature, (d) $4\times CO_2$-piControl, and (e) G1-piControl. The black box shows the area*
*averaging region for upper ocean temperature, and the black line shows the lower layer used for calculation of*
*stratification as a difference of upper and lower layer. Stipples indicate grid points with statistical significance*
*at 99 % cl using a non-parametric Wilcoxon rank-sum test.*

In the Discussion and conclusion (section 5, page 19, lines 1-14), we have added the following paragraph:

*To conclude, solar geoengineering can compensate many of the GHG-induced changes in the tropical Pacific,*
*but, importantly, not all of them. In particular, controlling the downward shortwave flux cannot correct one of*
*the climate system's most dominant modes of variability, i.e., ENSO, wholly back to preindustrial conditions.*
*The ENSO feedbacks (Bjerkness and heat flux) and more stratified ocean temperatures may induce ENSO to*
*behave differently under G1 than under piControl and $4\times CO_2$. Different meridional distributions of shortwave*
*and longwave forcings (e.g., Nowack et al., 2016) resulting in the surface ocean overcooling, and residual*
*warming of the deep ocean are the plausible reasons for the solar geoengineered climate not reverting entirely*
*to the preindustrial state. However, we note that this is a single model study, and more studies are needed to*
*show the robustness and model-dependence of any results discussed here, e.g. using long-term multimodel*
*ensembles from GeoMIP6 (Kravitz et al., 2015), once the data are released. The long-term Stratospheric*
*Aerosol Geoengineering Large Ensemble (GLENS; Tilmes et al., 2018) data can also be explored to investigate*
*ENSO variability under geoengineering.*

**3)**

**This manuscript pays a lot of efforts on how mean state of tropical Pacific might be modified under**
**4_CO2 and G1. A connection between mean state change and ENSO change is simply built by using the**
**previously proposed conclusions, i.e. the reduction of MSSTG in both 4_CO2 and G1 indicate increase**
**of extreme El Nino. However, more detailed explanations should be reviewed before applying this theory.**

1  In the revised manuscript, we have tested the change in frequency under both $4\times CO_2$ and G1, relative to
2  piControl, first by using rainfall > 5 mm day$^{-1}$ as a threshold for extreme El Niño events and then selecting only
3  those events for which rainfall > 5 mm day$^{-1}$ and MSSTG < 0. Both methods show a statistically significant
4  increase in extreme El Niño events. Choosing extreme events having MSSTG < 0 assures that strong convection
5  has established over the Niño3 region during the extreme. Further, we have shown the histograms of MSSTG
6  for all samples and exclusively for extreme El Niño events, which indicate more frequent reversals of MSSTG
7  both under $4\times CO_2$ and G1 relative to piControl. See also the discussion on mechanism now presented in Sect. 4
8  and included in a response above. In the revised manuscript, we have further incorporated the following
9  changes:
10
11  *A threshold of detrended Niño3 total rainfall of 5 mm day$^{-1}$ recognizes events as extremes even when the*
12  *MSSTG is positive and stronger, especially under $4\times CO_2$, which plausibly means that ITCZ might not shift over*
13  *the equator for strong convection to occur during such extremes. The El Niño event of 2015 is a typical example*
14  *of such events. We test our results with a more strict criterion by choosing only those events as extremes, which*
15  *have characteristics similar to that of 1982 and 1997 El Niño events (i.e., Niño3 rainfall > 5 mm day$^{-1}$ and*
16  *MSSTG < 0). We declare events having characteristics similar to that of the 2015 event as moderate El Niño*
17  *events (Fig. S5). Based on this method, we find a robust increase in the number of extreme El Niño events both*
18  *in $4\times CO_2$ (924 %) and G1 (61 %) at 99 % cl.* (Section 3.2.2, page14, lines 26-34)

[Figure]

21  ***Figure S5.*** *Relationship between MSSTG and quadratically detrended Niño3 rainfall for (a) observations (b)*
22  *piControl (c) $4\times CO_2$, and (d) G1. The solid black horizontal line indicates a threshold of 5 mm day$^{-1}$. A single*
23  *(double) asterisk indicates that the change in frequency, relative to piControl, is statistically significant at 99 %*
24  *(95 %) cl. Numbers with a ± symbol indicate s.d. calculated with 10,000 bootstrap realizations. Following Cai*
25  *et al. (2014), a non-ENSO related trend has been removed from the rainfall time series. Events are classified as:*
26  *Extreme (Niño3 rainfall > 5 mm day$^{-1}$ and MSSTG < 0), moderate (Niño3 rainfall > 5 mm day$^{-1}$ and MSSTG >*

0), weak (Standardized Niño3 SSTs > 0.5 °C and Niño3 rainfall < 5 mm day$^{-1}$), total is sum of extreme, moderate, and weak events.

[Figure]

***Figure S6.*** *Relationship between MSSTG and linearly detrended Niño3 rainfall for (a) observations (b) piControl (c) 4×CO₂, and (d) G1. The solid black horizontal line indicates a threshold of 5 mm day$^{-1}$. A single (double) asterisk indicates that the change in frequency, relative to piControl, is statistically significant at 99 % (95 %) cl. Numbers with a ± symbol indicate s.d. calculated with 10,000 bootstrap realizations. Following Cai et al. (2014), a non-ENSO related trend has been removed from the rainfall time series. Events are classified as: Extreme (Niño3 rainfall > 5 mm day$^{-1}$ and MSSTG < 0), moderate (Niño3 rainfall > 5 mm day$^{-1}$ and MSSTG > 0), weak (Standardized Niño3 SSTs > 0.5 °C and Niño3 rainfall < 5 mm day$^{-1}$), total is sum of extreme, moderate, and weak events.*

**Minor Points**

**1)**

**In P11, L24, the calculation of skewness of SST should be clarified in the context.**

In the revised manuscript we have made the following changes:

*Skewness is a measure of asymmetry around the mean of the distribution (see eq. S1). Positive skewness means that in given data distribution, the tail of the distribution is spread out towards high positive values, and vice versa (Ghandi et al., 2016).* (See section 2.4, page 7, lines 2-5)

(See Supplementary, page 13)

$$S = \left[\frac{1}{n-1}\right]\frac{\sum_i^n(X_i - \bar{X})^3}{\sigma^3} \dots\dots\dots\dots\dots\dots\dots\dots\dots\dots\dots \textit{(S1; Ghandi et al., 2016)}$$

*Where*

*S = skewness*
*n = sample size*
$X_i$ *= sample ith observation*
$\bar{X}$ *= sample mean*
$\sigma^3$ *= sample standard deviation*

**2)**

**In P9, L22-24 and P11, L36-39, the independent paragraphs seem abrupt for the context. Better to immerse in the other paragraphs.**

We have edited and merged the text with other paragraphs as follows:

*The weakening of the MSSTG is qualitatively in agreement with previous studies under increased GHG forcings (e.g., Cai et al., 2014; Wang et al., 2017).* (see section 3.1.4, page 11, lines 21-22)

*Previous studies found that climate models produced mixed responses (both increases and decreases in amplitude) in terms of how ENSO amplitude change with global warming (see Latif et al. 2009; Collins et al. 2010; Vega-Westhoff and Sriver 2017). However, Cai et al. (2018) found an intermodel consensus, for models capable of reproducing ENSO diversity, for strengthening of ENSO amplitude under A2, RCP4.5, and RPC8.5 transient scenarios.* (see section 3.2.1, page 13, lines 6-11)

**3)**

**In P12, L6-10, please clarify why quadratic trend to the time series of rainfall data should be excluded.**

In the revised manuscript, the text has been edited as follows:

*To study changes in El Niño frequency, we first need to define what constitutes an El Niño event. We here define extreme El Niño events as episodes when monthly-mean DJF Niño3 total rainfall exceeds 5 mm day$^{-1}$, following the threshold definition by Cai et al. (2014). However, as pointed out by Cai et al. (2017), trends in Niño3 rainfall are mainly driven by two factors: (1) the change in the mean state of the tropical Pacific and (2) the change in frequency of extreme El Niño events. Therefore, since we want to focus on the changes in the extremes, we need to remove contribution (1) from the raw Niño3 time series. We, therefore, fit a quadratic polynomial to the time series of rainfall data from which all extreme El Niño events (DJF total rainfall > 5 mm day$^{-1}$) have been excluded and then subtract this trend from the raw Niño3 rainfall time series. Linearly detrending the rainfall time series produces similar results.* (See section 3.2.2, page 14, lines 4-14)

**4)**

**In P13, L13-15, the central Pacific El Nino is not mentioned in the introduction. Also, the question backs to the major comment 1. The HadCM3L may not be able to capture ENSO diversity.**

We have deleted the referred text from the revised manuscript.

**5)**

**In Figure 4c, why the thermocline depth is not significantly changed over the eastern Pacific. If this is the case, is it due to the choice of 24 isotherms?**

In a CMIP3 multimodel (SRESA1B scenario) ensemble, Yeh et al. (2009) showed a deepening of the thermocline in the eastern equatorial Pacific; however, Nowack et al. (2017) did not find any change under $4xCO_2$. Both studies defined thermocline using a maximum vertical temperature gradient. Thus, we believe that no-significant-change in the eastern Pacific is not due to the choice of 24 $^{\circ}$C isotherm, but rather due to a

cancellation of competing effects on thermocline depth. In the revised manuscript, we have therefore added the following text:

*In 4xCO$_2$, most likely the weakened easterlies (as noticed in Sect. 3.1.3; e.g., Yeh et al., 2009, Wang et al., 2017) and greater ocean temperature stratification due to increased surface warming (see Sect. 4 and Cai et al., 2018) lead to a significant shoaling of the thermocline across the western and central equatorial Pacific. In contrast, relatively little change takes place between 130$^o$ W and 90$^o$ W. In a CMIP3 multimodel (SRESA1B scenario) ensemble, Yeh et al. (2009) found a more profound deepening of the thermocline in this part of the eastern equatorial Pacific; however, for example, Nowack et al. (2017) did not find such changes under 4xCO$_2$ (cf. their Fig. S9). One possible explanation for this behaviour is the competing effects of upper-ocean warming (which deepens the thermocline) and the weakening of westerly zonal wind stress, causing thermocline shoaling (see Kim et al. 2011a).* (see section 3.1.5, from page 11 and line 37 to next page line 8)

**6)**

**The significance level is 90% for differences between G1, 4_CO2 and piControl. How about 95% or even 99%? Will the significant regions be much less?**

All statistics have been recalculated either with a 95 % or 99 % confidence level. See the manuscript with track changes.

**7)**

**In P23, the height of color bars for figures can be smaller to enlarge the main part of figures. In Figure 2 d & e, symmetric colors are better to represent the negative and positive shadings.**

In the revised manuscript, all figures have been re-plotted with relatively small and diverging color bars.

**8)**

**In Figure 6 d & e, it's better to set the color bar range with the same ratio as in Figure 5 d & e.**

In the revised manuscript, both figures are re-plotted with the same color range.

---

## Referee Report (RR1)

Review Comments on the Manuscript: "Tropical Pacific Climate Variability under Solar Geoengineering: Impacts on ENSO Extremes"

The authors have addressed my questions concerning the model's performance in simulating ENSO variability and the underlying mechanisms which are responsible for ENSO changes under various climate scenarios. Overall, I am satisfied with the revised version. However, some minor modifications are needed:

1.  P3 line 3 the meaning of "until Cai et al. (2018) used SST indices based on Principal Component Analysis (PCA)." Is not clear.

2.  P3 line 13-14 the meaning of eastern and central Pacific ENSO mode should be clarified in the text somewhere (see studies in Wang et al., 2019).

3.  P3 line 18 "a significant mean warming response" might be better replaced as "a significant mean state warming response".

4.  P3 line 20 "CMIP 3" should be "CMIP3".

5.  P3 line 39 "argue" should be "argued".

6.  P3 line 42 "90 %" should be "90%".

7.  P6 lines 14-15 "BJ feedback is an equatorial zonal wind stress dynamic response to equatorial SST anomalies." might be revised as "BJ feedback is a dynamical response of equatorial zonal wind stress to equatorial SST anomalies." for clarity.

8. P14 lines 5-6, the definition of extreme events is not clear, do you mean the averaged rainfall anomalies over the Nino3 region exceeding 5 mm/day? Why 5 mm/day in Cai et al. (2014) as the threshold? This should be mentioned and clarified. Is it the same reason as Wang et al. (2020)? Thus, the first paragraph in section 3.2.2 can be better organized.

9. In section5, the possible implications of CP ENSO frequency and amplitude changes due to atmospheric and oceanic changes under $4\times CO_2$ and G1 scenarios should be discussed. The formation of EP and CP ENSO can be distinct since BJ feedback and heat flux feedback can play a relatively different role in determining the evolution of ENSO events. As inferred from the results based on $4\times CO_2$ and G1 simulations, how might the CP ENSO be changed?

---

## Author Response (AR2)

**Tropical Pacific Climate Variability under Solar Geoengineering: Impacts on ENSO Extremes**

**Abdul Malik[1,2,3], Peer J. Nowack[1,4,5,6], Joanna D. Haigh[1,4], Long Cao[7], Luqman Atique[7], Yves Plancherel[1]**

[1]Grantham Institute – Climate Change and the Environment, Imperial College London,
 London, United Kingdom
[2]Oeschger Centre for Climate Change Research, and Institute of Geography, University of
 Bern, Bern, Switzerland
[3]4700 King Abdullah University of Science and Technology, Thuwal 23955-6900, Kingdom of Saudi Arabia
[4]Department of Physics, Blackett Laboratory, Imperial College London, United Kingdom
[5]Data Science Institute, Imperial College London, United Kingdom
[6]School of Environmental Sciences, University of East Anglia, Norwich, United Kingdom
[7]School of Earth Sciences, Zhejiang University, Hangzhou, China

*Correspondence to:* Abdul Malik (abdul.malik@kaust.edu.sa)

**Point-by-Point Listing of Response to Referee Comments**

The authors thank the referees for their comments and suggestions, which have much helped us to improve our manuscript. Below, we reply point-by-point, highlighting the changes we have implemented. The response to Referee # 1 is given on pages 2-3, and for Referee # 2 on pages 4-6. The minor changes that we have made in the revised manuscript are provided on page 7.

**Referee #1**

**Minor Revisions**

**1)**

**Add \*a slight\* in this sentence: "Overall there is a change in sign and reduction of MSSTG in 4×CO2 (~-111 %, 99 % cl) and only \*a slight\* decrease in G1 (~-9 %, 99 % cl) (Fig. S3, and Table S2)." (Section 3.1.4, page11, lines 17-19).**

In the revised manuscript we have added 'a slight' in the text (See section 3.1.4, page 11, line 30)

**2)**

**There are many instances where the authors state that the model can "reproduce" observed events. It would be better to replace "reproduce" (which sounds like an exact copy) with "reasonably simulate or capture".**

We have replaced the word 'reproduce' either with 'capture' or 'simulate' at all instances in the revised manuscript. Please see the manuscript with tracked changes.

**3)**

**Regarding the definition of E-index and C-index, in the original definition by Takahashi et al. 2011, the first and second principal components PC1, PC2 are first normalized before calculating the E-Index, C-Index. Please ensure that this is mentioned. Takahashi et al. 2011 defined the E-Index and C-Index; Cai et al. 2018 applied these indices.**

In the revised manuscript we have cited Takahashi et al. (2011) at the relevant places. We have made the following changes:

*Based on Empirical Orthogonal Function Analysis (EOF) of Sea Surface Temperature (SST) in the tropical Pacific (see Takahashi et al., 2011), ENSO can be contrasted into two distinct modes of variability, i.e. eastern and central Pacific ENSO modes (Kao and Yu, 2009; Yu and Kim, 2010; Xie and Jin, 2018).* (See section 1, from page 2 and line 41 to page 3, line 4)

We have cited Takahashi et al. (2011) in the following text as well:

*The PCA is also useful for evaluating how well HadCM3L represents certain types of ENSO events. Eastern and central Pacific ENSO events can be described by an E-Index (PC1-PC2)/√2; Takahashi et al., 2011), which emphasises maximum warm anomalies in the eastern Pacific region (Cai et al., 2018), and a C-Index (PC1+PC2)/√2; Takahashi et al., 2011) respectively, which focuses on maximum warm anomalies in the central Pacific (Cai et al., 2018).* (See section 2.4, page 8, lines 11-16)

In caption of supplementary Figure S1 we have added the following text:

*The red line in m-n shows a quadratic fit between PC1 and PC2 averaged over DJF. Grey dots show monthly data whereas black dots indicate data averaged over DJF. EOF analysis is performed over the region 15$^o$ N-15$^o$ S and 140$^o$ E-80$^o$ W (Cai et al., 2018). Before analysis and calculating E- and C-index (Takahashi et al., 2011), PC1 and PC2 are normalized by their monthly standard deviations calculated over the corresponding observational and model simulation period.* (See Fig. S1, Supplementary page 1, lines 11-16)

**4)**

**P10, L24-40: On definition of Westerly Wind Burst and Easterly Wind Burst. These winds are ought to be identified using daily wind data (e.g., in Hu and Fedorov 2016). If this is not the case, please reword, e.g., "although not explicitly diagnosed, WWB and EWB are contained respectively in the positive and**

**negative values of this wind index." Then please be careful with calling them WWBs and EWBs in the rest of the manuscript.**

We have not omitted the use of WWBs and EWBs. Since these bursts can last for 5-40 days, thus the monthly data, which we have used in our analysis, includes monthly averages of these bursts. However, we have cited Hu and Fedorov, (2016) who calculated these bursts from daily data. In the revised manuscript we have added the following text:

*Although here not explicitly diagnosed through daily data, WWBs and EWBs are contained respectively in the positive and negative values of this wind stress index (see Hu and Fedorov, 2016). As the duration of WWBs is 5 to 40 days (Gebbie et al., 2007), the monthly mean data of westerly wind stress includes a monthly average of these bursts.* (See section 3.1.3, page 10, lines 37-41)

**5)**

**P15, L10-12: "Note that Wang et al. (2020) showed that extreme convective events can still happen even if the E-index is not greater than 5 mm day-1 (cf. 12 Figure 2 in Wang et al. 2020)." – E-index cannot be in the unit of mm/day.**

In the revised manuscript we have rephrased the text as follows:

*Note that Wang et al. (2020) showed that extreme El Niño events having E-Index > 1.5 s.d. can still happen even if the Niño3 rainfall is not greater than 5 mm day-1 (cf. Figure 2 in Wang et al., 2020).* (See section 3.2.2, page 15, lines 25-27)

**6)**

**P15, L24-41: On La Nina frequency change. The fact that there are no extreme La Nina events in 4xCO2 experiment is inconsistent with Cai et al. 2015. A remark on this is necessary to avoid confusion to description in L34-41 on G1.**

We have added the following sentence in the revised manuscript:

*Our findings are inconsistent to those of Cai et al. (2015b) who found nearly doubling of extreme La Nina events under increased GHG forcing.* (See section 3.2.3, page 16, lines 3-5)

**7)**

**P17, L3-23: Increased upper ocean stratification tends to enhance the Bjerknes feedback, likely through the coupling between the wind and thermocline. This is not yet diagnosed in this present analysis which instead represents the Bjerknes feedback solely on the coupling between SST and wind. The Bjerknes feedback has many components (e.g., Kim and Jin 2011), and some may increase and some may decrease under external forcing. It would be good to put a caveat like this in this paragraph.**

We have added the following paragraph in Sect. 5:

*Bjerknes feedback is a multi-component process (e.g., Kim and Jin, 2011a), where some components may increase and some may decrease under the influence of external forcing. For instance, increased upper ocean stratification tends to enhance the Bjerknes feedback, likely through coupling between the wind and thermocline. However, this study represents the Bjerknes feedback solely on the coupling between wind and SST, a caveat of this analysis.* (See section 4.1, from page 17 and line 39 to page 18 line 4)

**8)**

**The curve fitting in Fig. S1 (red curve) does not look a smooth parabolic.**

In the revised manuscript we have replotted the red curves, hope it looks okay now. (See supplementary Fig. S1, page 2)

**Referee #2**

**Minor Revisions**

**1)**

**P3 line3 the meaning of "until Cai et al. (2018) used SST indices basedon Principal Component Analysis (PCA)." Is not clear.**

We have rephrased the text as follows:

*As diagnosed from SST indices in state-of-the-art AOGCMs, there was no intermodel consensus about change in frequency of ENSO events and amplitude in a warming climate (Vega-Westhoff and Sriver, 2017; Yang et al., 2018). However recently, Cai et al. (2018), using SST indices based on Principal Component Analysis (PCA), showed an enhanced frequency of extreme El Niño events and strengthening of ENSO amplitude under increased GHG forcing.* (See section 1, page 3, lines 8-13)

**2)**

**P3 line 13-14 the meaning of eastern and central Pacific ENSO mode should be clarified in the text somewhere(see studies in Wang et al., 2019).**

In the revised manuscript, we have modified the text as follows:

*Based on Empirical Orthogonal Function Analysis (EOF) of Sea Surface Temperature (SST) in the tropical Pacific (see Takahashi et al., 2011), ENSO can be contrasted into two distinct modes of variability, i.e. eastern and central Pacific ENSO modes (Kao and Yu, 2009; Yu and Kim, 2010; Xie and Jin, 2018). The eastern Pacific ENSO mode (EOF1) shows maximum SST anomaly in the eastern equatorial Pacific (Niño3 region: 5$^o$ N-5$^o$ S; 150$^o$ W-90$^o$ W) whereas the central Pacific ENSO mode (EOF2) indicates maximum SST anomaly in the central Pacific (Niño4 region: 5$^o$ N-5$^o$ S; 160$^o$ E-150$^o$ W) (Kao and Yu, 2009; Cai et al., 2018).* (See section 1, from page 2 and line 41 to page 3 and line 7)

**3)**

**P3 line18 "a significant mean warming response" might be better replaced as "a significant mean state warming response".**

In the revised manuscript, we have replace "a significant mean warming response" with "a significant mean state warming response". (See section 1, page 3, lines 27-28)

**4)**

**P3 line 20 "CMIP 3" should be "CMIP3".**

The mentioned acronym is corrected in the revised manuscript. (See section 1, page 3, line 29)

**5)**

**P3 line 39"argue" should be "argued".**

Corrected. (See section 1, page 4, line 6)

**6)**

**P3 line 42 "90 %" should be "90%".**

Corrected (See section 1, page 4, line 9). We have also corrected it at all other instances in the revised manuscript. Please see manuscript with tracked changes.

**7)**

**P6 lines14-15 "BJ feedback is an equatorial zonal wind stress dynamic response to equatorial SST anomalies." might be revised as "BJ feedback is a dynamical response of equatorial zonal wind stress to equatorial SST anomalies." for clarity.**

In light of the comment we have modified the text as follows:

*BJ feedback is a dynamical response of equatorial zonal wind stress to equatorial SST anomalies.* (See section 2.3, page 6, lines 24-25)

**8)**

**P14 lines 5-6, the definition of extreme events is not clear, do you mean the averaged rainfall anomalies over the Nino3 region exceeding 5 mm/day? Why 5 mm/day in Cai et al. (2014) as the threshold? This should be mentioned and clarified. Is it the same reason as Wang et al. (2020)? Thus, the first paragraph in section 3.2.2 can be better organized.**

No these are not rainfall anomalies, we define an extreme El Nino event for which averaged DJF Niño3 total rainfall exceeds 5 mm day$^{-1}$. Cai et al. (2014, 2017) used the same definition. However, we have tried to make it clear by modifying the text as follows:

*We choose a threshold value of rainfall for defining extreme El Niño events based on the work of Cai et al., (2014, 2017), who chose averaged DJF Niño3 total rainfall exceeding 5 mm day$^{-1}$ for this threshold based on observations.* (See section 3.2.2, page 14, lines 20-22)

Regarding 2$^{nd}$ part of the comment that we should give a reason for using 5 mm/day as an extreme El Nino event threshold, the reason is already mention in section 2.3 (Definitions and statistical tests). We have not repeated the reason in section 3.2.2 due to redundancy. Please see the following text in Sect. 2.3, page 6, lines 8-13.

*The Niño3 index is chosen for studying the characteristics of extreme El Niño events since during an extreme El Niño event, following the highest SSTs, convective activity moves towards the eastern Pacific, and the ITCZ moves over the Niño3 region resulting in rainfall higher than 5 mm day$^{-1}$ (Cai et al., 2014). Similar to Cai et al. (2014, 2017) events with Niño3 rainfall greater than 5 mm day$^{-1}$ are considered extreme El Niño events,.....*

Further see Sect. 2.4, page 7, lines 17-20, as follows:

*During extreme El Niño events, the ITCZ moves equatorward, causing significant increases in rainfall (> 5 mm day$^{-1}$) over the eastern equatorial Pacific that skews the statistical distribution of rainfall in the Niño3 region.*

**9)**

**In section5, the possible implications of CP ENSO frequency and amplitude changes due to atmospheric and oceanic changes under 4×CO2 and G1 scenarios should be discussed. The formation of EP and CP ENSO can be distinct since BJ feedback and heat flux feedback can play a relatively different role in determining the evolution of ENSO events. As inferred from the results based on 4×CO2 and G1 simulations, how might the CP ENSO be changed?**

We have added the following paragraph in Sect. 5:

*The changes in ENSO feedbacks and more stratified ocean temperatures under both 4×CO2 and G1 can also affect the eastern and central Pacific ENSO variability differently. For instance, more stratified ocean and enhanced BJ feedback in G1 strengthens the eastern Pacific ENSO amplitude but not central Pacific ENSO amplitude (Table 1-2). Similarly, the enhanced hf and weaker BJ feedback in 4×CO2 results in a more substantial reduction in central Pacific ENSO amplitude than eastern Pacific ENSO amplitude (Table 1-2). In the current model system, we expect that changes in tropical Pacific mean state and feedback process, both under 4×CO2 and G1, may impact the occurrence ratio of central Pacific El Niño (La Niña) to eastern Pacific El Niño (La Niña) (e.g., Yeh et al., 2009), which requires further detailed analysis.* (See section 5, page 19, lines 32-41)

**Other Minor Changes that we have Made**

**1)**

In the revised manuscript '~50-yrs' replaced with '~50-year' (See page 1, line 24)

**2)**

Ammedded the text **'***Cai et al. (2014)***'** as *'Cai et al. (2014, 2017)'*. (See page 6, line 11-12)

**3)**

Some typographical errors were found in Table S3, so we have modified Table S3. In the previous version this sentence *'Based on the E-index definition, we also see a statistically significant increase in the total number of El Niño events in $4 \times CO_2$ (88%) and G1 (12 %) (Table S3).'* is thus replaced with *'Based on the E-index definition, we see a statistically significant increase in the total number of El Niño events in $4 \times CO_2$ (107%) and no statistically significant change in G1 (Table S3).'* (See page 15, lines 23-25 or see manuscript with tracked changes)

**4)**

In the acknowledgement we have added the following sentence:

*The authors thank the referees for their comments and suggestions, which have much helped us to improve our manuscript.* (See page 21, lines 16-17)

**5)**

Some references were not in accordance with the journal's prescribed format; we have modified them according to the journal's instructions. Please see manuscript with tracked changes.

**6)**

In Fig. 4a,b the longitudinal label '80$^o$ W' was incorrectly labelled as '140$^o$ W', so we have corrected it.

**7)**

Some citations in the text were not in accordance with the journal's prescribed format; we have modified them according to the journal's instructions. Please see manuscript with tracked changes.

**8)**

In the revised manuscript *'5mm day$^{-1}$'* is replaced with *'5 mm day$^{-1}$'*. (See page 6, lines 11)

**9)**

In caption of Fig. S1 'ER5' is corrected as 'ERA5' and 'g-l' is corrected as 'g-i'. (See supplementary page 2, line 9 and 10)

**10)** For other minor changes please see the manuscript with tracked changes.

[revised manuscript text omitted]